# Protein mimetic amyloid inhibitor potently abrogates cancer-associated mutant p53 aggregation and restores tumor suppressor function

L. Palanikumar [1], Laura Karpauskaite [1], Mohamed Al-Sayegh [1], Ibrahim Chehade [1], Maheen Alam [2], Sarah Hassan [1], Debabrata Maity[3], Liaqat Ali[4], Mona Kalmouni [1], Yamanappa Hunashal[5,6], Jemil Ahmed[7], Tatiana Houhou[1], Shake Karapetyan[8], Zackary Falls [9], Ram Samudrala [9], Renu Pasricha [4], Gennaro Esposito[5,10], Ahmed J. Afzal [1], Andrew D. Hamilton [3 ✉], Sunil Kumar [7 ✉] & Mazin Magzoub [1 ✉]

Missense mutations in p53 are severely deleterious and occur in over 50% of all human cancers. The majority of these mutations are located in the inherently unstable DNA-binding domain (DBD), many of which destabilize the domain further and expose its aggregation-prone hydrophobic core, prompting self-assembly of mutant p53 into inactive cytosolic amyloid-like aggregates. Screening an oligopyridylamide library, previously shown to inhibit amyloid formation associated with Alzheimer's disease and type II diabetes, identified a tripyridylamide, ADH-6, that abrogates self-assembly of the aggregation-nucleating sub-domain of mutant p53 DBD. Moreover, ADH-6 targets and dissociates mutant p53 aggregates in human cancer cells, which restores p53's transcriptional activity, leading to cell cycle arrest and apoptosis. Notably, ADH-6 treatment effectively shrinks xenografts harboring mutant p53, while exhibiting no toxicity to healthy tissue, thereby substantially prolonging survival. This study demonstrates the successful application of a bona fide small-molecule amyloid inhibitor as a potent anticancer agent.

[1] Biology Program, Division of Science, New York University Abu Dhabi, Saadiyat Island Campus, Abu Dhabi, United Arab Emirates. [2] Department of Biology, SBA School of Science and Engineering, Lahore University of Management Sciences, Lahore, Pakistan. [3] Department of Chemistry, New York University, New York, NY, USA. [4] Core Technology Platforms, New York University Abu Dhabi, Saadiyat Island Campus, Abu Dhabi, United Arab Emirates. [5] Chemistry Program, Division of Science, New York University Abu Dhabi, Saadiyat Island Campus, Abu Dhabi, United Arab Emirates. [6] DAME, Università di Udine, Udine, Italy. [7] Department of Chemistry and Biochemistry and Knoebel Institute for Healthy Aging, The University of Denver, Denver, CO, USA. [8] Physics Program, Division of Science, New York University Abu Dhabi, Saadiyat Island Campus, Abu Dhabi, United Arab Emirates. [9] Department of Biomedical Informatics, School of Medicine and Biomedical Sciences, State University of New York (SUNY), Buffalo, NY, USA. [10] INBB, Rome, Italy. ✉email: andrew.hamilton@nyu.edu; sunil.kumar97@du.edu; mazin.magzoub@nyu.edu

Dubbed the "guardian of the genome"[1], p53 is a tumor suppressor protein that is activated under cellular stresses, including DNA damage, oncogene activation, oxidative stress, or hypoxia[2,3]. Under normal conditions, p53 levels are kept low by its negative regulator, the E3 ubiquitin ligase MDM2, which targets p53 for proteasome-mediated degradation[2,4]. The aforementioned cellular stresses disrupt the p53–MDM2 interaction, via phosphorylation of both proteins, and stimulate p53 acetylation, leading to its accumulation and activation[4]. Activated p53 then triggers DNA damage repair, cell cycle arrest, senescence, apoptosis, or autophagy, all of which are directed towards the suppression of neoplastic transformation and inhibition of tumor progression[2,3,5].

p53 binds to several DNA sequences, functioning as a sequence-specific transcriptional activator[3,6]. p53 is also characterized by a high degree of structural flexibility, which facilitates its interactions with a myriad of protein partners, allowing it to exert its function as a master regulator of the cell[6]. Crucially, p53 missense mutations are found in over half of all human cancers, making it the most mutated protein in cancer, and these mutations are associated with some of the most pernicious manifestations of the disease[3,7]. Thus, p53 has taken on a pivotal role in the realm of cancer research and is considered a key target in the development of cancer therapeutics[4,7].

Under physiological conditions, p53 exists as a homotetramer, with each monomer composed of globular DNA-binding and tetramerization domains, connected by a flexible linker and flanked by intrinsically disordered regions (a transactivation domain followed by a proline-rich region at the N terminus, and a C-terminal regulatory domain) (Fig. 1a)[8]. The DNA-binding domain (DBD) consists of a central immunoglobulin-like β-sandwich serving as a scaffold for the DNA-binding surface, which is composed of a loop–sheet–helix motif and two large loops that are stabilized by the tetrahedral coordination of a single zinc ion (Fig. 1b)[8]. A majority (~90%) of cancer-associated p53 mutations occur within the DBD, where they cluster into discernible "hotspots"[9,10], resulting in the protein's inactivation through alterations in residues that are crucial for either DNA interactions (contact mutants) or proper folding (structural mutants), although it is now apparent that some mutations (such as R248W) possess both characteristics[6].

Several studies have reported that various p53 DBD mutants, along with fragments of these proteins, form amyloid-like aggregates in solution, cancer cell lines, and tumors[11,12]. p53 DBD is characterized by low thermodynamic and kinetic stability[8], and mutations in the domain often decrease its stability further and prompt its unfolding, which leads to exposure of its hydrophobic core[13,14]. Furthermore, many of the DBD mutations, including the commonly occurring R248W/Q, R273C/H, and R175H[10], involve replacing the cationic arginine, a so-called "gate-keeper" amino acid that prevents protein aggregation via the repulsive effect of its charge[15], with residues (tryptophan, glutamine, cysteine, or histidine) that have a high aggregation/amyloidogenic potential[15,16]. Thus, these DBD mutations serve to not only expose the hydrophobic core of the domain, but also to enhance its aggregation propensity. This prompts self-assembly of mutant p53 into amyloid-like aggregates within inactive cellular inclusions that incorporate the wild-type (WT) isoform, thereby blocking the protein's tumor suppressor function[17].

Increasing evidence implicates aggregation of mutant p53 (e.g. R248Q, R248W, and R175H) in the associated oncogenic gain-of-function (GoF), i.e. the acquisition of activities that promote tumor growth, metastasis and chemoresistance[11,12,18]. For instance, co-sequestration of mutant and WT p53 into inactive cellular inclusions may result in overexpression of antiapoptotic and pro-proliferative genes previously repressed by p53 (refs. [3,5]).

Aggregation of mutant p53 also induces misfolding of the p53 paralogs p63 and p73, which are then incorporated into the inclusions, facilitated by interactions of the aggregation-prone core of p53 DBD with near identical segments present in the p63 and p73 DBDs[17]. p63 and p73, which are rarely mutated in tumors, have partial functional overlap with p53 (refs. [6]). However, coaggregation with mutant p53 suppresses the regulatory functions of p63 and p73, resulting in deficient transcription of target genes involved in cell growth control, and apoptosis, which leads to uncontrolled proliferation, invasion, and metastasis[17]. Additionally, aggregation of mutant p53 has been shown to induce overexpression of heat-shock proteins, in particular Hsp70 (ref. [17]), that promote tumor cell proliferation and inhibit apoptosis[19]. Thus, amyloid-like aggregation of mutant p53 may contribute to both its loss of tumor suppressor function and its oncogenic GoF.

Of relevance, replacing a hydrophobic amino acid in the aggregation-prone core of p53 DBD with the "gate-keeper" arginine residue (I254R) abrogates coaggregation of mutant p53 with the WT protein and its paralogs, p63 and p73, as well as abolishes overexpression of Hsp70 (ref. [17]). Notably, a p53 DBD-derived peptide harboring the aggregation-suppressing I254R mutation (denoted ReACp53; Fig. 1c) was shown to block mutant p53 aggregation by masking the aggregation-prone core, which restored the mutant protein to a WT p53-like functionality and reduced cancer cell proliferation in vitro and halted tumor progression in vivo[14]. These studies indicate that targeting mutant p53 aggregation is a viable and effective cancer therapeutic strategy.

We previously reported the use of oligopyridylamide-based α-helix mimetics to effectively modulate self-assembly of the amyloid-β peptide (Aβ)[20,21] and islet amyloid polypeptide (IAPP)[22,23], which are associated with Alzheimer's disease (AD) and type II diabetes (T2D), respectively. α-Helix mimetics are small molecules that imitate the topography of the most commonly occurring protein secondary structure, serving as effective antagonists of protein–protein interactions (PPIs) at the interaction interface[24,25]. The appeal of α-helix mimetics stems from the fact that their side-chain residues can be conveniently manipulated to target specific disease-related PPIs[24,25]. In this study, we explored whether such a protein mimetic-based approach can be extended towards mutant p53 self-assembly. To that end, we asked the following questions: (i) if intrinsically disordered mutant p53 does indeed aggregate via an amyloid pathway, can the oligopyridylamide-based α-helix mimetics effectively abolish this process; (ii) if successful, does oligopyridylamide-mediated abrogation of mutant p53 aggregation lead to rescue of p53 function and inhibition of cancer cell proliferation in vitro; and (iii) can this oligopyridylamide-based strategy be applied to reverse tumor growth in vivo without adversely affecting healthy tissue? In addressing these questions, we establish the potential of using functionalized amyloid inhibitors as mutant p53-targeted cancer therapeutics.

## Results

**ADH-6 abrogates amyloid formation of the aggregation-nucleating sequence of p53 DBD**. We began with a reductionist approach commonly adopted in the amyloid research field, namely to target a short aggregation-prone segment within the protein of interest[26]. Across most amyloid systems, aggregation can be nucleated in a structure-specific manner by a small stretch of residues, which has a robust independent capacity for self-assembly[26,27]. Aggregation prediction algorithms developed from biophysical studies of amyloids identified an aggregation-nucleating subdomain (p53 residues 251–258) in the hydrophobic core of the p53 DBD

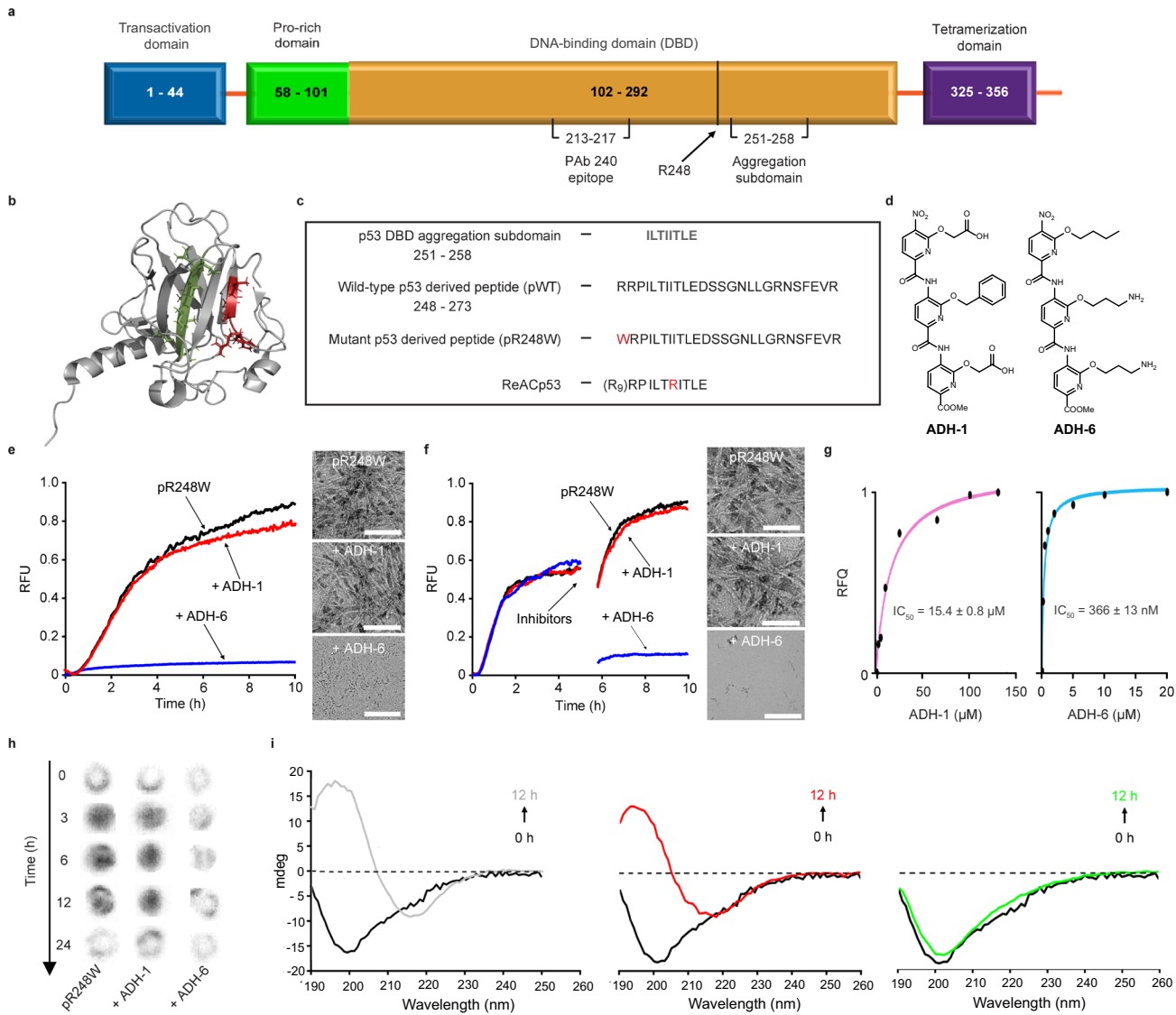

**Fig. 1 ADH-6 abrogates amyloid formation of aggregation-prone region of p53 DBD. a** Schematic representation of the different domains of p53. The DBD (residues 102–292) contains an aggregation-nucleating subdomain (residues 251–258) that is necessary and sufficient to drive p53 aggregation[14,17,28]. Another segment of interest comprises residues 213–217, which is the antigen recognized by the PAb 240 antibody that binds to partially unfolded p53. Also highlighted in the DBD is R248, one of the most common mutation hotspots in p53 (IARC TP53 database; https://p53.iarc.fr)[9]. **b** Structure of p53 DBD. Highlighted are the aggregation-nucleating subdomain (green) and the epitope recognized by PAb 240 (red). Both segments are buried in the fully folded p53 structure. The 3D image was generated using PyMOL 2.3.5 (Schrödinger, New York, NY). **c** Primary sequences of the studied WT and mutant R248W p53 DBD-derived peptides, denoted pWT and pR248W, respectively, which span residues 248–273. The peptides include the aggregation-prone 252–258 sequence, as well as R248 and another of the most common mutation hotspots in p53 and R273 (IARC TP53 database; https://p53.iarc.fr)[9]. **d** Chemical structures of the oligopyridylamides ADH-1 and ADH-6. **e, f** Effects of the oligopyridylamides on pR248W amyloid formation. Kinetic profiles (left panel) and representative transmission electron microscopy (TEM) images (right panel) for aggregation of 25 μM pR248W in the absence or presence of an equimolar amount of ADH-1 or ADH-6 co-mixed at the start of the reaction (**e**) or added during the growth phase (i.e. 5 h after the start of the reaction) (**f**). Kinetic aggregation profiles were acquired by measuring the fluorescence of the thioflavin T (ThT) reporter ($\lambda_{ex/em} = 440/480$ nm) at 5-min intervals at 37 °C ($n = 4$). TEM images were acquired at 10 h after the start of the aggregation reaction. Scale bar = 100 nm. **g** Characterization of the binding interaction of the oligopyridylamides and pR248W measured using steady-state intrinsic tryptophan fluorescence quenching. A 5 μM solution of pR248W was titrated with increasing concentrations of ADH-1 (left panel) or ADH-6 (right panel) and the tryptophan fluorescence after each addition was normalized to account for the dilution (total dilution during the titration was <1%) and plotted against the ligand concentration. The equilibrium dissociation constants ($K_d$) were then determined using a one-site-specific binding equation (Eq. 1). **h** Effects of the oligopyridylamides on pR248W oligomerization monitored using the dot blot assay. Samples of 10 μM pR248W were incubated with or without an equimolar amount of ADH-1 or ADH-6 for 0–24 h, and the presence of oligomers was detected using an amyloid oligomer-specific polyclonal antibody (A11)[35]. **i** Effects of the oligopyridylamides on the self-assembly driven structural transition of pR248W. Time-dependent circular dichroism (CD) spectra of 10 μM pR248W alone (left panel) or in the presence of an equimolar amount of ADH-1 (middle panel) or ADH-6 (right panel).

that is necessary and sufficient to drive p53 aggregation[14,17,28] (Fig. 1a, c). This segment forms a β-strand within the hydrophobic core of the DBD (Fig. 1b). However, many of the mutations in the p53 DBD destabilize its inherently unstable tertiary structure further[29] and increase exposure of the aggregation-nucleating sub-domain which, in turn, promotes aggregation of the protein[14,17]. Therefore, we chose to focus on p53$_{251-258}$ but expanded the sequence to include two of the most common mutation hotspots, R248 and R273 (IARC TP53 database; https://p53.iarc.fr)[9]. Two peptides were synthesized, one corresponding to residues 248–273 of WT p53 DBD (peptide denoted pWT), the other composed of the same sequence but harboring the R248W mutation (peptide denoted pR248W) (Fig. 1c). R248W is one of the most common p53 DBD mutation that occurs in a range of malignancies, including pancreatic cancer[9,10]. Pancreatic cancer is an intractable malignancy that often evades early diagnosis and resists treatment, and is consequently associated with a very poor prognosis: it is the seventh most common cause of death from cancer worldwide, with a five-year survival rate of <5%[30,31].

We tested the effects of 10 compounds (ADH-1–10), based on the same oligopyridylamide molecular scaffold (Fig. 1d and Supplementary Fig. 1a), on the aggregation of pR248W using the thioflavin T (ThT)-based amyloid kinetic assay[32]. The compounds were selected based on their distinct chemical fingerprints and their ability to modulate amyloid assemblies[20,21,23]. The p53 DBD-derived peptides alone are characterized by a sigmoidal ThT curve, which is indicative of a nucleation-dependent process typical for amyloids (Supplementary Fig. 1b)[33]. Comparing the two DBD-derived peptides, pR248W exhibited the greater aggregation propensity, as evidenced by the shorter lag phase, more rapid elongation phase and higher final ThT fluorescence intensity. This is not surprising given that the mutation involves replacing the cationic arginine, an aggregation "gate-keeper" residue[15] with the hydrophobic tryptophan, an aromatic residue with the highest amyloidogenic potential of all 20 naturally occurring amino acids[16]. Transmission electron microscopy (TEM) imaging confirmed that the aggregation-prone segment of mutant p53 DBD does indeed form fibrils (Fig. 1e, f).

The 10 oligopyridylamides varied in their antagonist activity against pR248W aggregation. While the anionic ADH-1 did not significantly affect pR248W's self-assembly, the cationic ADH-6 completely inhibited the peptide's amyloid formation, as indicated by both the ThT assay and TEM imaging (Fig. 1e). The determinants of efficacy of the oligopyridylamides appear to be the number and positioning of cationic sidechains (Fig. 1d, e and Supplementary Fig. 1a, b), suggesting that inhibition of pR248W aggregation occurs through specific interactions involving the compounds' cationic sidechains. A possibility is that the oligopyridylamide–pR248W binding is stabilized by cation–π interactions of a cationic sidechain of the protein mimetic and the aromatic tryptophan residue of the mutant peptide[34], which provides a basis for strong binding and confers a degree of specificity to the interaction. This is strongly supported by the much higher binding affinity of ADH-6 ($K_d = 366 \pm 13$ nM) compared to ADH-1 ($K_d = 15.4 \pm 0.8$ μM) for pR248W (Fig. 1g). Similar cation–π interactions between the cationic sidechains of ADH-6 and one or more of the several aromatic residues (tyrosines and tryptophans) present in the DBD[9] may facilitate strong binding of the oligopyridylamide to full-length mutant p53.

In order to restore WT p53-like activity in cancer cells harboring aggregation-prone mutant p53, potential therapeutics would need to dissociate pre-formed mutant p53 aggregates, as well as prevent additional aggregation. Therefore, we tested the capacity of ADH-6 to abrogate pre-formed pR248W aggregates. Addition of ADH-6 to the mutant peptide at the mid-point of the aggregation reaction, when a significant amount of fiber formation had already taken place (Supplementary Fig. 1c), resulted in a marked decrease in the ThT fluorescence intensity and near complete absence of fibers in the TEM images at the end of the reaction (Fig. 1f). The very few fibers that were detected were much shorter and thinner than those observed for the peptide alone (Fig. 1f and Supplementary Fig. 1d).

To further confirm the capacity of ADH-6 to prevent aggregation of pR248W, we evaluated the effect of the oligopyridylamide on the peptide's oligomerization using a dot blot immunoassay (Fig. 1h). pR248W was incubated with or without an equimolar amount of ADH-6 or ADH-1 for 0–12 h and then detected using A11, an antibody specific for amyloid oligomers, including those of mutant p53 (refs. [35,36]). The chemiluminescence signal intensity for samples of pR248W alone increased from 0 to 6 h, indicating an increase in the amount of soluble oligomers. Subsequently, the intensity diminished significantly at 12 h due to conversion of the oligomers into fibrils. Treatment with ADH-1 did not significantly change the intensity of the dots, indicating that the oligopyridylamide did not affect oligomerization of pR248W. In marked contrast, in the presence of ADH-6 no formation of pR248W oligomers was observed, as reflected by the weak intensity of the dots throughout the time course of the experiment. Finally, we probed the effect of ADH-6 on the secondary structure of pR248W using CD spectroscopy (Fig. 1i). The peptide transitioned from a random coil monomer to a β-sheet structure as it self-assembled into amyloid fibers. pR248W underwent the same conformational transition in the presence of an equimolar amount of ADH-1, which is not surprising given its inability to inhibit the mutant peptide's oligomerization or amyloid formation (Fig. 1e, f, h). On the other hand, in the presence of ADH-6 at an equimolar ratio, pR248W remained in its native conformation for the duration of the experiment, which confirms that ADH-6 potently inhibits self-assembly of pR248W (Fig. 1e, f, h). Taken together, these results demonstrate that ADH-6 not only strongly inhibits self-assembly of the aggregation-prone segment of mutant p53 DBD, the oligopyridylamide also effectively dissociates pre-formed aggregates of the segment and prevents further aggregation.

**Nuclear magnetic resonance (NMR) spectroscopy characterization of p53 DBD–ADH6 interaction interface.** NMR spectroscopy was used to elucidate the p53 DBD–ADH-6 interactions. Figure 2 shows the overlay of the heteronuclear single quantum coherence (HSQC) maps, with and without oligopyridylamide, for WT (Fig. 2a) and mutant R248W (Fig. 2b) p53 DBDs. Chemical shift perturbation (CSP) analysis was carried out to determine the protein–ligand interaction interface. The interaction with ADH-6 also involved partially unfolded species that were present in the samples of both DBD variants (Fig. 2a, b). Due to this additional involvement, no quantitative estimate was reliably feasible to assess a binding constant of ADH-6 to WT and R248W p53 DBDs. A qualitatively equivalent pattern was observed with either DBD variant (see Supplementary Section 2). The increment of the CSP values with ligand concentration (Fig. 2c) is the signature of a fast exchange regime, i.e. no stable complex formation, without any major favorable or adverse consequence of the hotspot mutation[37] for ADH-6 affinity. This result does not necessarily conflict with the nanomolar affinity of ADH-6 for pR248W inferred from the fluorescence quenching experiments (Fig. 1g) because p53 DBD presents numerous interaction sites for ADH-6 (vide infra) beyond the aggregation-nucleating subdomain. However, it is also possible that the fast exchange regime observed by NMR may be, in part, a consequence of the experimental conditions used to facilitate study of the inherently unstable p53 DBDs[37,38] (Supplementary Section 2).

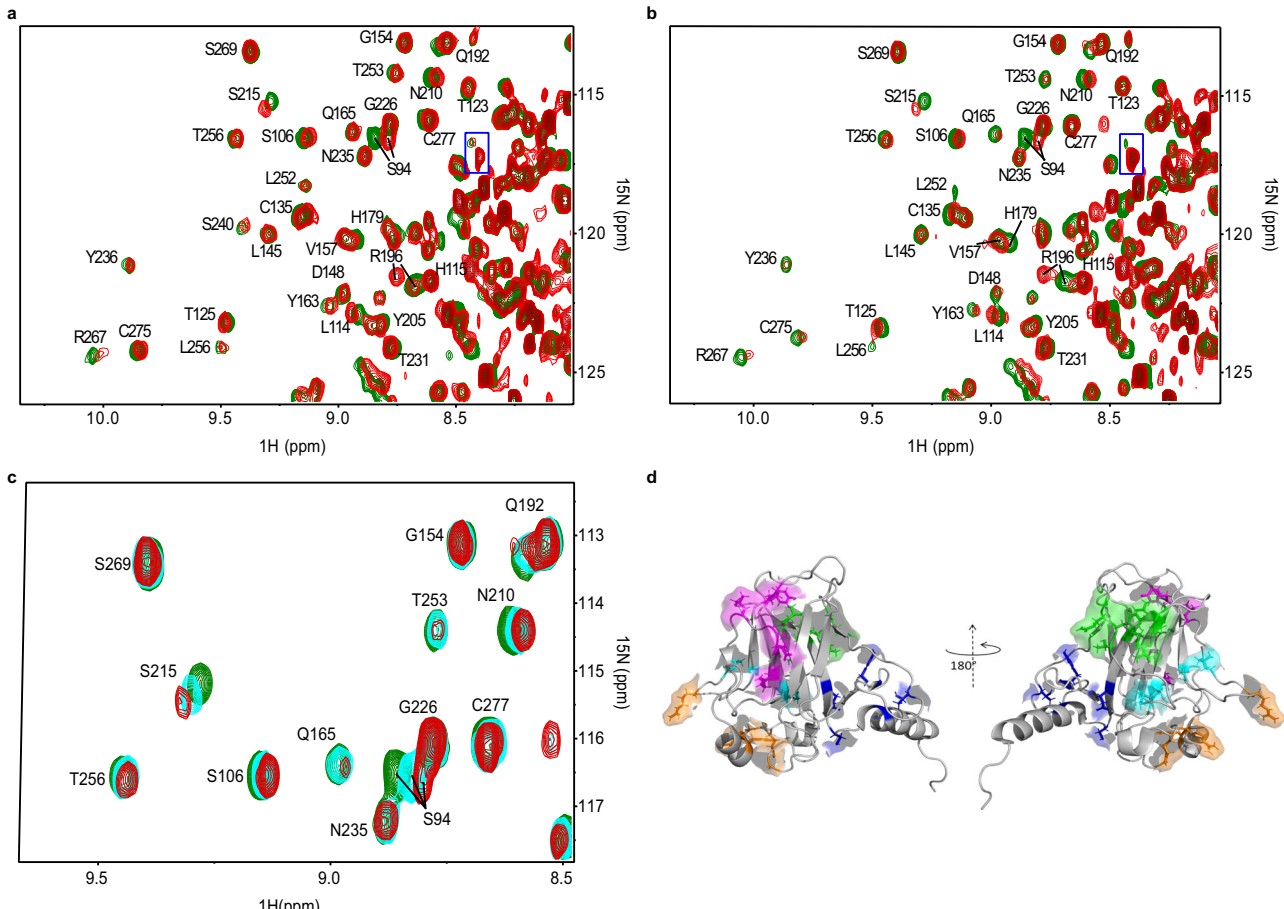

**Fig. 2 NMR-based determination of p53 DBD–ADH-6 interaction interface. a, b** Overlay of $^{15}$N-$^{1}$H HSQC maps of 19 µM WT (**a**) and 24 µM R248W (**b**) p53 DBD in $H_2O/D_2O$ (96/4) with 16.7 mM DTT, without (green contours) or with (red contours) ADH-6 addition (protein:ligand 1:11 in **a** and 1:15 in **b**). The assignments are reported only outside the rightmost regions. These regions are crowded because of the presence of partially unfolded species that also interact with ADH-6 as highlighted by the boxed peak in each panel. **c** HSQC contour maps overlay of mutant R248W p53 DBD at different protein: ADH-6 ratios (1:0 green, 1:8 cyan, and 1:15 red) showing the increment of cumulated chemical shift perturbation (CSP) with ligand concentration (Eq. 2). **d** The five clusters of the two p53 DBD variants (WT and mutant R248W) that show high (>0.025) or medium (>0.015) CSP values[137]. Cluster 1 (highlighted in blue) includes residues T118, Y126, E271, C275, and G279; cluster 2 (highlighted in magenta) includes residues R196, E198, G199, L201, Y220, and E221; cluster 3 (green) includes T102, Y103, Q104, G105, L257, L264, and R267; cluster 4 (orange) includes E171, R174, H179, R209, and G244; and cluster 5 (cyan) includes S94, A161, I162, L206, and S215. Clusters 1 and 2 are at the front in the cartoon on the left; clusters 3–5 are at the front in the cartoon on the right. The 3D image was generated using PyMOL 2.3.5 (Schrödinger, New York, NY).

The backbone NH signals that exhibit the most significant CSP values in both the considered p53 DBD variants appeared to involve five clusters of residues that are highlighted with different colors in Fig. 2d. The first cluster (highlighted in blue) encompasses the area around the start of the C-terminal helical fragment of the p53 DBD (helix 3, PDB:*2fej*[38]). The second cluster (magenta) extends along strand 4 of sheet A and the subsequent apical loop. The third cluster (green) is formed by a sequential stretch of the apical loop preceding strand 1 of sheet A along with the initial and final facing fragments of strands 3 and 4 in sheet B, respectively. The fourth cluster (orange) covers an extended surface surrounding the start of helix 2. And the fifth, and final, cluster (cyan) is assembled with contributions from very distant residues, i.e. N-terminal and start and end of strands 6 and 7, respectively, of sheet B. Interestingly, cluster 1 includes E271 and cluster 3 includes L257, L264, and R267, i.e. residues of the p53$_{248–273}$ segment we selected for our DBD-derived peptides, pWT and pR248W (Fig. 1c), which has a high frequency of mutations affecting p53 function[14,37,38]. In particular, L257 of cluster 3 belongs to the aggregation-nucleating subdomain

(p53$_{251–258}$) that promotes amyloid-like aggregation of mutant p53 (refs. [14,37]).

The conspicuous extension of the protein surface region that is affected by ADH-6 interaction, based on the NMR evidence, should not suggest the occurrence of a rather generic interaction altogether. Approximately 75% of residues involved in the clusters of Fig. 2d are predicted to be aggregation prone[14], which provides a basis for the effect of ADH-6 on sensitive locations other than p53$_{251–258}$ in p53 DBD. Chemical shift changes, however, may also be the consequence of local rearrangements in response to an allosteric interaction. Therefore, only part of the highlighted surface regions of Fig. 2d may actually be directly involved in the dynamic contacts with ADH-6. A possible clue for the determination of these contacts may come from the inspection of the relative peak intensities (RI), i.e. the ratio of the peak heights in the presence and absence of ADH-6 (Supplementary Section 2). Dynamic interactions may in fact induce intensity attenuations because of exchange and/or dipolar broadening, thereby providing a further interpretation tool. Unambiguous attenuations of the intensities leading to deviations

by more than one standard deviation from the average RI value were observed for K120, T123, Q192, and R196 in the WT DBD, and for Q192 and R196 in the mutant species. The first pair of highly attenuated NHs in the WT variant suggests a contact at cluster 1 (blue; Fig. 2d), close to the C-terminal helix, which is consistent with the designed properties of ADH-6. The attenuation of Q192 and R196 amide peaks points to a contact occurring at cluster 2 in both species (magenta; Fig. 2d).

Besides attenuations, intensity increments were also observed upon ADH-6 addition, suggesting an increase in local mobility at residues of the C-terminal region (E298 and R306) and E271 (cluster 1). However, it was only with mutant R248W p53 DBD that intensity increments were also sampled at L257 and L264 (cluster 3), thereby highlighting a hotspot mutation effect[14,37]. In particular, for the mutant species, the mobility increase associated with ADH-6 contact at cluster 2 (at least) seems to involve the adjacent extremities of strands 3 (L264) and 4 (L257) of sheet B, as well as at the opposite extremity of strand 3 (E271), in addition to some points of the C-terminal region. Conversely, for the WT DBD, the mobility increments accompanying ADH-6 contact affect only the residues of the C-terminal region.

**ADH-6 dissociates intracellular mutant p53 aggregates.** Given ADH-6's potent antagonism of the self-assembly of the aggregation-prone segment of mutant p53 DBD (Fig. 1e, f, h), as well as its multiple interaction sites on the DBD (Fig. 2), we probed the effects of the oligopyridylamide on intracellular mutant p53 aggregates (Fig. 3). MIA PaCa-2 cells harboring aggregation-prone mutant R248W p53 were stained with thioflavin S (ThS), which is commonly used as a marker for intracellular amyloid aggregates, including those of mutant p53 (refs. [36,39]) (Fig. 3a, b). For some experiments, MIA PaCa-2 cells were co-stained with ThS and the anti-p53 antibody PAb 240 (Fig. 3c–f). PAb 240 is specific for partially unfolded p53 as it recognizes an epitope (residues 213–217; Fig. 1a, b) that is buried in folded p53. Since p53 aggregation requires partial unfolding of the protein, PAb 240 is often used as a marker for aggregated p53 (ref. [14]).

Punctate cytosolic ThS staining was observed in MIA PaCa-2 cells, indicating that mutant R248W p53 self-assembles into amyloid aggregates in the cytosol (Fig. 3a, c). Treatment with ADH-6 for 6 h led to a substantial reduction in the number of ThS-positive puncta, with the proportion of cells containing these puncta decreasing to ~30% (Fig. 3a, b). Strong colocalization of the ThS and PAb 240 signals was observed in the vehicle-treated control cells (Fig. 3c), confirming that ThS stains cytosolic mutant p53 aggregates. Treatment of dual-stained MIA PaCa-2 cells with ADH-6 markedly reduced both the ThS and PAb 240 signals (Fig. 3c). The effect of ADH-6 on mutant p53 aggregates was dose dependent, with the PAb 240-positive cells decreasing to ~50 and 24% of controls at 5 and 10 μM ADH-6, respectively (Fig. 3f). Treatment with ReACp53, which has been reported to disaggregate mutant p53 in clinical samples and stable cells established from patients[14], also significantly reduced the proportion of ThS- and PAb 240-positive cells (PAb 240-positive cells were ~40% of controls at 10 μM ReACp53) (Fig. 3e and Supplementary Fig. 7). In contrast, ADH-1, which did not antagonize pR248W aggregation (Fig. 1e, f, h), had a negligible effect on the ThS or PAb 240 staining (Fig. 3d and Supplementary Fig. 7).

As a control, we tested the effects of ADH-1, ReACp53, and ADH-6 on mutant p53 DBD aggregates in plant cells. p53 is not found in plants, which are usually (i.e. in the absence of pathogens) not susceptible to neoplasia[40]. Yet, remarkably, stable transfection of p53 in the model plant Arabidopsis was shown to induce early senescence[41]. Thus, plants represent a viable system for studying mutant p53 DBD aggregation. Here, we amplified the genes using primers corresponding to p53 DBD and cloned them in a binary vector expressing an N-terminal yellow fluorescent protein (YFP) tag, under the control of cauliflower mosaic virus (CaMV35S) promoter. The cloned constructs, YFP-tagged WT and mutant R248W p53 DBDs (YFP:p53DBD$^{WT}$ and YFP:p53DBD$^{R248W}$, respectively), were subsequently expressed in Nicotiana benthamiana using Agrobacterium-mediated transient infiltrations. While the WT DBD accumulated exclusively in the nucleus, expression of R248W p53 DBD resulted in a number of foci spread throughout the cell (Supplementary Fig. 8a), indicating that the mutant protein forms cytosolic aggregates in plant cells similar to those detected in mammalian cells (Fig. 3 and Supplementary Fig. 7). Treatment of the N. benthamiana leaves with ADH-6 or ReACp53 resulted in significantly smaller puncta in the cells expressing the mutant p53 DBD (Supplementary Fig. 8a, c). Our results are in agreement with reports that ReACp53 disaggregates cytosolic mutant p53 puncta[14,42], which leads to accumulation of the released protein in the nucleus[14]. On the other hand, ADH-1 did not reduce the size of mutant R248W DBD puncta (Supplementary Fig. 8a, c).

To corroborate the imaging results, MIA PaCa-2 cells were treated with ADH-1, ReACp53 or ADH-6, lysed and fractionated, and the amount of mutant R248W p53 in the soluble and insoluble fractions was quantified by western blot (Supplementary Fig. 9). As expected, treatment with ADH-1 did not alter the distribution of mutant p53 relative to the vehicle-treated controls, with the aggregation-prone protein strongly detected in the insoluble fraction, but almost completely absent from the soluble fraction. However, treatment with ADH-6 or ReACp53 significantly decreased the mutant p53 content of the insoluble fraction, while markedly increasing the amount of protein in the soluble fraction. These results confirm that ADH-6 converts insoluble cytosolic mutant p53 aggregates into soluble protein.

In order to ascertain whether ADH-6-mediated dissociation of mutant R248W p53 aggregates is a result of direct interaction with the oligopyridylamide, PAb 240-stained MIA PaCa-2 cells were treated with FITC-labeled ADH-6 (ADH-6$_{FITC}$) (Fig. 3g). Initially, strong colocalization of ADH-6$_{FITC}$ and PAb 240 was observed, indicating direct interaction of the oligopyridylamide with mutant p53 aggregates. Eventually, the extent of colocalization decreased as the PAb 240 staining was reduced due to disaggregation of mutant p53. Intracellular target engagement was further confirmed using the cellular thermal shift assay (CETSA), which measures ligand-induced changes in thermal stability of target proteins[43,44]. The labile WT p53 has a melting temperature ($T_m$) of ~45 °C, and mutations destabilize the protein further and lower its $T_m$ by 5–10 °C[6,45]. Consistent with these reports, CETSA generated melting curves in MIA PaCa-2 and SK-BR-3 cells yielded $T_m$ of 39.6 ± 1.5 and 38.8 ± 1.4 °C for p53 mutants R248W and R175H, respectively (Fig. 3h, i and Supplementary Fig. 10). Treatment of the cells with ADH-6, but not ADH-1, significantly increased $T_m$ to 48.2 ± 0.9 and 45.1 ± 1.2 °C for R248W and R175H, respectively (Fig. 3h, i and Supplementary Fig. 10), indicating strong stabilization of aggregation-prone mutant p53 by the oligopyridylamide. Taken together, these results show that, similar to ReACp53 (ref. [14]), ADH-6 efficiently enters cells to directly interact with and stabilize mutant p53, which shifts the folding equilibrium towards the soluble state, leading to dissociation of the protein's inactive amyloid-like cytosolic aggregates.

**ADH-6 causes selective cytotoxicity in cancer cells bearing mutant p53.** Next, we determined the effects of the oligopyridylamides on viability of mutant p53-harboring cancer cells

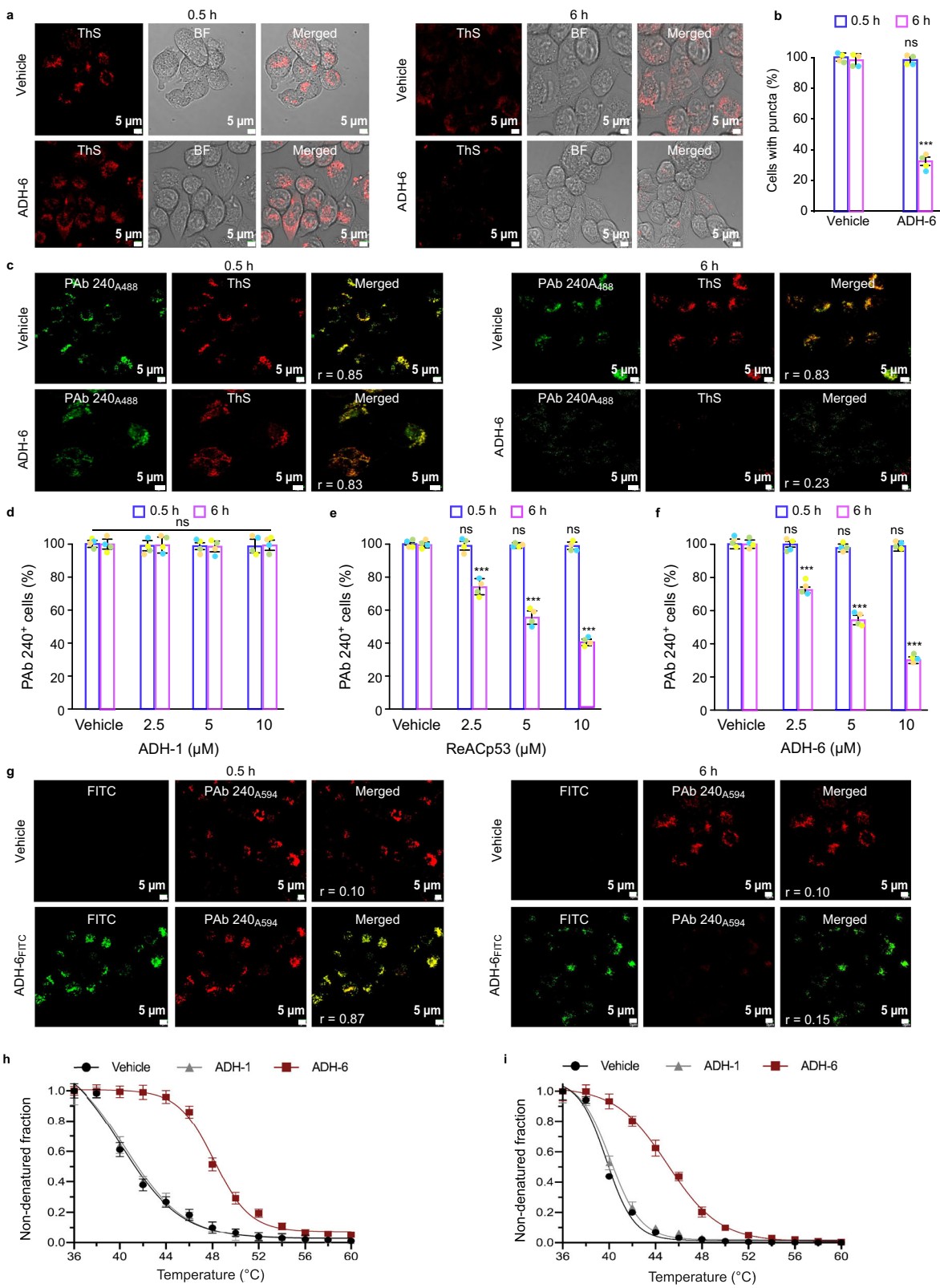

using the MTS assay. A screen of the compounds revealed a strong correlation between their toxicity in mutant R248W p53-bearing MIA PaCa-2 cells (Fig. 4a and Supplementary Fig. 11a, b) and their capacity to antagonize pR248W aggregation (Fig. 1e, f and Supplementary Fig. 1b). Indeed, ADH-6, which was the most potent antagonist of pR248W amyloid formation, was also the most toxic compound to MIA PaCa-2 cells. ADH-6 reduced MIA

PaCa-2 cell viability in a concentration-dependent manner, with an effective concentration ($EC_{50}$) of $2.7 \pm 0.4$ and $2.5 \pm 0.1$ µM at 24 and 48 h incubation times, respectively (Fig. 4a), which was almost half of that for ReACp53 ($EC_{50} = 5.2 \pm 0.5$ and $4.9 \pm 0.3$ µM at 24 and 48 h incubation, respectively) (Supplementary Fig. 11d). On the other hand, ADH-1, which did not inhibit pR248W aggregation, had no adverse effect on viability of MIA

**Fig. 3 ADH-6 dissociates mutant p53 aggregates in cancer cells. a** Confocal fluorescence microscopy images showing thioflavin S (ThS) staining of mutant p53 (R248W) aggregates in MIA PaCa-2 cells treated with vehicle (0.02% DMSO) or ADH-6 (5 μM) for 0.5 or 6 h. Imaging experiments were performed in quadruplicate and representative images are shown. **b** Quantification of ThS-positive MIA PaCa-2 cells after treatment with vehicle or ADH-6. The number of positively stained cells in 3–5 different fields of view are expressed as % of the total number of cells ($n = 4$ biologically independent samples). Data presented are mean ± SD. Statistical analysis was performed using two-tailed unpaired $t$-test. $P < 0.0001$ for ADH-6 vs vehicle at 6 h. **c** Confocal fluorescence microscopy images of ThS and PAb 240 antibody staining of R248W aggregates in MIA PaCa-2 cells treated with vehicle or 5 μM ADH-6 for 0.5 or 6 h. Images shown are representative of four independent experiments. **d–f** Quantification of PAb 240-positive MIA PaCa-2 cells after treatment with the indicated concentrations of ADH-1, ReACp53, or ADH-6 for 0.5 or 6 h relative to controls (vehicle-treated cells). The number of positively stained cells in 3–5 different fields of view are expressed as % of the total number of cells (mean ± SD; $n = 4$). Statistical analysis was performed using repeated measures two-way ANOVA followed by Holm-Sidak's post hoc test. $P < 0.0001$ for ReACp53 (2.5–10 μM) vs vehicle at 6 h (**e**); $P < 0.0001$ for ADH-6 (2.5–10 μM) vs vehicle at 6 h (**f**). **g** Colocalization of FITC-labeled ADH-6 (ADH-6$_{FITC}$) with PAb 240-stained R248W aggregates following incubation with the oligopyridylamide (5 μM) for 0.5 or 6 h. Colocalization was quantified using directional Pearson correlation coefficient, $r$, which measures pixel-by-pixel covariance in the signal level of two images[132]. Scale bar = 5 μm. **h, i** Cellular thermal shift assay (CETSA) analysis of intracellular target engagement. Melting curves for p53 mutants R248W (**h**) and R175H (**i**) in MIA PaCa-2 and SK-BR-3 cells, respectively, in the absence or presence of the oligopyridylamides (mean ± SD; $n = 3$). ***$P < 0.001$ or non-significant (n.s., $P > 0.05$) for comparisons with vehicle-treated controls.

PaCa-2 cells (Fig. 4a). Importantly, the oligopyridylamides, including ADH-6, were completely nontoxic to WT p53-bearing breast cancer MCF-7 cells (Fig. 4b and Supplementary Fig. 11c). Moreover, treatment of a range of human cancer cells bearing WT p53 (AGS, A549, CAKI-1, CESS, LS 174T, MDA-MB-175-VII, NCI-H1882, SH-SY5Y, and U-2 OS) with ADH-6 yielded no significant toxicity (Supplementary Fig. 11e). Conversely, the oligopyridylamide was highly toxic to mutant R175H p53-bearing SK-BR-3 cells (EC$_{50}$ = 2.6 ± 0.1 and 2.5 ± 0.2 μM at 24 and 48 h incubation, respectively) (Fig. 4c). Likewise, we observed a substantial (~75–95%) decrease in viability of human cancer cells harboring other aggregation-prone p53 mutants (R248W: COLO 320DM and NCI-H1770; R248Q: HCC70 and OVCAR-3; R175H: LS123; R273H: HT-29 and ARH-77; Y220C: NCI-H748 and NCI-H2342; and R280K: MDA-MB-231) following treatment with ADH-6 (Supplementary Fig. 11f).

Notably, ADH-6 did not adversely affect viability of p53 null human bone cancer Saos-2 cells (Fig. 4d). However, transfecting the cells with mutant R248W or R175H p53 rendered them highly susceptible to ADH-6 mediated cytotoxicity (EC$_{50}$ = 2.0 ± 0.2 and 2.3 ± 0.2 μM for R248W and R175H, respectively, at 48 h incubation) (Fig. 4e, f and Supplementary Fig. 12a). Similarly, p53 null human ovarian cancer SKOV-3 cells were unaffected by treatment with ADH-6, but following transfection with R248W or R175H, the oligopyridylamide induced significant toxicity in the cells (Supplementary Fig. 12a–d). Together with the screen of cancer cells bearing WT and mutant p53, these results indicate that the observed cytotoxicity of ADH-6 in cancer cells is directly related to the oligopyridylamide's capacity to antagonize mutant p53 amyloid formation.

A marker of restored p53 function is activation of apoptosis. Therefore, following treatment with vehicle, or 5 μM ADH-1, ReACp53, or ADH-6, MIA PaCa-2 cells were stained with Alexa 488-conjugated annexin V/propidium iodide (PI) and quantified by flow cytometry, a common method for detecting apoptotic cells[46] (Fig. 4g, h and Supplementary Fig. 13a). As expected, treatment with ADH-1 had a negligible effect. Exposure to ReACp53 resulted in 75.7 ± 1.1% and 2.0 ± 0.2% of cells undergoing early and late apoptosis, respectively, which is in line with the reported capacity of the peptide to induce apoptosis in aggregation-prone mutant p53-bearing cancer cells[14]. However, treatment with ADH-6 led to an even more pronounced effect, with 11.4 ± 0.9% and 77.9 ± 1.3% of cells undergoing early and late apoptosis, respectively. The rescue of p53 activity was further corroborated by cell cycle analysis (Fig. 4i and Supplementary Figs. 11g and 13b). In contrast to ADH-1, treatment with ReACp53 resulted in a small but significant shift in the cell cycle distribution of the asynchronous population, which is a hallmark

of p53 activation[14]. However, yet again we observed an even greater effect following exposure to ADH-6, with a larger shift in the cell cycle distribution occurring (i.e. more cells in the G0/G1 phase and fewer in G2/M) relative to ReACp53. Collectively, our results strongly suggest that ADH-6-mediated cytotoxicity in cancer cells is due to abrogation of mutant p53 aggregation by the oligopyridylamide, which leads to recovery of WT p53 function.

**ADH-6 induces transcriptional reactivation of p53.** To confirm that ADH-6 rescues p53 function, we first used ChIP-qPCR to detect recruitment of mutant R248W p53 to the WT protein's binding sites on promoters/enhancers of target genes in oligopyridylamide-treated MIA PaCa-2 cells (primers used are listed in Supplementary Table 1). Treatment with ADH-6 led to binding of R248W to *Cdkn1a* (also known as *P21*) (Supplementary Fig. 14a). Importantly, we did not detect significant binding of the p53 homologs, p63 and p73, to *Cdkn1a* under the same experimental conditions (Supplementary Fig. 14a), supporting the notion that ADH-6 activity is mutant p53 dependent. Interaction of R248W with other well-established primary p53 transcriptional targets, *PIG3* and *NOXA*, was also observed in ADH-6 treated MIA PaCa-2 cells (Supplementary Fig. 14b, c). (Please note that functions of the genes referenced in this section are described in Supplementary Section 4.)

In agreement with the ChIP-qPCR results, western blot analysis revealed elevated expression of both p21 and Noxa in ADH-6-treated MIA PaCa-2 cells (Supplementary Fig. 15). Interestingly, we also observed significantly higher expression of p53-inducible MDM2 and proapoptotic Bax in response to ADH-6 treatment (Supplementary Fig. 15). Surprisingly, recruitment to the *MDM2* and *BAX* genes was not observed by ChIP-qPCR, which may be a consequence of the interaction of mutant p53 with the WT protein's binding sites on the promoters of these genes being too transient or weak to be detected by the assay[47,48]. Of relevance, the mutant p53 disaggregating peptide ReACp53 was also reported to upregulate p21, Noxa, MDM2, and Bax in ovarian cancer cells that harbor another aggregating p53 mutant, R248Q[14]. Taken together, these results indicate that ADH-6 specifically targets and reactivates aggregation-prone mutant p53.

Next, transcriptome analysis was performed to assess the effects of ADH-6 on the mutant p53-bearing cells at the global gene level (Fig. 5). Total RNA samples were isolated from ADH-6, ADH-1, and vehicle (control, C) treated cells, which was followed by RNA-Seq library preparation. To establish the best condition for differential gene expression analysis, we applied correlation analysis to the data (Supplementary Fig. 16a). As assessed by principal component analysis (PCA), there were variable clustering patterns within and between samples, showing

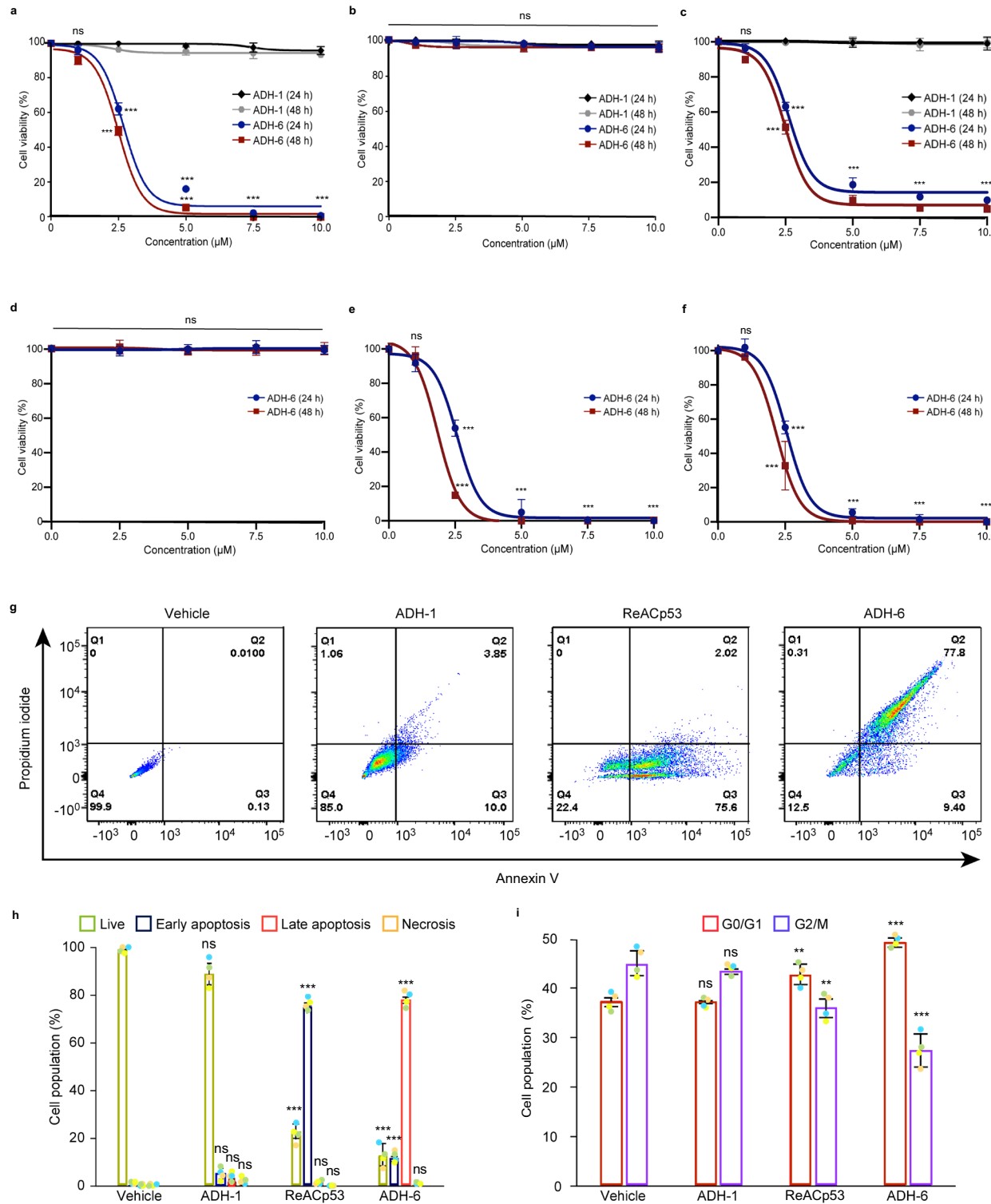

the highest effect to be based on the treatment at PC1: 36% (Supplementary Fig. 16a). ADH-6 treated cell replicates were observed to cluster independently from ADH-1 and C sample replicates. Subsequently, the number of differentially expressed genes (DEGs) was determined based on statistical cut-offs, including $P$-adj < 0.05, $P$-adj < 0.01 and $P$-adj < 0.001 between pairwise comparisons of ADH-6/C, ADH-1/C, and ADH-6/ADH-1. The number of DEGs varied, with the highest observed for ADH-6/C at 485 genes at $P$-adj < 0.05 (Supplementary Fig. 16b).

In order to further probe the effects of ADH-6 treatment, the expression patterns of the 485 DEGs identified were analyzed using hierarchical clustering (performed by the JMP Genomics software) following variance-stabilizing transformation (VST) of the count data. As shown in the heatmap, distinctive clustering patterns were observed, with 196 DEGs identified as upregulated, while 289 were downregulated, in the ADH-6 treatment group relative to the vehicle-treated controls (ADH-6/C) (Fig. 5a). To characterize the functions of these DEGs, Gene Ontology (GO) analysis was performed, focusing specifically on the "biological

**Fig. 4 ADH-6 causes death of cancer cells bearing mutant, but not WT, p53. a–c** Effects of ADH-6 on viability of cancer cells bearing WT or mutant p53. MIA PaCa-2 (mutant R248W p53) (**a**), MCF-7 (WT p53) (**b**), and SK-BR-3 (mutant R175H p53) (**c**), cells treated with increasing oligopyridylamide concentrations for 24 or 48 h. (**d–f**) p53 null Saos-2 cells before (**d**) and after transfection with p53 mutants, R248W (**e**) or R175H (**f**), treated with increasing concentrations of ADH-6 for 24 or 48 h. Cell viability in **a–f** was assessed using the MTS assay, with the % viability determined form the ratio of the absorbance of the treated cells to the control cells ($n = 3$ biologically independent samples). Data presented are mean ± SD. Statistical analysis in **a–f** was performed using two-tailed unpaired $t$-test. $P < 0.0001$ for ADH-6 vs ADH-1 at the same compound concentration (2.5–10 μM) and incubation time (24 or 48 h) (**a**, **c**); $P < 0.0001$ for ADH-6 treatment of Saos-2/R248W compared with untransfected cells (data shown in **d**) at the same compound concentration (2.5–10 μM) and incubation time (24 or 48 h) (**e**); $P < 0.0001$ for ADH-6 treatment of Saos-2/R175H compared with untransfected cells (data shown in **d**) at the same compound concentration (2.5–10 μM) and incubation time (24 or 48 h) (**f**). **g**, **h** Flow cytometry analysis of annexin V/ propidium iodide (PI) staining of MIA PaCa-2 cells that were treated with vehicle (control), or 5 μM ADH-1, ReACp53, or ADH-6, for 24 h. The bottom left quadrant (annexin V−/PI−) represents live cells; bottom right (annexin V+/PI−), early apoptotic cells; top right (annexin V+/PI+), late apoptotic cells; and top left (annexin V−/PI+), necrotic cells (**g**). A summary of the incidence of early/late apoptosis and necrosis in the different treatment groups determined from the flow cytometry analysis of annexin V/PI staining (mean ± SD; $n = 4$) (**h**). Statistical analysis in **h** was performed using one-way ANOVA followed by Dunnett's post hoc test. $P < 0.0001$ for ReACp53 vs vehicle (live and early apoptosis); $P < 0.0001$ for ADH-6 vs vehicle (live, early apoptosis, and late apoptosis). **i** Cell cycle distribution of MIA PaCa-2 cells treated with vehicle (control), or 5 μM ADH-1, ReACp53 or ADH-6, for 6 h as determined by measurement of DNA content using flow cytometry (mean ± SD; $n = 4$). Two-tailed unpaired $t$-test: $P = 0.0071$ and 0.0037 for ReACp53 vs vehicle (G0/G1 and G2/M, respectively); $P < 0.0001$ and $P = 0.0004$ for ADH-6 vs vehicle (G0/G1 and G2/M, respectively). ***P < 0.01, ***P < 0.001** or non-significant (n.s., $P > 0.05$) for comparisons with vehicle-treated controls.

process" category (Fig. 5b). With a $P$-value <0.05 (normalized to −log10) cut-off, several biological process enrichments were identified, the top five of which were related to regulation of cell cycle (GO:0051726), cell cycle arrest (GO:0007050), cell proliferation (GO:0008283), regulation of apoptosis (GO:0042981) and aging (GO:0007568) (Fig. 5b). A Venn diagram was then used to delineate overlap of DEGs between the selected processes in order to remove redundancies (Fig. 5c). Based on the Venn diagram, the selected enrichments were further refined into a finalized DEGs list, as displayed in the heatmap (scaled to log2 count per million reads (log2cpm_voom)) generated using DESeq2 (Fig. 5d). From the heatmap, 25 DEGs were identified as upregulated, whereas 49 were downregulated, in ADH-6/C.

The observed expression patterns strongly support transcriptional activation of p53 by ADH-6. In agreement with the ChIP-qPCR analysis, treatment of MIA PaCa-2 cells with ADH-6 resulted in significant (0.7–1.3-fold) upregulation of *Cdkn1a* relative to both the ADH-1 and control groups (Fig. 5e). ADH-6 treatment also significantly upregulated other p53 target genes that are important mediators of cell cycle arrest and apoptosis[49–51] (Fig. 5e). For instance, a 0.4-fold upregulation of the tumor protein p53-inducible nuclear protein 1 (*Tp53inp1*) was observed in the ADH-6 treatment group compared to controls. Likewise, ADH-6 induced a 0.6–1-fold upregulation of *FOS* relative to treatment with vehicle or ADH-1. Finally, treatment with ADH-6 resulted in a 0.9–1.3-fold upregulation of *EGR1* compared to both the vehicle and ADH-1 treatment groups. On the other hand, exposure to ADH-6 led to significant downregulation of p53 targets *TP73* and *SIX1* (Fig. 5e). ADH-6 induced a 1-fold downregulation of *TP73* compared to controls. Interestingly, downregulation of p73 was observed following treatment of mutant p53-bearing cells with ReACp53 (ref. [14]). ADH-6 treatment also resulted in a 0.3–0.5-fold downregulation of the *SIX1* oncogene relative to both controls and ADH-1 treatment groups.

Lastly, we performed ingenuity pathway analysis (IPA) of the DEGs to identify the upstream transcriptional regulators (TRs) of these genes. Over a dozen TRs were activated in the ADH-6 group, and the 10 candidates shown were identified on the basis of their $q$-value and $z$-score thresholds (Supplementary Fig. 17a, b). Of these TRs, *TP53* showed robust activation as indicated by the highest $q$-value in both the ADH-6 vs vehicle (Supplementary Fig. 17a) and ADH-6 vs ADH-1 (Supplementary Fig. 17b) comparisons. Forty-two genes in the ADH-6 vs control comparison and 44 in the ADH-6 vs ADH-1 comparison (Supplementary Fig. 17c) showed

an expression pattern that signified the activation of the *TP53* pathway. To further determine the biological consequence of ADH-6-mediated p53 activation, gene set enrichment analysis (GSEA) was performed (Supplementary Fig. 17d). The findings of IPA were corroborated using this approach as the *TP53* pathway was shown to be induced in the ADH-6 treatment group when compared to the ADH-1 group. Activation of the *TP53* pathway was strongly supported by the observed suppression of the *MYC* pathway, which is known to be negatively regulated by p53 (ref. [52]). Furthermore, the gene set for cell death/apoptosis was activated and the genes involved in cell cycle progression (G2/M) were suppressed in the ADH-6 treatment group. Collectively, these findings strongly point to ADH-6 reactivating the p53 transcriptional response in aggregation-prone mutant p53-bearing cancer cells.

**ADH-6 downregulates cancer-promoting phosphoproteins.** We subsequently carried out proteome analysis of oligopyridylamide-treated MIA PaCa-2 cells. We chose to focus specifically on the phosphoproteome (Supplementary Fig. 18) as many phosphoproteins are involved in regulating major pathways implicated in cancer[53–57]. Unsupervised hierarchical clustering (UHC) revealed two distinct expression profiles pertaining to the ADH-1 and the ReACp53/ADH-6 treatment groups (Fig. 6a). PCA analysis further highlighted that the main source of variation in the two groups was the peptide/compound treatment (Fig. 6a, b). Moreover, compared to ADH-1, most of the phosphoproteins downregulated upon ReACp53 treatment were also downregulated in the ADH-6 samples (Fig. 6c). Next, we carried out GSEA of hallmark signatures for the differentially expressed phosphoproteins. We observed a comparable reduction of phosphoprotein expression, in a number of pathways, in the ReACp53 and ADH-6 treated cells compared to the ADH-1 treatment group (Fig. 6d).

The biological roles of the downregulated phosphoproteins were inferred from published data (Supplementary Table 2), and revealed that these proteins can be divided into two major functional groups that positively regulate either DNA replication/repair or cell cycle progression/proliferation, although many downregulated proteins in the ADH-6 treatment group had overlapping gene signatures (Fig. 6e). Importantly, these two major groups, which play a critical role in cancer initiation and progression, are known to be directly regulated by p53 (refs. [58–62]). This strongly supports that both the ReACp53 and ADH-6 treatments result in reactivation of p53, which

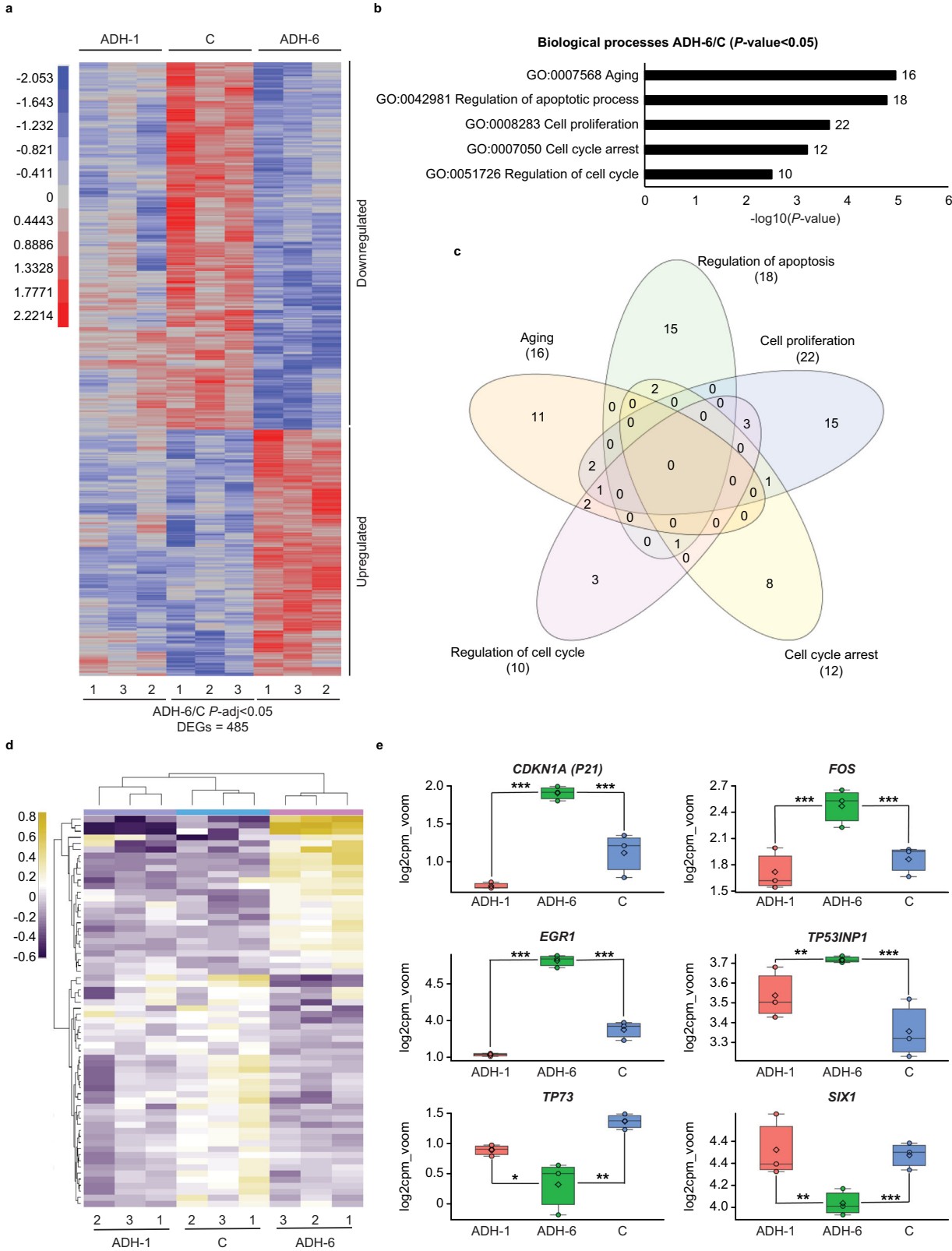

downregulates key cancer-promoting phosphoproteins (Supplementary Table 2a). Indeed, numerous studies have reported that p53 directly downregulates the expression of key positive regulators of cancer in the p53 pathway, such as CD44 (ref. [63]) and PCNA[64,65], as well as the oncoproteins c-Myc[52,66], E2F[67,68], and mTORC1[69–71] (Supplementary Fig. 19). Interestingly, only a few upregulated phosphoproteins were identified in our phosphoproteome screen (Fig. 6f and Supplementary Table 2b). Together, the transcriptomic and proteomic analyses clearly demonstrate that ADH-6-mediated dissociation of mutant p53 amyloid-like aggregates in cancer cells restores p53 function, leading to cell cycle arrest and activation of apoptosis (Fig. 4).

**Fig. 5 Transcriptome analysis of oligopyridylamide-treated MIA PaCa-2 cells. a** Variance-stabilized count data heatmap showing clustering patterns of differentially expressed genes (DEGs) identified based on statistical significance of P-adj < 0.05 from the ADH-6 treatment group relative to vehicle-treated controls (C) (denoted ADH-6/C). Significance was assessed by false discovery rate (FDR) adjusted P-value (P-adj or q-value), which was obtained from the hypergeometric P-value that was corrected for multiple hypothesis testing using the Benjamin and Hochberg procedure[122]. The ADH-1 treatment group is included for comparison. The adjacent legend indicates the scale of expression, with red signifying upregulation, and blue downregulation, in ADH-6/C. **b** Gene ontology (GO) analysis of ADH-6/C showing the top five "biological process" term enrichments based on a cut-off of P-value <0.05 (normalized to −log10). Statistical analysis was performed using two-tailed unpaired t-test. **c** Venn diagram delineating overlap of DEGs between the processes in **b**. **d** Heatmap (scaled to log2 counts per million reads (log2cpm_voom)) displaying a finalized list of DEGs based on the Venn diagram. **e** Gene expression boxplots of DEGs selected from the heatmap in **d** (n = 3 biologically independent samples). The boxplots display the five-number summary of median (center line), lower and upper quartiles (box limits), and minimum and maximum values (whiskers). Statistical analysis was performed using two-tailed unpaired t-test. P < 0.0001 for Cdkn1a, FOS, EGR1, and SIX1, P = 0.0002 for Tp53inp1, and P = 0.0026 for TP73 (ADH-6/C); P < 0.0001 for Cdkn1a, FOS, and EGR1, P = 0.0072 for Tp53inp1, P = 0.0471 for TP73, and P = 0.0019 for SIX1 (ADH-6/ADH-1). *P < 0.05, **P < 0.01, ***P < 0.001 or non-significant (n.s., P > 0.05) for comparisons with vehicle-treated controls and between the different treatment groups.

**In vivo administration of ADH-6 causes regression of mutant p53-bearing tumors.** Having established that ADH-6 potently abrogates mutant p53 amyloid formation and restores WT-like tumor suppressor function in vitro, we next assessed the in vivo efficacy of the oligopyridylamide (Fig. 7 and Supplementary Figs. 20 and 21). Following intraperitoneal (IP) injection, ADH-6 quickly entered circulation, with the peak concentration in serum mice (~21 μg/mL) occurring at 2 h post-injection (Fig. 7a). The in vivo circulation half-life[72] of ADH-6 ($T_{1/2}$ ~3.6 h) was much longer than that of ReACp53 ($T_{1/2}$ ~1.5 h)[14], or other chemotherapeutics of a comparable size, such as doxorubicin ($T_{1/2}$ < 30 min)[73] or paclitaxel ($T_{1/2}$ ~1.7 h)[74]. Moreover, ADH-6 was detected in the plasma up to 48 h after administration, whereas ReACp53 was largely eliminated from the bloodstream in 24 h[14]. The relatively long in vivo circulation time of ADH-6 should facilitate greater accumulation in tumor tissue. Indeed, the amount of ADH-6 in the MIA PaCa-2 xenografts increased continuously over 48 h post-treatment (Fig. 7b). The high in vivo stability suggested that ADH-6 would exhibit potent antitumor activity.

To test this hypothesis, mice bearing MIA PaCa-2 xenografts were treated with ADH-6, ADH-1 or ReACp53 (155.6 μM in PBS) (Supplementary Fig. 20a, b). Treatment consisted of IP injections every 2 days, for a total of 12 doses (Supplementary Fig. 20b). IP injection was used since this is the preferred administration route for the control ReACp53 peptide[14]. As expected, ADH-1 did not affect tumor growth (Supplementary Fig. 20d–f). Conversely, both ADH-6 and ReACp53 reduced tumor growth relative to the saline-treated control group. However, of the two treatments, ADH-6 exhibited significantly greater antitumor efficacy (Supplementary Fig. 20d–f). The effects of the oligopyridylamides were recapitulated in mice bearing SK-BR-3 xenografts, with ADH-6 again reducing tumor growth substantially compared to ADH-1 (Supplementary Fig. 21).

To confirm its specificity for aggregation-prone mutant p53 in vivo, we tested ADH-6 in a dual xenograft model: mice bearing MIA PaCa-2 (mutant R248W p53) xenografts on one flank and MCF-7 (WT p53) xenografts on the other flank as an internal control (Fig. 7c). For this model, treatment consisted of IP injections every 2 days, for a total of 12 doses, of ADH-6 or ReACp53 (716.4 μM in 0.02% DMSO) or vehicle (0.02% DMSO) (Fig. 7d). DMSO was introduced to increase the solubility of ADH-6 and allow administration of a higher dose of the oligopyridylamide and ReACp53, which was comparable to that used in the study by Eisenberg et al.[14]. While treatment with ADH-6 or ReACp53 did not have a significant effect on growth of the MCF-7 xenografts, both the protein mimetic and peptide markedly reduced the MIA PaCa-2 tumor growth relative to the vehicle-treated control group (Fig. 7f–i). Following treatment with ReACp53, the tumors increased from an initial volume of 25 ± 2 to

29 ± 1 mm³ (compared to an increase from 25 ± 1 to 44 ± 2 mm³ in the vehicle-treated controls; Fig. 7f), and the tumor mass was ~30% that of the controls (Fig. 7h, i). However, ADH-6 again exhibited significantly greater antitumor efficacy than ReACp53. Treatment with the oligopyridylamide reduced the MIA PaCa-2 xenografts from an initial volume of 26 ± 2 to 17 ± 2 mm³ (Fig. 7f), and decreased the tumor mass to ~3% that of the controls (Fig. 7h, i). Moreover, ADH-6 prolonged survival appreciably compared to ReACp53 (Fig. 7n and Supplementary Fig. 20g).

Histological analysis of tumor tissues using hematoxylin and eosin (H&E) staining confirmed the higher potency of ADH-6 compared to ReACp53 against MIA PaCa-2 xenografts (Fig. 7j and Supplementary Fig. 20h). Immunohistochemistry (IHC) analysis of residual tumor sections revealed that treatment with ADH-6 strongly reduced mutant p53 levels in the MIA PaCa-2 xenografts, but did not significantly alter the amount of WT p53 in MCF-7 xenografts (Fig. 7k–m). Thus, ADH-6 effectively targets aggregation-prone and inactive mutant, but not functional WT, p53 in vivo.

ADH-6 and ReACp53 did not adversely affect the body weight of treated mice (Fig. 7e and Supplementary Figs. 20c and 21c), and H&E-stained heart, lung, liver, kidney, and spleen sections showed no apparent abnormalities or lesions (Supplementary Figs. 22 and 23). Our observations are in agreement with the reported high tolerability and lack of measurable toxicity in healthy tissue in vivo of ReACp53 (ref. [14]). Taken together, these results demonstrate that ADH-6 potently shrinks xenografts harboring aggregation-prone mutant p53 in vivo, while exhibiting no toxicity to healthy tissue, leading to markedly prolonged survival.

## Discussion

A wide range of disorders are associated with misfolding and self-assembly of functional proteins or peptides into amyloids, which are aggregates that are characterized by a fibrillar morphology, a predominantly β-strand secondary structure, high thermodynamic stability, insolubility in common solvents, and resistance to proteolytic digestion[75–77]. These amyloid diseases include AD, T2D, Parkinson's disease, transmissible spongiform encephalopathies (or prion diseases), and Huntington's disease[76,77]. Rather unexpectedly, it is now evident that a subset of mutant p53-associated cancers can also be classed as amyloid diseases. The conformationally flexible p53 normally exists in an equilibrium between native/folded, partially unfolded and aggregated states[14,78]. At the core of p53 is the thermodynamically unstable DBD, which houses the vast majority of cancer-associated mutations[9,10]. Many of these mutations destabilize the DBD further and prompt its unfolding, leading to exposure of the normally hidden aggregation-nucleating subdomain, p53$_{251–258}$ (refs. [13,14]). This shifts the equilibrium towards the aggregated

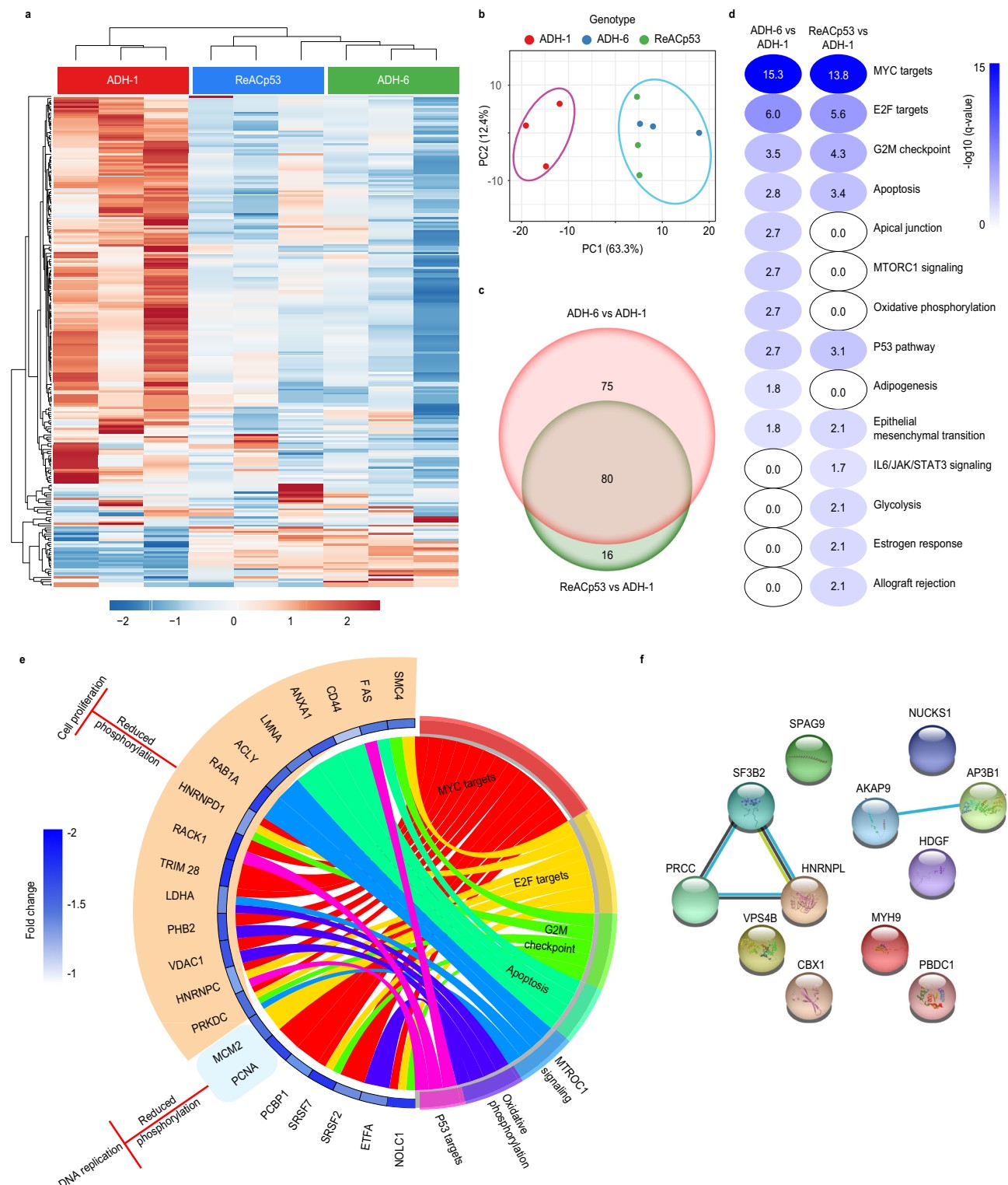

state[13,14], culminating in sequestration of mutant p53 in inactive cytosolic amyloid-like aggregates that often co-opt the WT isoform and its paralogs, p63 and p73 (ref. [17]).

The amyloidogenic nature of mutant p53 aggregation was first reported almost two decades ago[79]. Subsequent studies revealed amyloid-like mutant p53 aggregates in cancer cell lines, in vivo animal models and human tumor biopsies[11,12]. Increasing evidence now implicates these aggregates in mutant p53-associated loss of tumor suppressor function and oncogenic GoF[3,5,12,14,17,19].

Intriguingly, there are reports of transmission of this oncogenic GoF phenotype, whereby mutant p53 amyloid aggregates (in the form of oligomers or fibril fragments) are trafficked to contiguous cells and seed aggregation of endogenous p53, suggesting that aggregation-prone mutant p53-bearing cancers share a common mechanism of propagation with prion diseases[11,12]. It is therefore rather surprising that, so far, there are only a handful of reports of therapeutic strategies targeting mutant p53 aggregation. The most notable of these is ReACp53, a sequence-specific peptide inhibitor

**Fig. 6 Quantitative proteomic analysis of phosphoprotein expression in oligopyridylamide-treated MIA PaCa-2 cells. a** Heatmap showing differentially expressed phosphoproteins in ADH-1, ReACp53, and ADH-6 treated samples. Unsupervised hierarchical clustering reveals distinct segregation of ADH-1 vs ReACp53 and ADH-6. **b** Results from principal component analysis (PCA) showing that the main source of variation in the indicated groups was the compound/peptide treatment. **c** Venn diagram showing phosphoproteins that were differentially expressed in the ADH-6 vs ADH-1 and ReACp53 vs ADH-1 comparisons. The number of phosphoproteins that were unique or common in both comparisons are highlighted. **d** Gene set enrichment overlap analysis (GSEA) of hallmark signatures for the differentially expressed and downregulated phosphoproteins. Significance was assessed by $q$-value, or false discovery rate (FDR)-adjusted $P$-value, which was obtained from the hypergeometric $P$-value that was corrected for multiple hypothesis testing using the Benjamin and Hochberg procedure[122]. The hallmark signatures were ranked based on the $-\log10(q$-value) of overlap. Only gene signatures that were significantly different ($q$-value <0.05 or $-\log10(q$-value) >1.3) were further analyzed. **e** GOChord plot of phosphoproteins that belong to top dysregulated hallmark signatures (shown in **d**). The plot also depicts overlap of phosphoproteins between indicated gene signatures. Fold change of phosphoproteins (ADH-6 vs ADH-1) is represented by the blue track with the color spectrum depicting the level of reduction in phosphoprotein expression in the ADH-6 group. The biological roles of the downregulated phosphoproteins in ADH-6 were inferred from published data. The outer layer of the GOChord plot links the downregulation of phosphoproteins with the inhibition of DNA repair/replication and cell proliferation on the basis of previous reports (Supplementary Table 2a). **f** Protein–protein interaction (PPI) network map of upregulated phosphoproteins in ADH-6 vs ADH-1 treatments (biological roles of upregulated phosphoproteins are summarized in Supplementary Table 2b).

that blocks mutant p53 aggregation by masking the DBD's aggregation-nucleating p53$_{251-258}$ segment[14,17]. This prevents further aggregation and shifts the folding equilibrium toward p53's functional, WT-like state. Treatment with ReACp53 rescued p53 function in aggregation-prone mutant p53-bearing human ovarian and prostate cancer cells, leading to inhibition of cell proliferation in vitro and tumor shrinkage in vivo[14,42]. More recently, a bifunctional small molecule (denoted L$^I$), with a structure based on the amyloid reporter ThT, was reported to restore zinc binding in mutant p53 and to modulate its aggregation[80]. L$^I$ inhibited mutant p53 aggregation and restored the protein's transcriptional activity, leading to apoptosis in human gastric cancer cells in vitro. While promising, the in vivo efficacy of L$^I$ is yet to be determined.

Here, we have extended our protein mimetic-based approach, previously developed to modulate various aberrant PPIs[20,21,23], towards mutant p53 aggregation. Screening a focused library of oligopyridylamide-based α-helix mimetics, we identified a cationic tripyridylamide, ADH-6, that potently inhibited oligomerization and amyloid formation of pR248W, a mutant p53 DBD-derived peptide containing both the aggregation-nucleating sequence and the commonly occurring R248W mutation, by stabilizing the peptide's native conformation (Fig. 1). It should be noted that ADH-6 only modestly antagonized Aβ amyloid formation[20], indicating a degree of specificity of the oligopyridylamide for mutant p53. Importantly, ADH-6 also dissociated pre-formed pR248W aggregates, and prevented further aggregation of the peptide (Fig. 1). Subsequent studies in human pancreatic carcinoma MIA PaCa-2 cells harboring mutant R248W p53 (Fig. 3 and Supplementary Figs. 7 and 9), and *N. benthamiana* cells transfected with YFP-tagged WT and R248W p53 DBDs (Supplementary Fig. 8), showed that ADH-6 effectively dissociates intracellular mutant p53 amyloid-like aggregates. Evidence that these effects are due to direct intracellular target engagement comes from the following experiments: (i) confocal microscopy, where extensive colocalization of fluorescently labeled ADH-6 with antibody-stained intracellular mutant p53 was observed; and (ii) CETSA, which demonstrated strong stabilization of p53 mutants in cancer cells by ADH-6 (Fig. 3 and Supplementary Fig. 10).

NMR spectroscopy was used to gain insights into the interactions between ADH-6 and mutant R248W p53 DBD at the molecular level. The experiments revealed that the α-helix mimetic interacts with not only the aggregation-nucleating subdomain, but also several other regions of mutant p53 DBD (Fig. 2). This is not surprising given that the DBD contains several helical regions (Fig. 1), as well as non-helical regions in which amino acids—from neighboring or distal parts of the

protein—nevertheless align to present target side-chains with a topology that matches that of the surface functionalities of ADH-6 (Fig. 2). On the other hand, ReACp53, by virtue of its sequence, specifically targets the aggregation-nucleating subdomain (p53$_{251-258}$) of the DBD[14]. However, it has been shown that mutant p53 unfolding may expose other aggregation-prone sites, besides p53$_{251-258}$, which can contribute to the protein's self-assembly and enable it to partially circumvent the inhibitory effects of ReACp53 (ref. [81]). Thus, the targeting of multiple regions of mutant p53 DBD enables ADH-6 to antagonize intracellular mutant p53 aggregation more effectively than ReACp53.

Helical intermediates play a role in the amyloid assembly of not only Aβ and IAPP, but also a number of other disease-associated intrinsically disordered proteins, including α-synuclein (Parkinson's disease), prion protein (PrP, prion diseases) and tau (tauopathies)[82–84]. A distinct possibility is that the intrinsically disordered mutant p53 aggregates via a similar pathway, whereby destabilized α-helical intermediates transition to stable, β-sheet rich amyloid aggregates. Indeed, there are reports that aggregation-prone p53 mutants, and fragments of these proteins, sample helical structures during the process of self-assembling into amyloid fibers[85,86]. This raises the possibility that, in addition to the monomeric protein, ADH-6 also targets pre-amyloid helical intermediates of mutant p53's amyloid aggregation pathway. By stabilizing mutant p53 in its monomeric or helical intermediate states, ADH-6 is able to effectively prevent the protein's amyloid aggregation, thereby strongly shifting the folding equilibrium towards the functional, WT-like state.

Treatment with ADH-6 resulted in substantial toxicity in mutant R248W p53-bearing MIA PaCa-2 cells (Fig. 4). Several lines of evidence from multiple experiments establish that ADH-6-mediated cytotoxicity is directly related to the oligopyridylamide's capacity to antagonize mutant p53 amyloid formation: (i) a second screen, testing the effects of the oligopyridylamides on viability of MIA PaCa-2 cells (Fig. 4 and Supplementary Fig. 11), revealed a strong correlation between the compounds' cytotoxicity and their capacity to antagonize pR248W aggregation (Fig. 1 and Supplementary Fig. 1), and again identified ADH-6 as the most potent compound in the library; (ii) ADH-6 is significantly more toxic to MIA PaCa-2 cells than the control ReACp53 peptide (Fig. 4 and Supplementary Fig. 11), which reflects their relative capacities to abrogate intracellular mutant p53 aggregation (Fig. 3 and Supplementary Figs. 7–9); (iii) ADH-6 induced substantial loss of viability in a range of human cancer cells bearing aggregation-prone mutant p53, but was completely nontoxic to WT p53 harboring human cancer cells (Fig. 4 and Supplementary Fig. 11); and (iv) p53 null Saos-2 and SKOV-3 cells, which were insensitive

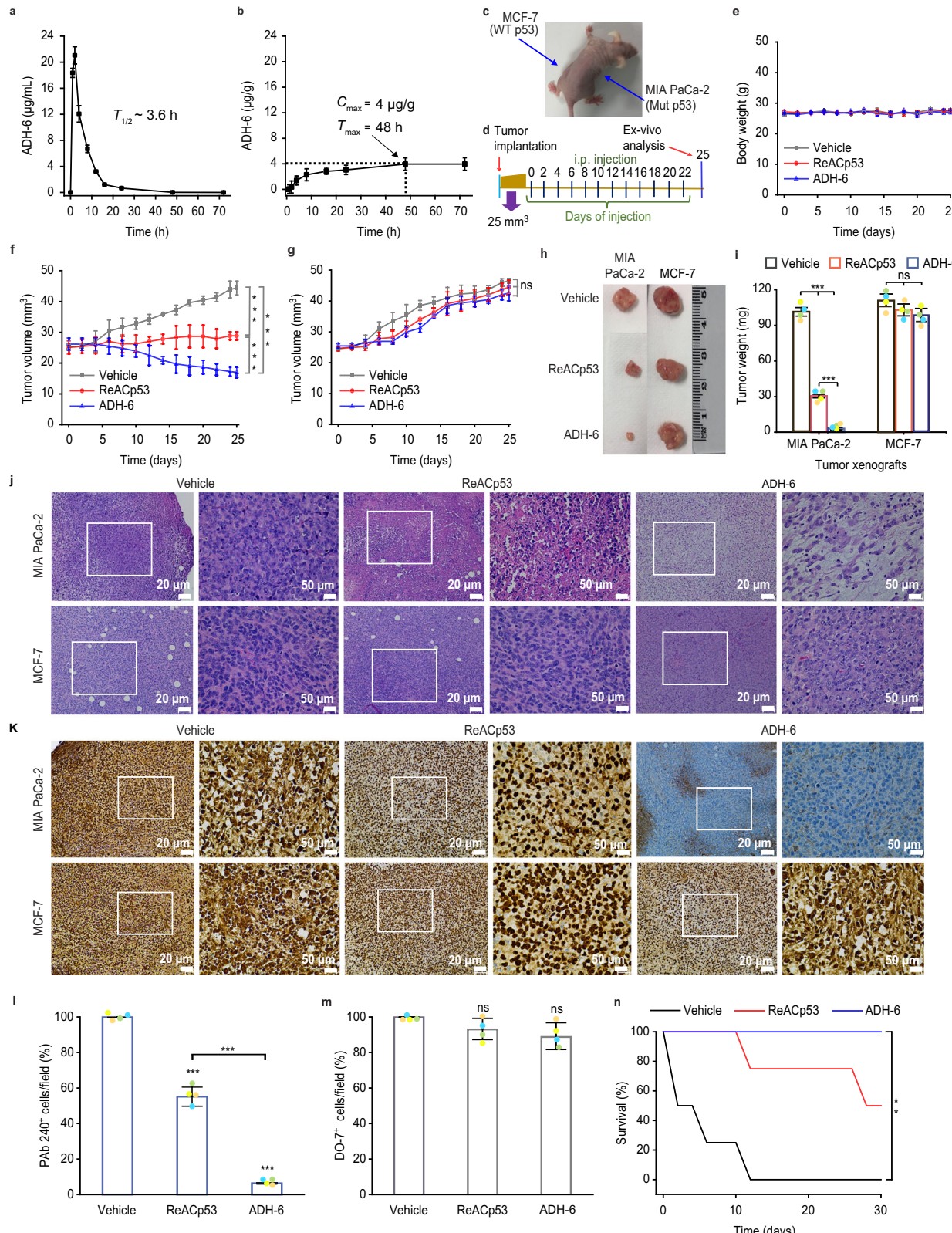

to ADH-6 treatment, became highly susceptible to the cytotoxic effects of the oligopyridylamide upon transfection with mutant R248W or R175H p53 (Fig. 4 and Supplementary Fig. 12).

Further analysis of the effects of ADH-6 on mutant p53 harboring cancer cells revealed that the oligopyridylamide induces cell cycle arrest and apoptosis, both of which are indicators of restored p53 function (Fig. 4 and Supplementary Fig. 11). ADH-

6-mediated restoration of WT-like activity of mutant p53 was confirmed using a number of complementary assays. ChIP-qPCR analysis revealed that ADH-6 treatment resulted in recruitment of mutant p53 to the WT protein's primary transcriptional targets, such as *Cdkn1a*, *PIG3*, and *NOXA* (Supplementary Fig. 14 and Supplementary Table 1), that are not only key mediators of p53-dependent cell cycle arrest and apoptosis[87–91], but also

**Fig. 7 ADH-6 causes regression of tumors bearing mutant, but not WT, p53. a, b** In vivo pharmacokinetics of ADH-6. Concentration of ADH-6 in plasma (**a**) and in MIA PaCa-2 xenografts (**b**) of mice ($n = 5–6$ per group), after an intraperitoneal injection of the oligopyridylamide (15 mg kg$^{-1}$), was quantified using LC-MS/MS[135]. Shown are the circulation half-life ($T_{1/2}$) (**a**) as well as the maximum (or peak) concentration ($C_{max}$) in tumors and the time to achieve $C_{max}$ ($T_{max}$) (**b**). Data presented are mean ± SD. **c, d** Design of the tumor reduction studies. A representative mouse bearing both MIA PaCa-2 (mutant R248W p53) and MCF-7 (WT p53) xenografts (**c**) and treatment schedule for the dual xenograft model (**d**). Once the tumor volume reached ~25 mm$^3$, the mice were randomized into the different treatment groups ($n = 8$ per group), which were injected intraperitoneally with vehicle (0.02% DMSO), ReACp53 (716.4 μM), or ADH-6 (716.4 μM). Injections were done every 2 days for a total of 12 doses, with the first day of treatment defined as day 0. **e** Body weight changes of the tumor-bearing mice in the different treatment groups monitored for the duration of the experiment (mean ± SD; $n = 8$). **f, g** Tumor volume growth curves for the MIA PaCa-2 (**f**) and MCF-7 (**g**) xenografts in the different treatment groups over the duration of the experiment (mean ± SD; $n = 8$). Tumor volume was calculated using Eq. (3). Statistical analysis was performed using one-way ANOVA followed by Tukey's post hoc test. $P < 0.0001$ for ADH-6 vs vehicle, ReACp53 vs vehicle and ADH-6 vs ReACp53 (**f**). **h, i** Tumor mass analysis for the different treatment groups. After 25 days of treatment, four mice per treatment group were sacrificed and the tumor tissues were isolated and imaged (**h**) and subsequently weighed to determine the tumor mass (**i**). Data presented are mean ± SD, and statistical analysis was performed using one-way ANOVA followed by Tukey's post hoc test. $P < 0.0001$ for ADH-6 vs vehicle, ReACp53 vs vehicle and ADH-6 vs ReACp53 (MIA PaCa-2 xenografts; **i**). **j** Hematoxylin and eosin (H&E)-stained xenograft sections from the different treatment groups following 25 days of treatment. Images on the right are magnified views of the boxed regions in the images on the left. Scale bar = 20 μm (50 μm for the magnified views). **k–m** Immunohistochemistry (IHC) analysis of the residual xenografts. Images of sections of MIA PaCa-2 and MCF-7 xenografts stained using the anti-p53 PAb 240 and DO-7 antibodies, respectively, from the different treatment groups (**k**). Images on the right are magnified views of the boxed regions in the images on the left. Scale bar = 20 μm (50 μm for the magnified views). Quantification of PAb 240 (**l**) and DO-7 (**m**) positive cells in 3–5 different fields of view expressed as % of the total number of cells (mean ± SD; $n = 4$). One-way ANOVA followed by Tukey's post hoc test: $P < 0.0001$ for ADH-6 vs vehicle, ReACp53 vs vehicle and ADH-6 vs ReACp53 (**l**). **n** Survival curves for the vehicle, ReACp53 and ADH-6 treatment groups over 30 days ($n = 4$ per group). Statistical analysis was performed using log-rank (Mantel-Cox) test. $P = 0.0062$ for ADH-6 vs vehicle. **$P < 0.01$, ***$P < 0.001$ or non-significant (n.s., $P > 0.05$) for comparisons with vehicle-treated controls and between the different treatment groups.

---

determinants of the cytotoxic response of cancer cells to standard chemotherapy and targeted cancer therapy[92,93]. Transcriptional reactivation of mutant p53 was further corroborated using RNA-Seq (Fig. 5 and Supplementary Figs. 16 and 17). Specifically, we observed activation of the *TP53* and apoptosis pathways, and suppression of genes involved in cell cycle progression (G2/M) and the *MYC* pathway, which are known to be negatively regulated by p53 (ref. [52]). Consistent with the transcriptome analysis, western blot quantification of protein levels in MIA PaCa-2 cells following treatment with ADH-6 revealed increased expression of direct p53 targets (Supplementary Fig. 15). Additionally, phosphoproteome analysis showed that ADH-6 downregulates key cancer-promoting phosphoproteins that are known to be directly negatively regulated by p53[58–62] (Fig. 6, Supplementary Figs. 18 and 19, and Supplementary Table 2). Taken together, our results clearly demonstrate that ADH-6 dissociates mutant p53 amyloid-like aggregates in cancer cells, and that the released protein is restored to a functional form, which elicits the observed inhibition of proliferation via cell cycle arrest and induction of apoptosis.

To establish whether the in vitro effects of ADH-6 are recapitulated in vivo, we evaluated the effects of ADH-6 on mice bearing MIA PaCa-2 (mutant R248W p53) or SK-BR-3 (mutant R175H p53) tumors, alone or with MCF-7 (WT p53) tumors on the opposite flank as an internal control (Fig. 7 and Supplementary Section 5). ReACp53, serving as a positive control, reduced MIA PaCa-2 tumor growth and prolonged survival relative to the vehicle-treated control groups, which is in agreement with the peptide's reported ability to inhibit growth of aggregation-prone mutant p53-bearing tumors[14]. However, ADH-6 was markedly more effective at decreasing MIA PaCa-2 tumor volume and mass, and prolonging survival, compared to ReACp53. The greater in vivo efficacy of ADH-6 compared to ReACp53 correlates well with their relative capacities to dissociate intracellular mutant p53 aggregates (Fig. 3 and Supplementary Figs. 7–9) and to induce toxicity in mutant p53-bearing cancer cells (Fig. 4 and Supplementary Fig. 11), as well as with their relative in vivo stabilities. Peptides possess a number of pharmaceutically desirable properties, including the ability to selectively bind to specific targets with

high potency, thereby minimizing off-target interactions and reducing the potential for toxicity[94,95]. On the other hand, a major disadvantage of peptides is their low in vivo stability[73]. Synthetic protein mimetics, with their constrained backbone, are inherently more stable than peptides[25,96], which is reflected in the substantially longer in vivo circulation half-life and prolonged presence in the bloodstream of ADH-6 relative to ReACp53. The extended in vivo circulation time of ADH-6 facilitates increased accumulation in tumor tissue and enhanced anticancer activity. Importantly, ADH-6 exhibited high potency against MIA PaCa-2 and SK-BR-3 tumors but did not affect growth of control MCF-7 xenografts, underlining the oligopyridylamide's specificity for tumors that harbor aggregation-prone mutant p53. Additionally, detailed necropsies of major organs revealed no damage or alterations, demonstrating the lack of toxicity of ADH-6 to healthy tissue. Thus, our results show that ADH-6 effectively shrinks tumors bearing aggregation-prone mutant p53 in vivo, without displaying the non-specific toxicity that is common to conventional cancer therapeutics, thereby greatly prolonging survival.

Mutant p53-associated cancers are predicted to lead to the deaths of more than 500 million people alive today[97]. This is fueled, in part, by the increasing incidences of notoriously difficult to treat malignancies, such as the highly lethal pancreatic cancer, which is on course to becoming the second leading cause of cancer-related mortality in Western countries within the next 5–10 years[98]. Consequently, there is a pressing need for new therapeutic strategies to supplement or supplant current cancer treatments. In the present study, we focused on a largely neglected—with a few notable exceptions—property of a sizeable subset of p53 mutants, namely their propensity to self-assemble into amyloid-like aggregates. This aggregation is implicated in mutant p53-associated tumor suppressor loss of function, oncogenic GoF and, potentially, prion-like propagation of these phenotypes[3,5,11,12,14,17,19]. Testing protein mimetics originally designed to antagonize amyloid formation associated with AD and T2D[20,21,23], we identified a cationic tripyridylamide, ADH-6, that effectively abrogates mutant p53 amyloid-like aggregation in human cancer cells, which restores p53's transcriptional activity, leading to cell cycle arrest and induction of apoptosis. Importantly, ADH-6 treatment causes

regression of xenografts harboring mutant, but not WT, p53 and prolongs survival, with no visible toxicity to healthy tissue. This study effectively establishes a bridge between amyloid diseases and cancer, providing a foundation for cross-informational approaches in the design of potent mutant p53-targeted cancer therapeutics.

## Methods

**Reagents**. 2-Deoxy-D-glucose, acetone, acetonitrile (ACN), 3′,5′-dimethoxy-4′-hydroxyacetophenone (acetosyringone), bovine serum albumin (BSA), calcium chloride (CaCl₂), chloroform, dimethyl sulfoxide (DMSO), 1,4-dithiothreitol (DTT), ethylenediaminetetraacetic acid (EDTA), gentamycin, glucose, glycolic acid, N-(2-hydroxyethyl)piperazine-N′-(2-ethanesulfonic acid) (HEPES), 1,1,1,3,3,3-hexafluoro-2-propanol (HFIP), indole-3-acetic acid (IAA), isoflurane, kanamycin, lithium chloride (LiCl), magnesium chloride (MgCl₂), methanol, 2-(N-morpho-lino)ethanesulfonic acid (MES), N-hydroxysuccinimide (NHS), paraformaldehyde (PFA), phosphotungstic acid (PTA), phosphate buffered saline (PBS), piperazine-N,N′-bis(2-ethanesulfonic acid) (PIPES), potassium chloride (KCl), Protease Inhibitor Cocktail, rifampicin, Sephadex G-25 DNA Grade column, sodium azide, sodium bicarbonate (NaHCO₃), sodium chloride (NaCl), sodium deoxycholate, sodium dodecyl sulfate (SDS), sodium hydroxide (NaOH), triethylammonium bicarbonate (TEAB), thioflavins S and T (ThS and ThT), trifluoroacetic acid (TFA), Triton X-100, Trizma, Tween 20, and Trypan Blue were all purchased from Sigma-Aldrich (St. Louis, MO). C18 Tips, Dead Cell Apoptosis kit (Alexa 488-conjugated annexin V/propidium iodide (PI)), formic acid, G418 sulfate (Geneticin), Hoechst 33342, hydroxylamine, Lipofectamine LTX Reagent, Lurai-bertani (LB), NP40 Cell Lysis Buffer Opti-MEM Reduced Serum Medium, Pierce BCA Protein Assay kit, Pierce Trypsin/Lys-C Protease Mix and tandem mass tag (TMT)-labeling kit were from Thermo Fisher Scientific (Waltham, MA). CellTiter 96 AQueous One Solution (MTS) Cell Proliferation Assay was purchased from Promega (Madison, WI).

**Synthesis of oligopyridylamide-based protein mimetics**. The protocols for synthesis, fluorescent dye-labeling and characterization of the relevant oligopyr-idylamides are presented in Supplementary Section 6. For each experiment, fresh stocks of the oligopyridylamides (10 mM) were prepared in DMSO and filtered using 0.22 μm Ultrafree-MC spin filters (Sigma).

**Peptide preparation**. Peptides (pWT, pR248W and ReACp53; sequences shown in Fig. 1c) were synthesized by Selleck Chemicals (Houston, TX) using standard Fmoc methods. The peptides were purified inhouse by reverse-phase high-performance liquid chromatography (Waters 2535 QGM HPLC), and purity was subsequently verified using mass spectrometry (Agilent 6538 QToF LC/MS). After purification, peptides were aliquoted into 0.5 mL Protein LoBind tubes (Eppendorf), lyophilized and stored at −80 °C until needed. For each experiment, fresh peptide stock solutions were prepared in DMSO and filtered using 0.22 μm Ultrafree-MC spin filters. pR248W concentration was determined by absorbance measurements at 280 nm ($\varepsilon = 5690$ cm$^{-1}$ M$^{-1}$ for tryptophan) on a Lambda 25 UV/Vis Spectro-photometer (PerkinElmer, Waltham, MA) using quartz cuvettes (1 cm path-length).

**Thioflavin T (ThT)-based aggregation assay**. Peptide (pWT and pR248W) amyloid formation kinetics were measured in quadruplicate in black 96-well plates with flat bottom (Corning Inc., NY) using a Synergy H1MF Multi-Mode micro-plate reader controlled by Gen5 software (version 2.0; BioTek, Winooski, VT). The aggregation was initiated by dilution of peptide from a freshly prepared stock solution (1 mM in DMSO) to PBS containing ThT, with or without oligopyr-idylamides. For some experiments, the oligopyridylamides were added (from a stock solution of 10 mM in DMSO) at the indicated timepoint after the start of the aggregation reaction. To maintain identical conditions, an equal amount of DMSO was added to the wells with peptide only reactions. Final concentrations in the wells were: 25 μM pWT or pR248W; 0 or 25 μM oligopyridylamide; 50 μM ThT. Peptide aggregation was monitored by shaking and measuring the ThT fluorescence ($\lambda_{ex/em}$ = 440/480 nm) at 5-min intervals at 37 °C. The sample data were processed by subtracting the blank and renormalizing the fluorescence intensity by setting the maximum value to one.

**Transmission electron microscopy (TEM)**. Twenty-five micromoles of 25 μM pR248W was incubated, alone or co-mixed with an equimolar concentration of ADH-1, ADH-6, or ReACp53, for 10 h at 37 °C. Thereafter, the solution was vortexed and a 5 μL droplet of the solution was placed on a freshly plasma-cleaned copper/formvar/carbon grid (400 mesh, Ted Pella, Redding, CA). After 2 min, the droplet was wicked away by carefully placing the grid perpendicularly on a filter paper. The grid was then gently washed by dipping the upper surface in a water droplet and dried as described above using a filter paper. Next, the grid was placed upside down for 20 s in a 1% phosphotungstic acid (PTA) solution that had been passed through a 0.2 μm filter and centrifuged at 24,000 × g for 10 min to remove potential PTA agglomerates. The PTA solution was wicked away and the grid was

further dried for 10 min under cover at room temperature. Images of at least five different grid regions were acquired per experimental condition on a Talos F200X TEM (Thermo Fisher Scientific) equipped with a Ceta 16 M camera operated at an accelerating voltage of 200 kV. Velox software (version 2.9.0; Thermo Fisher Sci-entific) was used for image analysis.

**Dot blot immunoassay**. Samples of 10 μM pR248W, in the absence or presence of an equimolar amount of the ligands (ADH-1 or ADH-6), were incubated for various durations. The samples were then applied to a nitrocellulose membrane and dried for 1 h at room temperature or overnight at 4 °C. The membranes were subsequently blocked with 5% nonfat milk in TBST buffer (10 mM Tris, 0.15 M NaCl, pH 7.4, supplemented with 0.1% Tween 20) for 2 h at room temperature, washed thrice with TBST buffer, and incubated overnight at 4 °C with the poly-clonal A11 antibody (AHB0052; Thermo Fisher Scientific) (1:1000 dilution in 5% nonfat milk in TBST buffer). Next, the samples were washed with TBST buffer (×3) and incubated with horseradish peroxidase (HRP)-conjugated anti-rabbit IgG (ab6721; Abcam, Cambridge UK) secondary antibody (1:500 dilution in 5% nonfat free milk in TBST buffer) for 1 h at room temperature. The dot blots were then washed with TBST buffer (×5), developed using the ECL reagent kit (Amersham, Piscataway, NJ), and finally imaged using a Typhoon FLA 9000 instrument (GE Healthcare Life Sciences, Pittsburgh, PA) with the settings for chemiluminescence.

**Circular dichroism (CD) spectroscopy**. Each freshly prepared pR248W stock solution was diluted in PBS to a concentration of 10 μM, alone or with an equi-molar concentration of the ligand (ADH-1 or ADH-6), and transferred to a 1-mm quartz cuvette. CD spectra were recorded from 190 to 260 nm, with a bandwidth of 0.5 nm, a step size of 0.5 nm, and a time-per-point of 10 s, at 25 °C using a Chirascan Plus Circular Dichroism Spectrometer (Applied Photophysics Limited, UK) with a Peltier temperature control system. For each experiment, the CD spectra from three independent trials were averaged and baseline corrected.

**Tryptophan fluorescence quenching assay**. The binding interaction between the oligopyridylamides (ADH-1 and ADH-6) and mutant pR248W was characterized using a published steady-state intrinsic tryptophan fluorescence quenching assay[99,100]. A 1 mM stock solution of pR248W was prepared in PBS. The peptide solution was then diluted to a final concentration of 5 μM and equilibrated at 20 °C for 5 min before commencing titration. Subsequently, the ligands (from a stock concentration of 10 mM in DMSO) were serially added to the peptide solution, which was stirred and allowed to equilibrate for 10 min before the intrinsic tryp-tophan fluorescence was measured from 300 to 420 nm ($\lambda_{ex}$ = 280 nm) on a Cary Eclipse Fluorescence Spectrophotometer (Agilent, Santa Clara, CA). The back-ground corrections in fluorescence changes because of dilution effects with the stepwise addition of ligands were made by buffer titrations. Curve fitting and the analysis of the data were performed using Prism (version 8.4.2; GraphPad Software Inc., La Jolla, CA, USA). All binding curves were fit to the one-site-specific binding model to determine equilibrium dissociation constants ($K_d$):

$$Y = \frac{B_{max} \times X}{K_d + X} \tag{1}$$

where $X$ is the final concentration of the ligand, $Y$ is the percentage increase in pR248W fluorescence quenching relative to the control fluorescence signal (i.e. pR248W alone), and $B_{max}$ is the extrapolated maximum specific binding.

**NMR spectroscopy**. NMR samples were prepared with the DBDs of WT and mutant R248W p53. The $^{15}$N-uniformly labeled proteins were expressed and purified using published procedures[37] by Giotto Biotech S.r.l. (Sesto Fiorentino, Italy) and supplied as 1 mg/mL solutions in buffer (50 mM Tris, 200 mM KCl, 5 mM DTT (pH 7.5)). After mild concentration by ultracentrifugation and spec-troscopic inspection, the original buffer was replaced by dialysis, first against 50 mM KH₂PO₄/K₂HPO₄, 150 mM KCl, 5 mM DTT (pH 6.8), and subsequently against H₂O with 5 mM DTT which resulted in final pH values of 5.6–5.8. The protein concentrations were in the 40–50 μM range in Tris and phosphate buffers, and in the 19–24 μM range in water, with 4–5% D₂O always present. Additions of a few microlitres of ADH-6 (from 5 or 24 mM stock solutions in H₂O or a 200 mM stock solution in DMSO) were performed to assess the protein–drug interaction.

NMR spectra were collected at 14.0 T ($^1$H resonance at 600.19 MHz) on a Bruker Avance III NMR system equipped with triple resonance cryoprobe. All the spectra were acquired at 293.2 K. Two-dimensional $^{15}$N–$^1$H HSQC experiments[101] were executed using sensitivity-improved Echo/Antiecho-TPPI pure phase detection in F1, gradient-based coherence selection and flip-back pulse for solvent suppression[102–104]. To improve the resolution, the corresponding TROSY-based experiments[105,106] were also collected. All acquisitions were carried out over spectral widths of 40 and 16 ppm in F1 and F2 dimensions, respectively, with 128 time-domain points in t1, 128–256 scans × 2048 points in t2 and 64 dummy scans to achieve steady state. The NMR data were processed with TOPSPIN (version 4.0.2; Bruker). Prior to Fourier transformation, linear prediction in t1 (up to 256 points) and zero filling were applied to yield a final data set of 2K × 1K points. The chemical shift variations ($\Delta\delta$) of the protein peaks from $^{15}$N-$^1$H HSQC spectra

were analyzed in terms of cumulated CSP:

$$CSP = \sqrt{\Delta\delta_H^2 + \left(\frac{\Delta\delta_N}{6.5}\right)^2} \qquad (2)$$

to properly account for both $^1H$ ($\Delta\delta_H$) and $^{15}N$ ($\Delta\delta_N$) frequency changes upon ligand interaction.

**Cell culture**. Prior to use, all cell lines were authenticated and tested for mycoplasma contamination by Charles River Laboratories (Margate, UK). The following cell lines were purchased from American Type Culture Collection (ATCC; Manassas, VA): human bone (U-2 OS [ATCC no. HTB-96] and Saos-2 [HTB-85]), brain (SH-SY5Y [CRL-2266]), breast (HCC70 [CRL-2315], MCF-7 [HTB-22], MDA-MB-175-VII [HTB-25], MDA-MB-231 [HTB-26] and SK-BR-3 [HTB-30]), colon (COLO 320DM [CCL-220], HT-29 [HTB-38], LS123 [CCL-255] and LS 174T [CL-188]), gastric (AGS [CRL-1739]), leukemia (ARH-77 [CRL-1621] and CESS [TIB-190]), lung (A549 [CCL-185], NCI-H1770 [CRL-5893], NCI-H1882 [CRL-5903], NCI-H2342 [CRL-5941] and NCI-H748 [CRL-5841]), ovarian (OVCAR-3 [HTB-161] and SKOV-3 [HTB-77]), pancreatic (MIA PaCa-2 [CRL-1420]), and renal (CAKI-1 [HTB-46]), cancer cells. The cells were cultured in medium (DMEM, RPMI 1640 or Ham′s Nutrient Mixture F12; Sigma) supplemented with 10% fetal bovine serum (FBS; GE Healthcare Life Sciences, Logan, UT), 4 mM L-glutamine, 1 mM sodium pyruvate and 1% penicillin/streptomycin (all from Sigma) in 5% $CO_2$ at 37 °C, and viability was monitored regularly using the Trypan Blue exclusion test on a TC20 automated cell counter (Bio-Rad, Hercules, CA). Once the cells reached ~95% confluence, they were split (using 0.25% trypsin-EDTA; Sigma) into fractions and propagated or used in experiments.

**ThS fluorescence imaging**. The benzothiazole dye ThS was used to stain intracellular mutant p53 aggregates[39]. MIA PaCa-2 cells were seeded at a density of $2 \times 10^4$ cells/well in 500 μL complete DMEM in four-chambered 35 mm glass bottom Cellview cell culture dishes (Greiner Bio-One, Monroe, NC). After culturing for 24 h, the medium was replaced by fresh medium containing 125 μM of ThS and incubated for 1 h. Thereafter the medium was removed, the cells were washed, and fresh DMEM containing 10 μM ADH-1, ADH-6, or ReACp53 was added and incubated for 0.5–6 h. Finally, the medium was once again replaced with fresh DMEM and the cells were imaged on an Olympus Fluoview FV-1000 confocal laser scanning microscope, using a ×63 Plan-Apo/1.3 NA oil immersion objective with DIC capability. Images were acquired using the FV10-ASW Viewer software (version 4.2; Olympus) and analyzed using the Fiji image processing software (ImageJ2 core with plugins for scientific image analysis; https://imagej.net/Fiji).

**Immunofluorescence**. MIA PaCa-2 and MCF-7 cells were seeded at a density of $2 \times 10^4$ cells/well in 500 μL complete DMEM in four-chambered 35 mm glass bottom Cellview cell culture dishes and cultured for 24 h. Following treatment with vehicle or 2.5–10 μM ADH-1, ReACp53, or ADH-6 (unlabeled or FITC-labeled, ADH-6$_{FITC}$) for 0.5 or 6 h, the cells were rinsed with ice-cold PBS, fixed with 4% PFA, permeabilized with 0.5% Triton X-100, and blocked in 5% BSA. Subsequently, the cells were incubated with 5 μg/ml p53 antibody (PAb 240, sc-99; Santa Cruz Biotechnology, Dallas, TX), overnight at 4 °C, which was followed by incubation with Alexa 488- or Alexa 594-labeled anti-mouse IgG (ab150117 or ab150120, respectively; Abcam) secondary antibody (1:800 dilution) for 1 h at room temperature. Finally, the cells were imaged on an Olympus Fluoview FV-1000 confocal laser scanning microscope, and the images were analyzed using the Fiji image processing software.

**Cell fractionation and western blot analysis**. Cell fractionation and western blot were done using standard procedures[107,108]. Cells were washed with ice-cold PBS, harvested by trypsinization, and centrifuged. The resulting cell pellet was lysed using 100 μL lysis buffer (Bio-Rad), and lysis was confirmed by light microscopy. For separation of the insoluble (pellet) and soluble (supernatant) fractions, the whole-cell lysate was centrifuged ($20,000 \times g$ for 20 min at 4 °C), and the protein content of the two fractions was determined using the bicinchoninic acid (BCA) protein assay[109].

Samples (20 μg/lane) were electrophoresed on a 4–12% sodium dodecyl sulfate (SDS)-polyacrylamide gel in a Mini-PROTEAN Tetra Cell (Bio-Rad) and subsequently transferred to a nitrocellulose membrane (Sartorius AG, Göttingen, Germany), which was blocked with 5% nonfat milk and then incubated with p53 (DO-7, sc-47698; Santa Cruz), p21 (EPR3993, ab109199; Abcam), Noxa (114C307, sc-56169; Santa Cruz), Bax (E63, ab32503; Abcam) or MDM2 (2A10, ab16895; Abcam) antibodies (1:1000), or β-actin antibody (2A3, sc-517582; Santa Cruz) (1:2000), overnight at 4 °C. This was followed by incubation for 3 h at room temperature with horseradish peroxidase (HRP)-conjugated anti-mouse or anti-rabbit IgG (ab205719 or ab6721, respectively; Abcam) secondary antibody (1:10,000). Finally, the samples were visualized using Clarity Western ECL Substrate (Bio-Rad) on a Bio-Rad Gel Doc XR+ Gel Documentation System (Image Lab software, version 4.1). For densitometric quantification, the immunoblot bands were normalized to β-actin.

**Cellular thermal shift assay (CETSA)**. CETSA experiments were performed according to a published protocol[43]. MIA PaCa-2 or SK-BR-3 cells were treated with vehicle or 5 μM ADH-1 or ADH-6. After incubation for 6 h, the cells were washed with ice-cold PBS, harvested by trypsinization, centrifuged, and re-suspended in PBS supplemented with complete Protease Inhibitor Cocktail. Equal amounts of cell suspensions were aliquoted into 0.2 mL PCR microtubes and incubated for 30 min at room temperature. Subsequently, the cell suspension aliquots were heated individually at the indicated temperatures for 3 min (Ristretto Thermal Cycler; VWR, Darmstadt, Germany), followed by cooling for 3 min at on ice. Finally, the cells were lysed using two cycles of freeze–thawing, and the soluble fractions were isolated by centrifugation and analyzed by SDS-polyacrylamide gel electrophoresis (SDS-PAGE) followed by western blot as described above. Fold-changes in the immunoblot band densities (normalized to the β-actin controls and using the lowest temperature condition as reference) were plotted as a function of temperature to generate mutant p53 melt curves for the different treatments. Curve fitting and calculation of the melting point, i.e. the temperature at which half of the protein amount has been denatured ($T_m$), were done using published procedures[43,44].

**Plant cell studies**

*Plants and growth conditions*. Nicotiana benthamiana were sown in soil and grown at 24 °C (morning) and 22 °C (night) in a growth chamber under an 8 h-light/16 h-dark cycle. After 2 weeks, the seedlings were transplanted into pots (one seedling per pot) and grown for a further 5–6 weeks before conducting Agrobacterium tumfaciens-mediated transient assays.

*Plasmid construction*. WT and mutant R248W p53 DBDs were cloned using the Gateway system (Thermo Fisher Scientific). Genes for WT and R248W p53 DBDs were respectively amplified from the vector pCMV-Neo-Bam carrying WT and R248W p53 constructs (plasmids #16434 and #16437, respectively; Addgene, Watertown, MA). Gene-specific primers for the p53 DBDs were designed to incorporate the CACC tag at the 5′ end of both constructs. The following primers were used for amplifying p53 DBDs: p53 forward, 5′-caccTCATCTTCTGTCCC TTCCCAGAAAACC-3′; and p53 reverse, 5′-tcaGGGCAGCTCGTGGTGAGGC TCCCCT-3′ (note that the underlined CACC tag in lowercase is not a part of the p53 sequence) (Supplementary Table 1a). Phusion DNA Polymerase (New England Biolabs) was used for the gene-specific amplification of the p53 DBD constructs. The PCR amplified DNA constructs were subcloned in pENTR Directional TOPO (D-TOPO) (Thermo Fisher Scientific). The gateway binary vectors pEarleyGate 101 (35S promoter, C:YFP) and pEarleyGate 104 (35S promoter, N:YFP) were used as the destination vectors for the p53 DBD constructs. The WT p53 DBD was cloned with a YFP tag at the N terminus and an amino acid sequence, RVGAPTQLSCTKWC, at the C-terminal end that corresponds to the att site (5′-aagggtgggcgcgccgacccagctttcttgtacaaagtggTGCTAG-3′). The R248W p53 DBD, also cloned with a YFP tag at the N terminus, had a stop codon (TAA) at the C-terminal end. (Note: the nucleotides in lowercase are the att sites, while the uppercase nucleotides are part of the vector. The stop codon is underlined.) Since the entry and destination vectors had the same antibiotic selection marker (kanamycin), we followed the PCR amplification based (PAB) method[110] to recombine the DNA fragment from the entry vector into the destination vector. The DNA fragment was amplified from the entry vector using M13 primers (M13 forward, 5′-GTAAAACGACGGCCAG-3′, and M13 reverse, 5′-CAGGAAA-CAGCTATGAC-3′; Supplementary Table 1a) and was subsequently recombined with the destination vector using LR Clonase (Thermo Fisher Scientific). The plasmids were then transformed in Agrobacterium tumfaciens strain GV3101.

*Agrobacterium-mediated transient expression*. N. benthamiana leaves were infiltrated with A. tumfaciens carrying the desired constructs using published protocols[111,112]. A. tumfaciens strain GV3101 carrying the WT and mutant R248W p53 DBDs were grown overnight at 28 °C in Lurai-bertani (LB) medium containing the appropriate antibiotics (100 μg/mL kanamycin, 50 μg/mL gentamycin, or 100 μg/mL of rifampicin). Cells were re-suspended in induction media (10 mM MES, pH 5.6, 10 mM $MgCl_2$ and 200 μM acetosyringone) and incubated for 2 h at room temperature before infiltration (at a final OD of 1.0) of N. benthamiana leaves.

*Confocal microscopy analysis*. Intracellular localization of YFP:p53DBD$^{WT}$ and YFP:p53DBD$^{R248W}$ was observed at 72 h post infiltration (hpi). To determine the effects of ADH-1, ReACp53, and ADH-6 on p53 DBD aggregation, at 48 hpi the oligopyridylamides/peptide were diluted in 10 mM $MgCl_2$ to a final concentration of 5 μM and then infiltrated into the N. benthamiana leaves for 24 h. The leaves from the infiltrated area were isolated and cells towards the abaxial side were imaged on a NIKON Eclipse LV100 upright microscope, and the images were analyzed using the Fiji image processing software.

**Transfection of p53 null cancer cells with mutant p53**. Human bone cancer Saos-2 and ovarian cancer SKOV-3 cells were plated at a density of $1 \times 10^5$ cells/well in 2 mL complete medium in six-well plates and cultured for 48 h. For each well of cells to be transfected, 2.5 μg pCMV-Neo-Bam p53 R175H or pCMV-Neo-Bam p53 R248W vectors (plasmids #16436 or #16437, respectively; Addgene)

containing the $amp^r$ gene was diluted in 500 μL Opti-MEM Reduced Serum Medium in an Eppendorf tube, to which 10 μL Lipofectamine LTX Reagent was added, mixed gently, and incubated for 30 min at room temperature. The medium in the wells was then replaced with the Opti-MEM Reduced Serum Medium containing the DNA–Lipofectamine LTX Reagent complexes and the cells were incubated in 5% $CO_2$ at 37 °C for 24 h. Subsequently, the cells were harvested and plated in T25 flasks (Corning). After culturing until ~95% confluence was reached, the cells were selected with 400 μg/mL G418 sulfate-supplemented medium and maintained thereafter in medium containing 100 μg/mL G418 sulfate.

To verify that the transfection was successful, the pCMV-Neo-Bam p53 R248W and pCMV-Neo-Bam p53 R175H vectors containing the $amp^r$ gene were purified from the transfected SKOV-3 and Saos-2 cells using the Qiaprep Miniprep Kit (Qiagen, Hilden, Germany) according to the manufacturer's instructions. The miniprep-purified DNA samples were subsequently quantified using a NanoDrop spectrophotometer. PCR was carried out to amplify the $amp^r$ genes using the following primers: forward (5′-AGATTATCAAAAAGGATCTTCACCT-3′) and reverse (5′-CCTCGTGATACGCCTATTTTTATAG-3′) (Integrated DNA Technologies, Leuven, Belgium) (Supplementary Table 1b). Each PCR reaction was prepared in 50 μL volume containing: 1 μL of each primer (0.2 μM final primer concentration), 50 ng DNA, 25 μL Taq 2× PCR Master Mix (New England Biolabs, Ipswich, MA), and nuclease-free water. The conditions for each PCR reaction were initial denaturation at 95 °C for 30 s, 35 cycles of denaturation at 95 °C for 30 s, annealing at 50 °C for 30 s, and extension at 68 °C for 1 min. As a negative control, a PCR reaction was carried out with the same reagents sans DNA. The amplified products were then resolved by electrophoresis using a 3% agarose gel prepared in 1× TAE (Tris/acetic acid/EDTA) buffer (Bio-Rad) containing GelGreen Nucleic Acid Stain (Biotium, Freemont, CA), which was visualized on an E-Gel Imager Blue-Light Base (Thermo Fisher Scientific). The presence of the successfully amplified $amp^r$ PCR products was confirmed using a standard 500 bp ladder.

**Cell viability/toxicity assays**. Cell viability/toxicity was measured using two complementary assays: (i) CellTiter 96 AQueous One Solution (MTS) assay, which measures reduction of the tetrazolium compound MTS (3-(4,5-dimethylthiazol-2-yl)-5-(3-carboxymethoxyphenyl)-2-(4-sulfophenyl)-2H-tetrazolium, inner salt) to soluble formazan, by mitochondrial NAD(P)H-dependent dehydrogenase enzymes, in living cells[113,114] and (ii) Dead Cell Apoptosis assay, in which Alexa 488-conjugated annexin V is used as a sensitive probe for detecting exposed phosphatidylserine in apoptotic cells[115], and red-fluorescent propidium iodide (PI), a membrane impermeant nucleic acid binding dye, assesses plasma membrane integrity and distinguishes between apoptosis and necrosis[116].

For the MTS assay, cells were seeded at densities of $5 \times 10^3$–$2 \times 10^4$ cells/well in 100 μL complete medium in standard 96-well plates. After culturing for 24 h, the medium was replaced with serum-free medium containing the oligopyridylamides at the desired concentrations, and incubated for the indicated durations, at 37 °C. Thereafter, the incubation medium was replaced with fresh complete medium, and 20 μL MTS reagent was added to each well and incubated for 2 h at 37 °C. Finally, absorbance of the soluble formazan product ($\lambda = 490$ nm) of MTS reduction was measured on a Synergy H1MF Multi-Mode microplate reader (BioTek), with a reference wavelength of 650 nm to subtract background. Wells treated with vehicle were used as control, and wells with medium alone served as a blank. MTS reduction was determined from the ratio of the absorbance of the treated wells to the control wells.

For the Dead Cell Apoptosis assay, MIA PaCa-2 cells were treated with 5 μM ADH-1, ADH-6, or ReAcp53 for 24 h at 37 °C. Subsequently, the cells were washed with ice-cold PBS, harvested by trypsinization, centrifuged, and re-suspended in 1× annexin-binding buffer (10 mM HEPES, 140 mM NaCl, 2.5 mM $CaCl_2$, pH 7.4) to a density of ~$1 \times 10^6$ cells/mL. The cells were then stained with 5 μL Alexa 488-conjugated annexin V and 0.1 μg PI per 100 μL of cell suspension for 15 min at room temperature. Immediately afterwards, fluorescence was measured using flow cytometry (≥10,000 cells/sample) on a BD FACSAria III cell sorter (BD Biosciences, San Jose, CA) controlled by BD FACSDiva software (version 8.0), and the fractions of live (annexin V−/PI−), early and late apoptotic (annexin V+/PI−, and annexin V+/PI+, respectively), and necrotic (annexin V−/PI+) cells were determined using FlowJo (version 10.6; FlowJo LLC, Ashland, OR).

**Cell cycle analysis**. Cell cycle distribution was determined using flow cytometry[117,118]. MIA PaCa-2 cells were seeded at a density of $1 \times 10^6$ cells/well in 1 mL complete DMEM. After culturing for 24 h, the medium was replaced with serum-free DMEM containing 5 μM ADH-1, ReAcp53, or ADH-6, and the cells were incubated for 6 h. Subsequently, the cells were washed with PBS, harvested by trypsinization, and centrifuged ($800 \times g$ for 5 min). The supernatant was discarded, and the cells were washed with PBS and fixed in 70% cold ethanol overnight at 4 °C. The cells were then washed with PBS, filtered through a nylon sieve, and centrifuged again ($800 \times g$ for 5 min), and the supernatant was discarded. The cells were stained with Nuclear Green CCS1 (Abcam) for 30 min at 37 °C and data (10,000 cells/sample) were collected on a BD FACSAria III cell sorter and subsequently analyzed using FlowJo.

**Chromatin immunoprecipitation (ChIP) coupled with quantitative PCR (ChIP-qPCR)**. MIA PaCa-2 cells were cultured in complete DMEM in T175 flasks (Corning) until 70–80% confluence was reached. Thereafter, the medium was replaced with serum-free DMEM containing vehicle or 5 μM ADH-6, and the cells were incubated for a further 24 h. The cells were then treated with 1% PFA for 10 min at 37 °C for crosslinking, followed by a single wash with 5 mL PBS. In all, 1.5 mL buffer A (5 mM PIPES, pH 8, 85 mM KCl, 0.5% NP40, Protease Inhibitor Cocktail) was added to the cells, which were harvested using a pipette tip and scrapper. The cell suspension was rotated for 10 min at 4 °C, centrifuged ($1700 \times g$ for 5 min at 4 °C), and the supernatant was discarded. To the cell pellet, 400 μL buffer B (1% SDS, 10 mM EDTA, 50 mM Tris-HCl, pH 8.1) was added, and the cell suspension was sonicated (10 cycles of 5 s each at maximum power, with 2 min on ice between cycles to prevent overheating). Cell debris was removed by centrifugation ($21,000 \times g$ for 10 min at 4 °C), and the supernatant was collected. The samples were diluted tenfold with the immunoprecipitation buffer (0.01% SDS, 1.1% Triton X-100, 1.2 mM EDTA, 16.7 mM Tris-HCl, pH 8.1), rotated for 30 min at room temperature, followed by centrifugation ($300 \times g$ for 20 s at 4 °C). Of the collected supernatant, 100 μL was used as input for each sample, with the remainder incubated with 1 μg/mL of the antibody for p53 (DO-1, sc-126; Santa Cruz), p63 (D2K8X, #13109; Cell Signaling Technology, Danvers, MA), or p73 (EPR19884, ab215038; Abcam), overnight on a rotator at 4 °C. Fifty microliters magnetic beads (Magna ChIP protein A+G bead blend; Sigma) was added to each sample solution and rotated for 30 min at room temperature. After rotation, the samples were incubated for 30 s on a magnetic stand to separate the chromatin, which was then washed twice with wash buffer 1 (0.1% SDS, 1% Triton X-100, 2 mM EDTA, 20 mM Tris-HCl, pH 8.1, 150 mM NaCl), followed by twice with wash buffer 2 (0.1% SDS, 1% Triton X-100, 2 mM EDTA, 20 mM Tris-HCl, pH 8.1, 300 mM NaCl), twice with wash buffer 3 (0.25 M LiCl, 1% NP40, 1% sodium deoxycholate, 1 mM EDTA, 10 mM Tris-HCl, pH 8.1), and finally once with TE buffer (10 mM Tris-HCl, pH 8.1, 1 mM EDTA). Immediately after the washes, the samples were eluted by adding the elution buffer (0.5% SDS, 5 mM EDTA, 25 mM Tris-HCl, pH 7.5, 20 mM NaCl), followed by vortexing and incubation for 15 min at 65 °C. Subsequently, the samples were incubated on a magnetic stand and 200 μL supernatant was collected, to which 200 μL immunoprecipitation buffer was added, followed by 1 μL Protease Inhibitor Cocktail, and the samples were incubated overnight at 65 °C. DNA was recovered with a conventional inhouse DNA precipitation method using sodium acetate, with samples eluted in volumes of 20 μL. Finally, the recovered DNA was amplified and quantified (for qPCR reaction: 1 μL DNA from ChIP samples or input, 1 μL 2 mM primers (Supplementary Table 1c), 5 μL SYBr Green PCR Master Mix (Thermo Fisher Scientific), and 3 μL nuclease-free water). PCR thermal reaction cycle included one cycle of 10 min at 95 °C and 39 cycles of 15 s at 95 °C followed by 45 s at 60 °C. Normalization of PCR reaction was based on calculating the enrichment by comparing the threshold cycle number ($C_T$) for the input and ChIP samples.

**RNA sequencing (RNA-Seq)**

*RNA isolation and purification*. Total RNA was extracted from MIA PaCa-2 cells ($1 \times 10^6$ cells/well) treated with vehicle (control (C), $n = 3$) or oligopyridylamides (5 μM ADH-1 or ADH-6, $n = 3$) using a combination of both TriZol (Thermo Fisher Scientific) and RNAeasy Mini Kit (Qiagen) with modification of the manufacturers' protocol. Prior to extraction, cells were washed once with 1 mL PBS, suspended in 1 mL TriZol LS reagent and transferred to 1.5 mL Lobind tubes (Eppendorf) along with 0.2 mL chloroform. The TriZol lysate and chloroform mix were vortexed and centrifuged ($18,000 \times g$ at 4 °C) for 15 min, forming three distinctive phases of DNA (lower phase), protein (middle phase), and RNA (upper phase). The upper aqueous phase (~400 mL of RNA) was transferred to a new 1.5 mL tube where an equivalent volume of 70% ethanol was added and mixed. The mixture was further purified using an RNAeasy mini spin column (Qiagen) according to the manufacturer's protocol. RNA concentrations were determined using a NanoDrop 2000 spectrophotometer (Thermo Fisher Scientific, Waltham, MA).

*RNA-Seq library preparation, sequencing, and processing*. Total RNA quality was estimated based on the RNA integrity number (RIN) using a BioAnalyzer 2100 (Agilent, Santa Clara, CA). RNA samples with a RIN > 8 were used for library preparation. RNA-Seq libraries were prepared using an Illumina TruSeq Stranded mRNA prep kit as per the manufacturer's protocol. Samples were barcoded, multiplexed, and sequenced (100 bp pair-end) using a NextSeq 550 System (Illumina, San Diego, CA). DESeq2 computational pipeline[119] was used to estimate the raw count reads aligned to the reference genome. Human reference genome (GRCh38/hg38) from UC Santa Cruz Genome Browser (https://genome.ucsc.edu/) was utilized as a reference genome[120]. Computations were run on a Linux based command system on the NYU Abu Dhabi High Performance Computing (HPC) server platform Dalma (https://wikis.nyu.edu/display/ADRC/Cluster+-+Dalma). Raw data were processed, and count data were normalized (to log2 counts per million reads (log2cpm)) or transformed (using variance-stabilizing transformation (VST)).

*Bioinformatic and computational analysis*. Correlation (i.e. PCA, distance dendrogram) and expression (i.e. heatmaps, boxplots) analyses were generated either via RNA-Seq START (Shiny Transcriptome Analysis Resource Tool) application

(NYU Abu Dhabi Center of Genomic and Systems Biology (NYUAD-CGSB) Bioinformatics Online Analysis and Visualization Portal (http://tsar.abudhabi.nyu.edu/))[121] or the JMP genomics software (version 9.1; https://www.jmp.com/en_us/software/genomics-data-analysis-software.html). Significance was assessed by false discovery rate (FDR) adjusted P-value (P-adj or q-value), which was obtained from the hypergeometric P-value that was corrected for multiple hypothesis testing using the Benjamin and Hochberg procedure[122]. For GO analysis, DEGs, based on corrected statistical significance (P-adj < 0.05), were submitted to the Database for Annotation, Visualization Integrated Discovery (DAVID) bioinformatics tool (version 6.8; https://david.ncifcrf.gov/home.jsp)[123] to determine characteristics under the "biological processes" category. Enrichment term values were based on statistical significance of P < 0.05, which were normalized to −log10(P-value). GO analysis charts were prepared using Excel.

IPA, performed with Upstream Regulator Analysis module of Ingenuity Pathway Analysis software (QIAGEN Inc., Redwood City, CA; https://digitalinsights.qiagen.com), was used to identify TRs responsible for gene dysregulation in ADH-6 vs ADH-1 and ADH-6 vs control comparisons. For each TR, the q-value and z-score, along with the number of genes, was obtained. The number of genes that showed a pattern consistent with a TR's activated or repressed status were plotted along with the z-score and −log10(q-value) in a 2D plot generated in R. Gene set enrichment analysis (GSEA, version 2.2.3; http://software.broadinstitute.org/gsea/) was performed using the variance-stabilized counts obtained from DESeq2 analysis against the Molecular Signatures Database (MSigDB; http://www.gsea-msigdb.org/gsea/msigdb/index.jsp)[124,125] for identification of hallmark signatures. The enrichment plots, along with normalized enrichment scores and q-values, were obtained from GSEA and are shown on each plot.

## Quantitative proteomics

*Protein extraction and preparation of digests.* MIA PaCa-2 cells ($1 \times 10^6$ cells/well) were treated with vehicle or 5 μM ADH-1, ReACp53, or ADH-6 for 16 h, harvested, centrifuged, and lysed by adding five cell-pellet volumes of lysis buffer (100 μL lysis buffer for every 20 μL cell pellet). The cell lysate was centrifuged ($16,000 \times g$ for 10 min at 4 °C) and the supernatant was transferred to a 1.5 mL LoBind tube. The protein content of the sample was determined using the BCA protein assay. An aliquot (100 μg of protein per treatment) was diluted to a final volume of 100 μl with 100 mM TEAB buffer. Thereafter, the proteins were reduced (10 mM DTT for 1 h at 55 °C), S-alkylated (25 mM IAA, 30 min in the dark) and precipitated by adding six volumes (~600 μL) of pre-chilled (−20 °C) acetone for 16 h. The protein pellet collected by centrifugation ($8000 \times g$ for 10 min at 4 °C), washed with pre-chilled acetone, air dried, and re-suspended in 100 μL TEAB (100 mM). The proteins were digested with 1:40 (w/w) MS-grade Pierce Trypsin/Lys-C Protease Mix for 24 h at 37 °C[126].

*Phosphopeptide enrichment using titanium dioxide (TiO₂).* Protein digests were dried using SpeedVac and re-suspended in 1 mL loading buffer (80% acetonitrile (ACN), 5% TFA and 1 M glycolic acid)[127]. TiO₂ beads (0.6 mg per 100 μg peptide solution) were added, mixed well, and placed on a shaker for 15 min at room temperature. The solution was then table centrifuged to pellet the beads, which were subsequently mixed with 100 μL loading buffer, transferred to LoBind tube and table centrifuged. Thereafter, the beads were washed first with 100 μL wash buffer 1 (80% ACN, 1% TFA) followed by 100 μL wash buffer 2 (20% ACN, 0.2% TFA). The bead pellet, collected by table centrifugation, was dried using SpeedVac, mixed with 50 μL alkaline solution (28% NH₃ in H₂O, Thermo Fisher Scientific), and incubated for 15 min at room temperature for phosphopeptide elution. The solution was table centrifuged and the eluate was passed over a C18 Tip to recover the phosphopeptides. Finally, the C18 tip was washed with 20 μL of 30% ACN solution, collected in the same tube, and dried using SpeedVac.

*TMT labeling.* The phosphopeptide enriched digests were re-suspended in 100 μL TEAB (100 mM). A labeling kit was used according to the manufacturer's protocol for TMT labeling[128]. TMT10plex amine-reactive reagents (0.8 mg per vial) were re-suspended in 41 μL anhydrous ACN and all 41 μL of each reagent was added to each sample and mixed briefly. Reactions were allowed to proceed at room temperature for 1 h, which were then quenched by addition of 8 μL of 5% hydroxylamine for 15 min. Subsequently, the samples were combined in equal amount (1:1), transferred to a LoBind tube, and dried using SpeedVac. The TMT-labeled peptide mixtures were desalted on an Isolute MFC18 cartridge (100 mg/3 mL; Biotage, Charlotte, NC). Eluted peptides were fractionated into 5 equal aliquots, dried, and re-suspended in 20 μL of 0.1% formic acid solution prior to liquid chromatography with tandem mass spectrometry (LC-MS-MS).

*LC-MS/MS of TMT-labeled peptides using $MS^2$.* Liquid chromatography was performed on a fully automated EASY-nLC 1200 (Thermo Fisher Scientific) fitted with a C18 column (PepMap RSLC, 75 μm inner diameter, 15 cm length; Thermo Fisher Scientific), which was kept at a constant temperature of 40 °C. Mobile phases consisted of 0.1% formic acid for solvent A and 0.1% formic acid in ACN for solvent B. Samples were loaded in solvent A, and a linear gradient was set as follows: 0–5% B for 5 min, followed by a gradient up to 30% B in 50 min and to 60% B in 65 min. A 10-min wash at 95% B was used to prevent carryover, and a

15-min equilibration with 0% B completed the gradient. The liquid chromatography system was coupled to a Q Exactive HF Hybrid Quadrupole-Orbitrap (Thermo Fisher Scientific) equipped with an Easy-Spray Ion Source and operated in positive ion mode. The spray voltage was set to 1.7 kV, S-lens RF level to 35, and ion transfer tube to 275 °C. The full scans were acquired in the Orbitrap mass analyzer that covered an m/z range of 350–1500 at a resolution of 120,000. The automatic gain control (AGC) target was set to 3E6 and maximum ion time to 50 ms. The $MS^2$ analysis was performed under data-dependent mode to fragment the top 15 most intense precursors using higher-energy collisional dissociation (HCD) fragmentation. The $MS^2$ parameters were set as follows: resolution, 60,000; AGC target, 1E5; minimum AGC target, 8.0E3; intensity threshold, 8.0E4; maximum ion time, 100 ms; isolation width, 1.2 m/z; precursor charge state, 2–7; peptide match, preferred; dynamic exclusion, 30 s; and fixed first mass, 100 m/z. The normalized collision energy for HCD was set to 32%.

*Relative protein quantification using TMT.* Relative protein abundances in the different treatment groups were determined using PEAKS Studio proteomics software (version 10.0, build 20190129; https://www.bioinfor.com//) with the following settings[129]: enzyme, trypsin; instrument, Orbitrap; fragmentation, HCD; acquisition, data-dependent acquisition (DDA) without merging the scans; and precursor and fragment mass tolerance, 20 ppm and 0.5 Da, respectively. The correct precursor was detected using mass only. All peptide identifications were performed within PEAKS using the UniProtKB_TrEMBL database with human taxonomy (https://www.uniprot.org/uniprot). PEAKS PTM search tool was used to find peptides with fixed (carbamidomethylation and TMT10plex) and variable (phosphorylation (STY), oxidation (M) and deamidation (NQ)) modifications. SPIDER search tool was used to find novel peptides that were homologous to peptides in the protein database. The maximum number of variable posttranslational modifications per peptide was 5, and the number of de novo dependencies was 18. The de novo score threshold for SPIDER was 15, and the peptide hit score threshold was 30. FDR threshold was set to 1%. Reporter ion quantification with TMT-10plex (CID/HCD) was used. PEAKS was allowed to autodetect the reference sample and automatically align the sample runs. Reporter ion intensity ≥1E4, significance ≥16, fold change ≥1 and 2 unique peptides per protein were used to filter the quantified protein lists.

Further subgroup analysis of differentially expressed phosphoproteins for ADH-6 vs ADH-1 and ReACp53 vs ADH-1 was done by t-test for each group, and only those proteins that differed significantly (P < 0.05) were considered subsequently. Fold change of protein expression was obtained from the averaged intensity value for each protein in ADH-6/ADH-1 or ReACp53/ADH-1.

*Principal component and pathway analyses.* PCA was performed to analyze the main source of variation within and amongst the different treatment groups. Raw counts were used for differently abundant proteins and were plotted using the ClustVis application[130]. GSEA was performed for the "compute overlap function" using MSigDB to identify hallmark signatures. For this analysis, the protein accession IDs were converted to gene symbols using David Gene ID conversion tool[131]. Significance was assessed by q-value (i.e. FDR-adjusted P-value)[122], and the data were ordered based on −log10(q-values).

*Plots and network plots.* Heatmaps showing differentially expressed phosphoproteins were generated in R using the Pearson correlation analysis[132]. GO-Chord plot showing the pathways and the corresponding proteins that are part of each pathway/gene signature was also generated in R. Biological roles of the phosphoproteins in cancer were further inferred from published data (Supplementary Table 2). Upregulated phosphoproteins were depicted using the PPI network map generated via the search tool for retrieval of interacting genes (STRING)[133].

## In vivo tumor inhibition studies

All animal experiments were approved by the NYU Abu Dhabi Institutional Animal Care and Use Committee (NYUAD-IACUC; Protocol No. 18-0001), and were carried out in accordance with the Guide for Care and Use of Laboratory Animals[134]. Athymic nude NU/J mice (Foxn1$^{nu}$) were purchased from The Jackson Laboratory (Bar Harbor, ME) and housed at the NYU Abu Dhabi Vivarium Facility. Mice were maintained in air-filtered cages with controlled temperature (20 °C) and humidity (50%), in a 12 h light/dark cycle, and fed standard mouse chow (Research Diets; New Brunswick, NJ).

For pharmacokinetics, tumor-bearing mice (tumor volume ~25 mm³) were randomly assigned to saline and ADH-6 treatment groups. A single dose of saline or ADH-6 (15 mg kg⁻¹) was administered via IP injection. Mice (n = 5–6 per timepoint) were sacrificed at 0.5, 1, 2, 4, 8, 16, 24, 48, and 72 h following injection. The blood was collected via terminal cardiac puncture using K3-EDTA as an anticoagulant under isoflurane anesthesia and processed for plasma by centrifugation ($1500 \times g$ for 5 min). Plasma and tissues were placed in cryopreservation vials, preserved by snap freezing using liquid N₂, and stored at −80 °C until analysis by LC-MS/MS, which was performed according to a published protocol[135]. ADH-6 stock solutions (1.0 mg/mL) were prepared in acetonitrile (ACN). The matrix for the standard curve and quality controls (QC) consisted of control mouse plasma for all plasma samples, or control tissue homogenate for the tissue being analyzed. ADH-6 was extracted from 50 μL standard, QC, or unknown sample by protein precipitation with 200 μL ACN/0.1%

formic acid containing 20 ng/mL ADH-6 internal standard. Samples were vortexed for 5 min, then centrifuged at $5000 \times g$ for 10 min at 4 °C. Subsequently, 150 μL supernatant was transferred to a clean 1.5 mL tube, lyophilized under $N_2$, and reconstituted in 60 μL ACN/0.1% formic acid. Fifty microliters of this was transferred to a salinized glass 96-well plate insert containing 50 μL ddH2O and 10 μL of the sample was injected for analysis by LC-MS/MS.

For the single (mutant p53-harboring) xenograft inhibition studies, $5 \times 10^5$ viable MIA PaCa-2 or SK-BR-3 cancer cells were injected subcutaneously into the right flank of each mouse at age 6–8 weeks. For the dual (WT and mutant p53-bearing) xenograft model, $5 \times 10^5$ viable MIA PaCa-2 and MCF-7 cancer cells were injected into the right and left flanks, respectively, of each mouse at age 6–8 weeks. Mice were assessed daily for overt signs of toxicity. Tumor volume was determined by high-precision calipers (Thermo Fisher Scientific) using the following formula:

$$\text{tumor volume } (\text{mm}^3) = \frac{W^2 \times L}{2} \quad (3)$$

where $W$ and $L$ are tumor width and length in mm, respectively[136]. Mice were euthanized once tumor volume approached burden defined by NYUAD-IACUC.

Once the tumor volume reached ~25 mm³, the mice were randomized into four treatment groups ($n = 8$ per group), which were injected intraperitoneally with: vehicle or the indicated concentrations of ADH-1, ReACp53, or ADH-6. Injections were done every 2 days for a total of 12 doses, with the first day of treatment defined as day 0. Body weight and tumor volume were recorded for the duration of treatment, and survival ($n = 4$ per group) was monitored for a total of 30–60 days. At the end of treatment, four mice per treatment group were sacrificed and the tumor tissues were isolated to determine the tumor mass.

Isolated tumors and vital organs were formalin-fixed and paraffin-embedded and sectioned into 7-μm slices. For hematoxylin and eosin (H&E) analysis, the tissue sections were dewaxed and stained using standard procedures[73]. For IHC analysis, the samples were prepared according to a published protocol[14]. The tissue sections were treated by heat-induced epitope retrieval in 10 mM citrate buffer (pH 6.0) for antigen recovery, blocked with 8% BSA, then incubated overnight at 4 °C in primary antibodies (1:400 DO-7 or 5 μg/mL PAb 240). This was followed by sequential 45 min incubations at room temperature in biotinylated secondary antibody and streptavidin-horseradish peroxidase (Abcam). Finally, the signal was visualized upon incubation with the 3,3'-diaminobenzidine (DAB; Thermo Fisher Scientific) substrate. The tissue sections were imaged on a NIKON LV100 upright microscope and processed using the ECLIPSE LV software.

**Statistics and reproducibility**. For in vitro studies, investigators were blinded for all parts of the experiments (treatment, data acquisition, and data analysis), and a different investigator carried out each part. For in vivo studies, power calculation was used to select sample sizes from the NYU Abu Dhabi Institutional Animal Care and Use Committee (NYUAD-IACUC) Protocol (Protocol No. 18-0001), and investigators were blinded for data acquisition and data analysis. Unless otherwise indicated, error bars represent the mean ± standard deviation of at least three biological replicates (i.e. $n \geq 3$). Statistical analysis was performed using Prism (version 8.4.2; GraphPad Software Inc., La Jolla, CA, USA).

**Reporting summary**. Further information on research design is available in the Nature Research Reporting Summary linked to this article.

## Data availability

RNA-Seq data have been deposited in the Gene Expression Omnibus (GEO) repository (accession code: GSE161952) at the National Center for Biotechnology Information (NCBI). Proteomics data have been deposited in the MassIVE repository (accession code: MSV000086563; https://doi.org/10.25345/C5X78B) at the ProteomeXchange Consortium. The following publicly available databases were used in the study: IARC TP53 database (https://p53.iarc.fr), Database for Annotation, Visualization Integrated Discovery (DAVID; https://david.ncifcrf.gov/home.jsp), Molecular Signatures Database (MSigDB; http://www.gsea-msigdb.org/gsea/msigdb/index.jsp), and UniProtKB_TrEMBL database (https://www.uniprot.org/uniprot). All the datasets generated and/or analyzed during the current study are also available from the corresponding authors on reasonable request. Source data are provided with this paper.

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

## Acknowledgements

The authors thank Dr. Priyatansh Gurha (The University of Texas Health Science Center at Houston) for assistance with the transcriptome and phosphoproteome analyses, Khulood Alawadi (Assistant Director, Research Visualization, Design, and Manu-facturing, NYU Abu Dhabi) for preparing the graphic illustrations, and Mohayed Magzoub (MEng MSc CEng MIMarEST RN) for critical reading of the manuscript. The authors also thank the NYU Abu Dhabi Center for Genomics and Systems Biology (NYUAD-CGSB) for use of their BD FACSAria III for flow cytometry measurements and Illumina NextSeq 550 System for RNA-Seq. Confocal fluorescence imaging, NMR, quantitative proteomics and TEM experiments were carried out using the Core Tech-nology Platforms (CTP) resources at NYU Abu Dhabi. RNA-Seq data processing was done using the high performance computing (HPC) resources at NYU Abu Dhabi. This work was supported by funding from NYU Abu Dhabi and an ADEK Award for Research Excellence grant (AARE17-089) to M.M., from NYU to A.D.H., and from the University of Denver to Sunil Kumar.

## Author contributions

M.M. conceived and planned the project in consultation with A.D.H. and S. Kumar. M.M., A.D.H. and S. Kumar supervised the project and provided funding. S. Kumar and D.M. synthesized the oligopyridylamides. M.M. and L.P. designed the aggregation, intracellular imaging, cell viability/toxicity, cell cycle distribution, and related experiments, and L.P., L.K., S.H., M.K., R.P., and S. Karapetyan conducted these experiments. S. Kumar designed the peptide oligomerization and binding experiments, and J.A. conducted these experiments. S. Kumar and L.P. designed and conducted the CD experiments. G.E., Y.H., Z.F., and R.S. designed and conducted the structural studies. M.M., L.P., and L.K. designed the CETSA experiments, and L.P. and L.K. conducted these experiments. M.A.-S., L.K., I.C., L.P., and M.M. designed the transfection, RNA-Seq and ChIP-qPCR experiments; M.A.-S., L.K., I.C., L.P., and T.H. conducted these experiments, and M.A.-S., L.K., I.C., L.P., and A.J.A. performed the analyses. A.J.A., L.A., L.P., and M.M. designed the quantitative proteomics experiments; L.A. and L.P. conducted these experiments, and A.J.A., M.A., and L.A. performed the analysis. A.J.A. and M.A. designed the plant experiments, and M.A. and L.P. conducted these experiments. M.M. and L.P. designed the in vivo tumor studies, and L.P. conducted these studies. M.M. and L.P. wrote the manuscript with contributions from (in alphabetical order) A.J.A., G.E., I.C., L.A., L.K., M.A., M.A.-S., R.S., S.H., S. Kumar, and Z.F. All authors subsequently reviewed and edited the manuscript.

## Competing interests

New York University and New York University Abu Dhabi hold a patent for the use of oligopyridylamides to inhibit mutant p53 amyloid formation, with S. Kumar, A.D.H., M.M. and S.H. listed as inventors (US10500197B2).
