## [Peer Review File · Nature Communications]

Reviewers' Comments:

Reviewer #1:

Remarks to the Author:

In this manuscript, Palanikumar and co-authors identified that ADH-6, a tripyridylamide, can inhibit mutant p53 aggregation by binding to the aggregation-nucleating subdomain of mutant p53 DNA-binding domain. By doing so, ADH-6 dissociates mutant p53 aggregation and reactivates the transactivation activity of mutant p53 (p53R248W) in pancreatic cancer cell, resulting in cell cycle arrest and apoptosis. Furthermore, they showed that ADH-6 suppressed xenograft tumor growth derived from cancer cells with p53R248W. Overall, this is an important finding demonstrating the potential ant-tumor efficacy for the ADH-6. I have the following concerns:

1. As R248W is a contact mutation, how inhibiting the aggregation-nucleating subdomain to block p53R248W aggregation restore the transactivation activity is mechanistically unclear. Can ADH-6 restore p53R248W DNA-binding activity? Can ADH-6 restore the function of other p53 contact mutants or structural mutants? ChIP assay or ChIPseq should be performed. As only p53R248W was examined here, additional p53 mutants should be examined.
2. The authors should at least test the protein expression of some of the well-know p53 target genes such as p21, bax and mdm2 etc.
3. The direct binding of ADH-6 to p53 aggregation-nucleating subdomain should be further confirmed in vitro.
4. Only once cell line (Miapaca2) was examined, the authors should test the efficacy of ADH-6 in other mutant p53 containing cell lines.
5. Figure 2: the score changes between LTIITLE and LTRITLE peptides were subtle. Statistics analysis may be helpful.
6. The proteomics analysis of phosphoprotein (Fig. 7) is interesting. However, how the study is connected to p53 reactivation is not clear.

Reviewer #2:

Remarks to the Author:

This manuscript by Palanikumar and coauthors, titled "Protein mimetic amyloid inhibitor potently abrogates

cancer-associated mutant p53 aggregation and restores

tumor suppressor function" reports the study of oligopyridylamides as molecules able to inhibit the aggregation of mutant p53 proteins in cancer cells and to cause apoptotic cell death.

The study is very well described and developed.

The data highlight the potential of the identified compound to reactivate p53 activity in mutant p53-carrying cells without showing toxicity in cells bearing wt-p53 or in p53-null systems.

A few concerns are listed below to improve the strength of the study.

Major:

Figures 6-7: Authors interestingly show that treatment with ADH-6, but not ADH-1, not only impair the formation of aggregates by mutant p53 proteins but also reactivate p53 function. It is surprising and very relevant that mutant p53 proteins acquire again the ability of wt-p53 to transactivate its bona fide targets. Is p53 really regulating these genes or is this an indirect effect of other p53 family members?

To confer additional strength to these findings, the authors should analyze the recruitment of mutant p53 and other p53 family members on the p53 binding sites of target genes (ChIP), thus

formally demonstrating the eventual reacquired transcriptional activity of mutant p53.

Authors list a number of cell lines as having wt-p53. To my knowledge SW480 and SW1990 cell lines carry mutated TP53, with mutants p.R273H and p.P309S reported in SW480, and mutant p.P191del in SW1990 (as from IARC database). Did the authors check the mutational status of TP53 coding exons in the cell lines used in their study? This surely needs to be done.

Minor:

Figures 6A and 6D: Please explain better the different analyses from which these expression matrices were generated.

Did the authors compare the gene expression changes obtained with ADH-6 treatment with those observed previously with ReACp53 in ovarian cancer?

The description of the various functions of the genes deregulated by ADH-6 treatment could be moved from the results to the discussion.

Reviewer #3:

Remarks to the Author:

This is a comprehensive paper describing the synthesis of a new compound, a tripyridylamide, ADH-6, that binds to mutant p53 causing it to reduce its state of aggregation and be reactivated as a tumor suppressor. The compound induces tumor regression only in p53 wild type tumors in vivo and does not act on p53 wild type cancers. The work builds on earlier studies on a peptide known as ReACp53 that is reported to act by a similar mechanism of blocking and reversing aggregation. The work here though represents a significant advance because the ADH-6 compound is much more stable and potent than ReACp53 because of its chemical nature. On the face of it then we have a remarkable new medicine that can correct the effects of mutations in p53 and show highly selective tumor cell killing. Of especial importance is the demonstration that expression of the mutant p53 allele in cells that do not normally express the protein (SKOV3 cells) makes them sensitive to ADH-6 Figure 5c,d. This experiment is the strongest direct proof of the proposed mechanism in the manuscript. Given the huge number of screens done in the past to obtain such compounds and their comprehensive failure to be validated which has usually revealed other mechanisms of action the authors do need to explore every avenue of validation of "on target" activity. Extension of the SKOV 3 study using other null cell lines, other mutants and inducible expression systems would add additional weight to the data.

One of the most powerful approaches is the use of large cell line panels as the existing limited data look so promising an extended analysis of many more cell lines would be very convincing. This has been done for example for the small molecule MDM2 inhibitors where an almost perfect correlation between p53 wild type status and drug sensitivity has emerged. As a second approach a CRISPR knockout screen can also be very helpful as this has recently demonstrated that many small molecule drugs are off target (Lin et al., *Sci. Transl. Med.* 11, eaaw8412 (2019)).

Finally a CETSA analysis can be used to establish target engagement (*Scientific Reports* | 7: 13000 | DOI:10.1038/s41598-017-12513-1).

. These additional studies will be needed frankly to convince a skeptical community and justify publication in this journal. If these further validations are successful then this paper would represent a great breakthrough in the field and generate enormous enthusiasm.

Response to Reviewer #1

In this manuscript, Palanikumar and co-authors identified that ADH-6, a tripyridylamide, can inhibit mutant p53 aggregation by binding to the aggregation-nucleating subdomain of mutant p53 DNA-binding domain. By doing so, ADH-6 dissociates mutant p53 aggregation and reactivates the transactivation activity of mutant p53 (p53R248W) in pancreatic cancer cell, resulting in cell cycle arrest and apoptosis. Furthermore, they showed that ADH-6 suppressed xenograft tumor growth derived from cancer cells with p53R248W. Overall, this is an important finding demonstrating the potential ant-tumor efficacy for the ADH-6.

Response: We thank the Reviewer for the positive comments regarding the manuscript. We are grateful for the feedback provided and have worked diligently to address all of the concerns. In the following pages, we respond point-by-point to the Reviewer's comments. We believe, and hope the Reviewer agrees, that the paper is much improved in content and clarity.

I have the following concerns:

Concern #1. As R248W is a contact mutation, how inhibiting the aggregation-nucleating subdomain to block p53R248W aggregation restore the transactivation activity is mechanistically unclear. Can ADH-6 restore p53R248W DNA-binding activity? Can ADH-6 restore the function of other p53 contact mutants or structural mutants? ChIP assay or ChIPseq should be performed. As only p53R248W was examined here, additional p53 mutants should be examined.

Response: As the Reviewer points out, R248W is generally classed as a p53 'contact' mutation. However, increasing evidence indicates that the traditional classification of p53 mutations as either 'contact' (mutations that affect DNA-interacting amino acids, thereby hindering wild-type transcriptional activity, but without dramatically affecting the conformation of the protein) or 'structural'/'conformational' (mutations that disrupt the three-dimensional structure of the protein) may be an oversimplification, as several p53 mutants appear to possess both characteristics (Muller and Vousden 2014; Joerger and Fersht 2016). In other words, various so-called p53 'contact' mutations cause alterations in the structure and conformational stability of the protein (Joerger and Fersht 2016). Destabilization of the protein's structure, in turn, often leads to exposure of the DNA-binding domain (DBD)'s hydrophobic core and self-assembly of the protein into amyloid-like aggregates within inactive cellular inclusions (Xu et al. 2011; Soragni et al. 2016). Indeed, this appears to be the case for R248W and the closely related R248Q mutation (Wong et al. 1999; Song, Hollstein, and Xu 2007), which has led to these mutants sometimes being designated as 'structural' mutants (Muller and Vousden 2014), despite their "official status" as 'contact' mutants.

Thus, in the case of several p53 'contact' mutants, including R248W and R248Q, loss of tumor suppressor function is likely due to not only alteration of a DNA-binding amino acid, but also destabilization of the protein's structure and its sequestration in inactive cytosolic amyloid-like aggregates. This suggests that disaggregating these mutants and stabilizing their three-dimensional structure would rescue, at least in part, their function. This hypothesis is strongly supported by the study by Soragni *et al.*, in which they demonstrate that mutant R248Q p53 forms amyloid-like aggregates, and that blocking this aggregation, by masking the aggregation-prone core with a sequence-specific peptide inhibitor (denoted ReACp53), restores the mutant protein to a WT p53-like structure and functionality (Soragni et al. 2016). Similarly, we show that ADH-6 effectively targets and inhibits mutant R248W

p53’s amyloid aggregation, thereby strongly shifting the folding equilibrium towards the functional, WT-like state. The restoration of p53’s transcriptional activity leads to cell cycle arrest and apoptosis *in vitro* and reduction in tumor growth *in vivo*.

As suggested by the Reviewer, we have extended the work to incorporate more cell lines and other mutants. These additional experiments, which are outlined in our response to the Reviewer’s **Concerns #3 & #4**, confirm that the observed cytotoxicity of ADH-6 in cancer cells is mutant p53 dependent. Further, the results underline the generality of the effects of ADH-6 by demonstrating the efficacy of the oligopyridylamide at targeting a number of aggregation-prone p53 mutants *in vitro* and *in vivo*.

To address the Reviewer’s specific concern regarding the mechanism of ADH-6 mediated restoration of mutant R248W p53’s transactivation activity, we have performed chromatin immunoprecipitation (ChIP) coupled with quantitative PCR (ChIP-qPCR) to detect recruitment of mutant p53 to the WT protein’s binding sites on promoters/enhancers of target genes in oligopyridylamide treated cancer cells. Treatment of MIA Paca-2 cells with ADH-6 led to binding of R248W to p53 target genes, *CDKN1A* (also known as *P21*), *PIG3* and *NOXA*. Importantly, we did not detect significant binding of the p53 homologs, p63 and p73, to *CDKN1A* following treatment with ADH-6. Moreover, as presented in the response to the Reviewer’s **Concern #2**, western blot analysis confirmed elevated expression of p53 targets in ADH-6 treated MIA PaCa-2 cells. Thus, the ChIP-qPCR results strongly support our original conclusion that ADH-6 specifically targets and reactivates aggregation-prone mutant p53.

The primers used in the ChIP-qPCR experiments are listed in the new **Supplementary Table 1**:

Supplementary Table 1. Primers used for ChIP-qPCR analysis of ADH-6 mediated recruitment of mutant R248W p53, p63 and p73 to the p53 binding sites on promoters/enhancers of target genes

Supplementary Table 1a. Mutant R248W p53			
Gene	Primer	References	Sequence
CDKN1A (P21)	Forward (5’-3’)	(el-Deiry et al. 1993)	GTGGCTCTGATTGGCTTTCTG
	Reverse (3’-5’)		CTGAAAACAGGCAGCCCAAG
BAX	Forward (5’-3’)	(Miyashita and Reed 1995)	TAATCCCAGCGCTTTGGAA
	Reverse (3’-5’)		TGCAGAGACCTGGATCTAGCAA
PUMA	Forward (5’-3’)	(Nakano and Vousden 2001)	GCGAGACTGTGGCCTTGTGT
	Reverse (3’-5’)		CGTTCAGGGTCCACAAAGT
NOXA	Forward (5’-3’)	(Oda et al. 2000)	CAGCGTTTGCAGATGGTCAA
	Reverse (3’-5’)		CCCCGAAATTACTTCCTTACAAAA
PIG3	Forward (5’-3’)	(Kaesler and Iggo 2002)	CACTCCCAACGCCTCCTTT
	Reverse (3’-5’)		GCCCATCTTGAGCATGGGT
MDM2	Forward (5’-3’)	(Wu et al. 1993)	GTTGACTCAGCTTTCTCTTG
	Reverse (3’-5’)		GGAAAATGCATGGTTTAAATAGCC
GADD45	Forward (5’-3’)	(Kastan et al. 1992)	AGCGGAAGAGATCCCTGTGA
	Reverse (3’-5’)		CGGGAGGCAGGCAGATG

Supplementary Table 1b. p63			
Gene	Primer	References	Sequence
CDKN1A (P21)	Forward (5’-3’)	(Kouwenhoven et al. 2010)	GTGGCTCTGATTGGCTTTCTG
	Reverse (3’-5’)		CTGAAAACAGGCAGCCCAAG

Supplementary Table 1c. p73			
Gene	Primer	References	Sequence
CDKN1A (P21)	Forward (5’-3’)	(Koeppel et al. 2011)	GTGGCTCTGATTGGCTTTCT
	Reverse (3’-5’)		AGCCTCTTCTATGCCAGAGC

The results of the ChIP-qPCR experiments are presented in the new **Supplementary Figure 14**:

Supplementary Figure 14. ChIP-qPCR analysis of recruitment of mutant R248W p53 to the WT protein's binding sites on promoters/enhancers of target genes. Binding of mutant R248W p53 to the WT protein's transcriptional targets, *CDKN1A* (*P21*) (a), *PIG3* (b) and *NOXA* (c), in MIA PaCa-2 cells treated with vehicle or 5 μ M ADH-6 for 24 h. For comparison, binding of p63 and p73 to *CDKN1A* in ADH-6 treated MIA PaCa-2 cells is also presented (a). * $P < 0.05$, ** $P < 0.01$, *** $P < 0.001$ or non-significant (ns, $P > 0.05$) for comparisons with vehicle-treated controls.

To address the Reviewer's concern, we have also expanded the relevant sections of the **Revised Manuscript** and **Revised Supplementary Material**, which now read as follows:

RESULTS; ADH-6 induces transcriptional reactivation of p53:

“To confirm that ADH-6 rescues p53 function, we first used ChIP-qPCR to detect recruitment of mutant R248W p53 to the WT protein's binding sites on promoters/enhancers of target genes in oligopyridylamide treated MIA PaCa-2 cells (primers used are listed in Supplementary Table 1). Treatment with ADH-6 led to binding of R248W to *CDKN1A* (also known as *P21*) (Supplementary Figure 14a). Importantly, we did not detect significant binding of the p53 homologs, p63 and p73, to *CDKN1A* under the same experimental conditions (Supplementary Figure 14a), supporting the notion that ADH-6 activity is mutant p53 dependent. Interaction of R248W with other well-established primary p53 transcriptional targets, *PIG3* and *NOXA*, was also observed in ADH-6 treated MIA PaCa-2 cells (Supplementary Figure 14b,c). (Please note that functions of the genes referenced in this section are described in Supplementary Section 4.)”

SUPPLEMENTARY SECTION 4.1.; Chromatin immunoprecipitation (ChIP) with quantitative real-time PCR (ChIP-qPCR) assay:

“ChIP-qPCR (primers used are listed in Supplementary Table 1) revealed that treatment of MIA PaCa-2 cells with ADH-6 resulted in binding of mutant R248W p53 to the WT protein's binding sites on promoters/enhancers of target genes, *CDKN1A* (also known as *P21*), *PIG3* and *NOXA* (Supplementary Figure 14). p21, a cyclin-dependent kinase inhibitor, is required for p53-mediated cell cycle arrest in response to DNA damage and other cellular stresses (Abbas and Dutta 2009; Karimian, Ahmadi, and Yousefi 2016). *PIG3* (p53-induced gene 3) is involved in both the early cellular response to DNA damage and p53-induced apoptosis (Polyak et al. 1997; Lee et al. 2010). *NOXA* (Latin for damage) encodes a member of the Bcl-2 family of apoptosis

regulator proteins; upon p53-induced expression, Noxa localizes to mitochondria, where the protein binds to and inhibits anti-apoptotic Bcl-2 family members (Aubrey et al. 2018). Notably, Noxa has also been identified as a key determinant of cytotoxic response of cancer cells to standard chemotherapy and targeted cancer therapy (Albert, Brinkmann, and Kashkar 2014; Montero et al. 2019).”

DISCUSSION:

“Further analysis of the effects of ADH-6 on mutant p53 harboring cancer cells revealed that the oligopyridylamide induces cell cycle arrest and apoptosis, both of which are indicators of restored p53 function (Figure 4 and Supplementary Figure 11). ADH-6-mediated restoration of WT-like activity of mutant p53 was confirmed using a number of complementary assays. ChIP-qPCR analysis revealed that ADH-6 treatment resulted in recruitment of mutant p53 to the WT protein’s primary transcriptional targets, such as *CDKN1A*, *PIG3* and *NOXA* (Supplementary Figure 14 and Supplementary Table 1), that are not only key mediators of p53-dependent cell cycle arrest and apoptosis (Abbas and Dutta 2009; Karimian, Ahmadi, and Yousefi 2016; Polyak et al. 1997; Lee et al. 2010; Oda et al. 2000), but also determinants of the cytotoxic response of cancer cells to standard chemotherapy and targeted cancer therapy (Albert, Brinkmann, and Kashkar 2014; Montero et al. 2019). Transcriptional reactivation of mutant p53 was further corroborated using RNA-Seq (Figure 5 and Supplementary Figures 16,17). Specifically, we observed activation of the *TP53* and apoptosis pathways, and suppression of genes involved in cell cycle progression (G2/M) and the *MYC* pathway, which are known to be negatively regulated by p53 (Sachdeva et al. 2009).”

MATERIALS AND METHODS; Chromatin immunoprecipitation (ChIP) coupled with quantitative PCR (ChIP-qPCR):

“MIA PaCa-2 cells were cultured in complete DMEM in T175 flasks (Corning) until 70–80% confluence was reached. Thereafter, the medium was replaced with serum-free DMEM containing vehicle or 5 μ M ADH-6, and the cells were incubated for a further 24 h. The cells were then treated with 1% PFA for 10 min at 37 $^{\circ}$ C for crosslinking, followed by a single wash with 5 mL PBS. 1.5 mL buffer A (5 mM PIPES, pH 8, 85 mM KCl, 0.5% NP40, Protease Inhibitor Cocktail) was added to the cells, which were harvested using a pipette tip and scraper. The cell suspension was rotated for 10 min at 4 $^{\circ}$ C, centrifuged (3,700 rpm for 5 min at 4 $^{\circ}$ C), and the supernatant was discarded. To the cell pellet, 400 μ L buffer B (1% SDS, 10 mM EDTA, 50 mM Tris-HCL, pH 8.1) was added, and the cell suspension was sonicated (10 cycles of 5 s each at maximum power, with 2 min on ice between cycles to prevent overheating). Cell debris was removed by centrifugation (13,000 rpm for 10 min at 4 $^{\circ}$ C), and the supernatant was collected. The samples were diluted tenfold with the immunoprecipitation buffer (0.01% SDS, 1.1% Triton X-100, 1.2 mM EDTA, 16.7 mM Tris-HCL, pH 8.1), rotated for 30 min at room temperature, followed by centrifugation (1,500 rpm for 20 s at 4 $^{\circ}$ C). Of the collected supernatant, 100 μ L was used as input for each sample, with the remainder incubated with 1 μ g/mL of the antibody for p53 (DO-1; Santa Cruz), p63 (D2K8X; Cell Signaling Technology, Danvers, MA) or p73 (EPR19884; Abcam), overnight on a rotator at 4 $^{\circ}$ C. 50 μ L magnetic beads (Magna ChIP protein A+G bead blend; Sigma) was added to each sample solution and rotated for 30 min at room temperature. After rotation, the samples were incubated for 30 s on a magnetic stand to separate the chromatin, which was then washed twice with wash buffer 1 (0.1% SDS, 1% TritonX-100, 2 mM EDTA, 20 mM Tris-HCL, pH 8.1, 150 mM NaCl), followed by twice with wash buffer 2 (0.1% SDS, 1% TritonX-100, 2 mM EDTA, 20 mM Tris-HCL, pH 8.1, 300 mM NaCl), twice with wash buffer 3 (0.25 M LiCl, 1% NP40, 1% sodium deoxycholate, 1 mM EDTA, 10 mM Tris-HCL, pH 8.1), and finally once with TE buffer (10 mM Tris-HCL, pH 8.1, 1 mM EDTA).

Immediately after the washes, the samples were eluted by adding the elution buffer (0.5% SDS, 5 mM EDTA, 25 mM Tris-HCL, pH 7.5, 20 mM NaCl), followed by vortexing and incubation for 15 min at 65 °C. Subsequently, the samples were incubated on a magnetic stand and 200 μ L supernatant was collected, to which 200 μ L immunoprecipitation buffer was added, followed by 1 μ L Protease Inhibitor Cocktail, and the samples were incubated overnight at 65 °C. DNA was recovered with a conventional in-house DNA precipitation method using sodium acetate, with samples eluted in volumes of 20 μ L. Finally, the recovered DNA was amplified and quantified (for qPCR reaction: 1 μ L DNA from ChIP samples or input, 1 μ L 2 mM primers (Supplementary Table 1), 5 μ L SYBr Green PCR Master Mix (Thermo Fisher Scientific), and 3 μ L nuclease-free water). PCR thermal reaction cycle included one cycle of 10 min at 95 °C, and 39 cycles of 15 s at 95 °C followed by 45 s at 60 °C. Normalization of PCR reaction was based on calculating the enrichment by comparing the threshold cycle number (C_T) for the input and ChIP samples.”

Concern #2. The authors should at least test the protein expression of some of the well-know p53 target genes such as p21, bax and mdm2 etc.

Response: As per the Reviewer’s recommendation, we have tested the effects of ADH-6 on expression of well-established p53 targets – p21, Noxa, MDM2 and Bax – in mutant R248W p53 bearing pancreatic cancer MIA PaCa-2 cells. Western blot analysis revealed markedly elevated expression of all four p53 targets in ADH-6 treated MIA PaCa-2 cells compared to both the vehicle and ADH-1 treatment groups. Of relevance, the mutant p53 disaggregating peptide ReACp53 was also reported to upregulate p21, Noxa, MDM2 and Bax in ovarian cancer cells that harbor another aggregating p53 mutant, R248Q (Soragni et al. 2016). These results further support our original conclusion that ADH-6 restores p53 function in aggregation-prone mutant p53-bearing cancer cells.

The results of the control experiment requested by the Reviewer are presented in the new **Supplementary Figure 15**:

Supplementary Figure 15. Western blot analysis of expression of direct p53 targets in oligopyridylamide-treated MIA PaCa-2 cells. (a) Immunoblots of p21, Noxa, MDM2 and Bax in MIA PaCa-2 cells treated with vehicle or 5 μ M ADH-1 or ADH-6 for 24 h. (b) Densitometric quantification of the immunoblot bands of the p53 targets (n = 3) (a). * $P < 0.05$, ** $P < 0.01$, *** $P < 0.001$ or non-significant (ns, $P > 0.05$) for comparisons with controls.

In addressing the Reviewer’s **Concern #2**, we have expanded the relevant sections of the **Revised Manuscript** as follows:

RESULTS; ADH-6 induces transcriptional reactivation of p53:

“In agreement with the ChIP-qPCR results, western blot analysis revealed elevated expression of both p21 and Noxa in ADH-6 treated MIA PaCa-2 cells (Supplementary Figure 15). Interestingly,

we also observed significantly higher expression of p53-inducible MDM2 and proapoptotic Bax in response to ADH-6 treatment (Supplementary Figure 15). Surprisingly, recruitment to the *MDM2* and *BAX* genes was not observed by ChIP-qPCR, which may be a consequence of the interaction of mutant p53 with the WT protein's binding sites on the promoters of these genes being too transient or weak to be detected by the assay (Kaeser and Iggo 2002; Swift and Coruzzi 2017). Of relevance, the mutant p53 disaggregating peptide ReACp53 was also reported to upregulate p21, Noxa, MDM2 and Bax in ovarian cancer cells that harbor another aggregating p53 mutant, R248Q (Soragni et al. 2016). Taken together, these results indicate that ADH-6 specifically targets and reactivates aggregation-prone mutant p53.”

SUPPLEMENTARY SECTION 4.1.; Chromatin immunoprecipitation (ChIP) with quantitative real-time PCR (ChIP-qPCR) assay:

“The ChIP-qPCR results were confirmed using western blot, which showed elevated expression of both p21 and Noxa in ADH-6 treated MIA PaCa-2 cells (Supplementary Figure 15). Interestingly, we also observed significantly higher expression of p53-inducible MDM2 and Bax in response to ADH-6 treatment (Supplementary Figure 15). Bax is another member of the Bcl-2 family, which, once activated, induces permeabilization of the outer mitochondrial membrane, leading to release of cytochrome c and activation of the apoptosis initiator caspase-9 (Oda et al. 2000). Surprisingly, recruitment to the *MDM2* and *BAX* genes was not observed by ChIP-qPCR. This could be due to the interaction of mutant p53 with these genes being too transient or weak to be detected by the assay (Kaeser and Iggo 2002; Swift and Coruzzi 2017).”

DISCUSSION:

“Consistent with the transcriptome analysis, western blot quantification of protein levels in MIA PaCa-2 cells following treatment with ADH-6 revealed increased expression of direct p53 targets (Supplementary Figure 15). Additionally, phosphoproteome analysis showed that ADH-6 downregulates key cancer-promoting phosphoproteins that are known to be directly negatively regulated by p53 (Harris and Levine 2005; Giono and Manfredi 2006; Brady et al. 2011; J. Chen 2016; Williams and Schumacher 2016) (Figure 6, Supplementary Figures 18,19, and Supplementary Table 2). Taken together, our results clearly demonstrate that ADH-6 dissociates mutant p53 amyloid-like aggregates in cancer cells, and that the released protein is restored to a functional form, which elicits the observed inhibition of proliferation via cell cycle arrest and induction of apoptosis.”

MATERIALS AND METHODS; Cell fractionation and western blot analysis:

“Cell fractionation and western blot were done as previously described (Woldetsadik et al. 2017; Palanikumar, Al-Hosani, Kalmouni, Saleh, et al. 2020). Briefly, cells were washed with ice-cold PBS, harvested by trypsinization and centrifuged. The resulting cell pellet was lysed using 100 μ L lysis buffer (Bio-Rad), and lysis was confirmed by light microscopy. For separation of the insoluble (pellet) and soluble (supernatant) fractions, the whole cell lysate was centrifuged (20,000 \times g for 20 minutes at 4 $^{\circ}$ C), and the protein content of the two fractions was determined using the bicinchoninic acid (BCA) protein assay (Smith et al. 1985).

Samples (20 μ g/lane) were electrophoresed on a 4–12% sodium dodecyl sulfate (SDS)-polyacrylamide gel in a Mini-PROTEAN Tetra Cell (Bio-Rad) and subsequently transferred to a nitrocellulose membrane (Sartorius AG, Göttingen, Germany), which was blocked with 5% nonfat milk and then incubated with p53 (DO-7; Santa Cruz), p21 (EPR3993; Abcam), Noxa (114C307; Santa Cruz), Bax (E63; Abcam) or MDM2 (2A10; Abcam) antibodies (1:1000), or β -actin antibody (2A3, 1:2000; Santa Cruz), overnight at 4 $^{\circ}$ C. This was followed by incubation for 3 h at room temperature with secondary antibodies (horseradish peroxidase-conjugated anti-

mouse or anti-rabbit IgG, 1:10,000; Abcam). Finally, the samples were visualized using Clarity Western ECL Substrate (Bio-Rad) on a Bio-Rad Gel Doc XR+ Gel Documentation System. For densitometric quantification, the immunoblot bands were normalized to β -actin.”

Concern #3. The direct binding of ADH-6 to p53 aggregation-nucleating subdomain should be further confirmed in vitro.

Response: To address the Reviewer’s concern, binding of ADH-6 to mutant p53 was assessed using the cellular thermal shift assay (CETSA), which measures ligand-induced changes in thermal stability of target proteins (Molina et al. 2013; Savitski et al. 2014). The inherently unstable p53 has a melting temperature (T_m) of ~ 45 °C, and mutations destabilize the protein further and lower its T_m by 5–10 °C (Selivanova and Wiman 2007; Joerger and Fersht 2016). Consistent with these reports, the CETSA generated melting curve in human pancreatic cancer MIA PaCa-2 cells yielded T_m of 39.6 ± 1.5 °C for mutant R248W p53. Treatment of the cells with ADH-6, but not ADH-1, markedly increased the R248W T_m to 48.2 ± 0.9 °C. Likewise, CETSA generated melting curves in human breast cancer SK-BR-3 cells revealed that treatment with ADH-6, but not ADH-1, significantly increased T_m of mutant R175H p53 from 38.8 ± 1.4 °C to 45.1 ± 1.2 °C. Along with the confocal fluorescence microscopy imaging of colocalization of fluorescently-labeled ADH-6 and antibody-stained mutant p53, the CETSA results confirm that ADH-6 directly interacts with and stabilizes mutant p53, which shifts the folding equilibrium towards the soluble state, leading to dissociation of the protein’s inactive amyloid-like cytosolic aggregates.

The results of the new control experiments done to address the Reviewer’s concern are presented in the new **Supplementary Figure 10** and updated **Figure 3h,i**:

Supplementary Figure 10. Cellular thermal shift assay (CETSA) analysis of target engagement in MIA PaCa-2 and SK-BR-3 cells. (a) Immunoblots of mutant R248W (a) and R175H (b) p53 in the soluble fractions of MIA PaCa-2 and SK-BR-3 cells, respectively, following treatment with vehicle or 5 μ M ADH-1 or ADH-6 for 6 h and then heating to the indicated temperatures for 3 min. Mutant p53 was detected by the anti-p53 antibody DO-7. Densitometric quantification of the immunoblot bands was used to generate the melting curves shown in Figure 3h,i.

Figure 3. ADH-6 dissociates mutant p53 aggregates in cancer cells. (a) Confocal fluorescence microscopy images showing thioflavin S (ThS) staining of mutant p53 (R248W) aggregates in MIA PaCa-2 cells treated with vehicle (0.02% DMSO) or ADH-6 (5 μM) for 0.5 or 6 h. (b) Quantification of ThS-positive MIA PaCa-2 cells after treatment with vehicle or ADH-6. (c) Confocal fluorescence microscopy images of ThS and PAb 240 antibody staining of R248W aggregates in MIA PaCa-2 treated with vehicle or 5 μM ADH-6 for 0.5 or 6 h. (d-f) Quantification of Pab 240-positive MIA PaCa-2 cells after treatment with the indicated concentrations of ADH-1, ReAcP53 or ADH-6 for 0.5 or 6 h relative to controls (vehicle-treated cells). (g) Colocalization of FITC-labeled ADH-6 (ADH-6_{FITC}) with Pab 240-stained R248W aggregates following incubation with the oligopyridylamide (5 μM) for 0.5 or 6 h. Imaging experiments were performed in triplicate and representative images are shown. Colocalization was quantified using Pearson's correlation coefficient, *r*, which measures pixel-by-pixel covariance in the signal level of two images (Dunn, Kamocka, and McDonald 2011). Scale bar = 5 μm. For quantification, the number of positively stained cells in 3–5 different fields of view are expressed as % of the total number of cells (*n* = 4). (h,i) Cellular thermal shift assay (CETSA) analysis of intracellular target engagement. Melting curves for p53 mutants R248W (h) and R175H (i) in MIA PaCa-2 and SK-BR-3 cells, respectively, in the absence or presence of the oligopyridylamides (*n* = 3). **P* < 0.05, ***P* < 0.01, ****P* < 0.001 or non-significant (ns, *P* > 0.05) for comparisons with controls.

In addressing the Reviewer’s concern, the relevant sections of the **Revised Manuscript** have been updated as follows:

RESULTS; ADH-6 dissociates intracellular mutant p53 aggregates:

“In order to ascertain whether ADH-6-mediated dissociation of mutant R248W p53 aggregates is a result of direct interaction with the oligopyridylamide, PAb 240-stained MIA PaCa-2 cells

were treated with FITC-labeled ADH-6 (ADH-6_{FITC}) (Figure 3g). Initially, strong colocalization of ADH-6_{FITC} and PAb 240 was observed, indicating direct interaction of the oligopyridylamide with mutant p53 aggregates. Eventually, the extent of colocalization decreased as the PAb 240 staining was reduced due to disaggregation of mutant p53. Intracellular target engagement was further confirmed using the cellular thermal shift assay (CETSA), which measures ligand-induced changes in thermal stability of target proteins (Molina et al. 2013; Savitski et al. 2014). The labile WT p53 has a melting temperature (T_m) of ~ 45 °C, and mutations destabilize the protein further and lower its T_m by 5–10 °C (Selivanova and Wiman 2007; Joerger and Fersht 2016). Consistent with these reports, CETSA generated melting curves in MIA PaCa-2 and SK-BR-3 cells yielded T_m of 39.6 ± 1.5 and 38.8 ± 1.4 °C for p53 mutants R248W and R175H, respectively (Figure 3h,i and Supplementary Figure 10). Treatment of the cells with ADH-6, but not ADH-1, significantly increased T_m to 48.2 ± 0.9 and 45.1 ± 1.2 °C for R248W and R175H, respectively (Figure 3h,i and Supplementary Figure 10), indicating strong stabilization of aggregation-prone mutant p53 by the oligopyridylamide. Taken together, these results show that, similar to ReACp53 (Soragni et al. 2016), ADH-6 efficiently enters cells to directly interact with and stabilize mutant p53, which shifts the folding equilibrium towards the soluble state, leading to dissociation of the protein's inactive amyloid-like cytosolic aggregates.”

DISCUSSION:

“Here, we have extended our protein mimetic-based approach, previously developed to modulate various aberrant PPIs (Kumar and Hamilton 2017; Kumar et al. 2018; 2015), towards mutant p53 aggregation. Screening a focused library of oligopyridylamide-based α -helix mimetics, we identified a cationic tripyridylamide, ADH-6, that potently inhibited oligomerization and amyloid formation of pR248W, a mutant p53 DBD-derived peptide containing both the aggregation-nucleating sequence and the commonly occurring R248W mutation, by stabilizing the peptide's native conformation (Figure 1). It should be noted that ADH-6 only modestly antagonized A β amyloid formation (Kumar and Hamilton 2017), indicating a degree of specificity of the oligopyridylamide for mutant p53. Importantly, ADH-6 also dissociated pre-formed pR248W aggregates, and prevented further aggregation of the peptide (Figure 1). Subsequent studies in human pancreatic carcinoma MIA PaCa-2 cells harboring mutant R248W p53 (Figure 3 and Supplementary Figures 7,9), and *N. benthamiana* cells transfected with YFP-tagged WT and R248W p53 DBDs (Supplementary Figure 8), showed that ADH-6 effectively dissociates intracellular mutant p53 amyloid-like aggregates. Evidence that these effects are due to direct intracellular target engagement comes from the following experiments: (i) confocal microscopy, where extensive colocalization of fluorescently labeled ADH-6 with antibody-stained intracellular mutant p53 was observed; and (ii) CETSA, which demonstrated strong stabilization of p53 mutants in cancer cells by ADH-6 (Figure 3 and Supplementary Figure 10).”

MATERIALS AND METHODS; Cellular thermal shift assay (CETSA):

“CETSA experiments were performed according to a previously published protocol (Molina et al. 2013). Briefly, MIA PaCa-2 or SK-BR-3 cells were treated with vehicle or 5 μ M ADH-1 or ADH-6. After incubation for 6 h, the cells were washed with ice-cold PBS, harvested by trypsinization, centrifuged and resuspended in PBS supplemented with complete Protease Inhibitor Cocktail. Equal amounts of cell suspensions were aliquoted into 0.2 mL PCR microtubes and incubated for 30 min at room temperature. Subsequently, the cell suspension aliquots were heated individually at the indicated temperatures for 3 min (Ristretto Thermal Cycler; VWR, Darmstadt, Germany), followed by cooling for 3 min at on ice. Finally, the cells were lysed using 2 cycles of freeze-thawing, and the soluble fractions were isolated by centrifugation and analyzed

by SDS-polyacrylamide gel electrophoresis (SDS-PAGE) followed by western blot as described above. Fold-changes in the immunoblot band densities (normalized to the β -actin controls and using the lowest temperature condition as reference) were plotted as a function of temperature to generate mutant p53 melt curves for the different treatments. Curve fitting and calculation of the melting point, i.e. the temperature at which half of the protein amount has been denatured (T_m), were done using published procedures (Molina et al. 2013; Savitski et al. 2014).”

Concern #4. Only once cell line (Miapaca2) was examined, the authors should test the efficacy of ADH-6 in other mutant p53 containing cell lines.

Response: As per the Reviewer’s recommendation, we have tested the efficacy of ADH-6 in a wide range of cancer cell lines, using a variety of assays. These additional experiments are as follows:

1. We have tested the cytotoxic effects of ADH-6 in a wide range of cancer cell lines (as many with confirmed p53 status as we could acquire). This expanded screen contains more than twice as many cell lines with three times as many mutations as in the original screen. The cells lines tested include:
 - i. 11 cancer cell lines bearing wild-type p53 (compared to 5 cell lines in the original screen): Human bone (U-2 OS), brain (U-87 MG and SH-SY5Y), breast (MCF-7 and MDA-MB-175-VII), colon (LS 174T), renal (CAKI-1), leukemia (CESS), lung (A549 and NCI-H1882) and gastric (AGS), cancer cells.
 - ii. 12 cancer cells bearing 6 different aggregation-prone p53 mutations (compared to 5 cell lines bearing 2 mutations in the original screen): mutant R248W p53 (colon: COLO 320DM; lung: NCI-H1770; pancreatic: MIA PaCa-2), R248Q (breast: HCC70; ovarian: OVCAR-3), R175H (breast: SK-BR-3; colon: LS123), R273H (colon: WiDr; leukemia: ARH-77), Y220C (lung: NCI-H748 and NCI-H2342) or R280K (breast: MDA-MB-231), cancer cells.

Treatment of the cancer cells bearing WT p53 with ADH-6 yielded no significant toxicity. Conversely, we observed a substantial (~75–95%) decrease in viability of cancer cells harboring aggregation-prone p53 mutants at the same oligopyridylamide concentration and incubation time.

2. We have also probed the effects of transfecting two p53 null human cancer cell lines with mutant p53 on their susceptibility to ADH-6 mediated cytotoxicity:
 - i. We have transfected p53 null human ovarian cancer SKOV-3 cells with another commonly occurring aggregation-prone p53 mutant, R175H. Similar to the results of transfecting SKOV-3 cells with mutant R248W p53 reported in the original version of the manuscript, transfection with R175H rendered SKOV-3 cells highly susceptible to ADH-6 mediated cytotoxicity.
 - ii. Furthermore, we have tested another p53 null cell line, human bone cancer Saos-2. Saos-2 cells were not adversely affected by ADH-6. However, following transfection with mutant R248W or R175H p53, the oligopyridylamide induced significant toxicity in the cells.

Along with the expanded screen, these studies with p53 null cell lines strongly support our original conclusion that ADH-6 induces its toxic effects in cancer cells via targeting of mutant p53.

3. As outlined in the response to the Reviewer’s **Concern #3**, target engagement was also confirmed in MIA PaCa-2 and SK-BR-3 cells using CETSA. The results of these experiment are presented in the updated **Figure 3h,i** and new **Supplementary Figure 10** shown in the response to the Reviewer’s **Concern #3**.

4. Finally, we have studied the effects of the oligopyridylamides on SK-BR-3 xenografts (harboring mutant R175H p53) in mice. The results confirm that the oligopyridylamide effectively shrinks aggregation-prone mutant p53 bearing xenografts *in vivo*.

The results of the screen of WT and mutant p53 bearing cancer cells are presented in the updated **Supplementary Figure 11**:

Supplementary Figure 11. Effects of the oligopyridylamides on cancer cells harboring WT or mutant (R248W) p53. (a–c) Screen to identify oligopyridylamides that are toxic to cancer cells bearing mutant, but not WT, p53. MIA PaCa-2 (mutant R248W p53) (a,b) or MCF-7 (WT p53) (c) cells were treated with the indicated concentrations of ADH compounds for 24 or 48 h. (d) Effects of control peptide ReAcP53 on cancer cells bearing mutant p53. MIA PaCa-2 cells were treated with the indicated concentrations of ReAcP53 for 24 or 48 h. (e,f) Probing the effects of ADH-6 on viability of different cancer cells bearing WT or mutant p53. Bone (U-2 OS), brain (U-87 MG and SH-SY5Y), breast (MDA-MB-175-VII), colon (LS 174T), renal (CAKI-1), leukemia (CESS), lung (A549 and NCI-H1882) and gastric (AGS), cancer cells harboring WT p53 (e), or aggregation-prone R248W (colon: COLO 320DM; lung: NCI-H1770), R248Q (breast: HCC70; ovarian: OVCAR-3), R175H (colon: LS123), R273H (colon: WiDr; leukemia: ARH-77), Y220C (lung: NCI-H748 and NCI-H2342) and R280K (breast: MDA-MB-231), mutant p53 harboring cancer cells (f), were treated with 10 μM ADH-6 for 48 h. Cell viability (a–f) was assessed using the MTS assay, with the % viability determined from the ratio of the absorbance of the treated cells to the control cells (treated with vehicle alone). (g) Effects of ADH-6 on cell cycle distribution of mutant p53 bearing cancer cells. MIA PaCa-2 cells were treated with vehicle or 5 μM ADH-1, ReAcP53 or ADH-6 for 6 h. Cell cycle distribution of the cells was then evaluated using a cell cycle assay kit (Abcam), with measurements done on a BD FACSAria III cell sorter (n = 4). Shown are representative flow cytometry histograms for the different treatment groups. *P < 0.05, **P < 0.01, ***P < 0.001 or non-significant (ns, P > 0.05) compared with controls.

The results of the studies on mutant p53 transfected cancer cells are presented in the updated **Figure 4** and new **Supplementary Figure 12**:

Figure 4. ADH-6 causes death of cancer cells bearing mutant, but not WT, p53. (a–d) Effects of ADH-6 on viability of cancer cells bearing WT or mutant p53. (a–c) MIA PaCa-2 (mutant R248W p53) (a), MCF-7 (WT p53) (b), and SK-BR-3 (mutant R175H p53) (c), cells treated with increasing oligopyridylamide concentrations for 24 or 48 h. (d–f) p53 null Saos-2 cells before (d) and after transfection with p53 mutants, R248W (e) or R175H (f), treated with increasing concentrations of ADH-6 for 24 or 48 h. Cell viability in (a–f) was assessed using the MTS assay, with the % viability determined from the ratio of the absorbance of the treated cells to the control cells (n = 3). (g,h) Flow cytometry analysis of annexin V/propidium iodide (PI) staining of MIA PaCa-2 cells that were treated with vehicle (control), or 5 μM ADH-1, ReAcP53 or ADH-6, for 24 h (g). The bottom left quadrant (annexin V-/PI-) represents live cells; bottom right (annexin V+/PI-), early apoptotic cells; top right (annexin V+/PI+), late apoptotic cells; and top left (annexin V-/PI+), necrotic cells. A summary of the incidence of early/late apoptosis and necrosis in the different treatment groups determined from the flow cytometry analysis of annexin V/PI staining (n = 4) (h). (i) Cell cycle distribution of MIA PaCa-2 cells treated with vehicle (control), or 5 μM ADH-1, ReAcP53 or ADH-6, for 6 h as determined by measurement of DNA content using flow cytometry (n = 4). *P < 0.05, **P < 0.01, ***P < 0.001 or non-significant (ns, P > 0.05) compared with controls.

Supplementary Figure 12. Effects of aggregation-prone mutant p53 transfection on cancer cell susceptibility to ADH-6 mediated cytotoxicity. (a) Verification of successful transfection of p53 null Saos-2 and SKOV-3 cells with mutant R248W or R175H p53. The vectors were purified from the transfected cells, and the amplified amp^r PCR product was electrophoresed on a 3% agarose gel containing GelGreen Nucleic Acid Stain. (b,c) Viability of SKOV-3 cells before (b) and after transfection with mutant R248W p53 (c) treated with increasing concentrations of ADH-6 for 24 or 48 h. Cell viability in (b,c) was assessed using the MTS assay, with the % viability determined from the ratio of the absorbance of the treated cells to the control cells (n = 4). *P < 0.05, **P < 0.01, ***P < 0.001 or non-significant (ns, P > 0.05) for comparisons with controls.

The results of testing ADH-6 in a new xenograft model (SK-BR-3 bearing mutant R175H p53) are shown in the new **Supplementary Figure 21**:

Supplementary Figure 21. Effects of ADH-6 on SK-BR-3 xenografts in vivo. (a,b) A representative mouse bearing SK-BR-3 (mutant R175H p53) xenografts (a) and the treatment schedule (b). Once the tumor volume reached ~25 mm³, the mice were randomized into the different treatment groups (n = 8 per group), which were injected intraperitoneally with vehicle (0.02% DMSO) or 716.4 μM ADH-1 or ADH-6. Injections were done every 2 days for a total of 12 doses, with the first day of treatment defined as day 0. (c) Body weight changes of the tumor-bearing mice in the different treatment groups monitored for the duration of the experiment. (d) Tumor volume growth curves for the SK-BR-3 xenografts in the different treatment groups over 25 days of treatment (n = 8 per group). (e,f) Tumor mass analysis for the different treatment groups. After 25 days of treatment, the mice were sacrificed and the tumor tissues were isolated and imaged (e) and subsequently weighed to determine the tumor mass (f). *P < 0.05, **P < 0.01, ***P < 0.001 or non-significant (ns, P > 0.05) for comparison with controls and amongst the different treatment groups.

To address the Reviewer's concern, we have modified the relevant section of the **Revised Manuscript** as follows:

RESULTS; ADH-6 causes selective cytotoxicity in cancer cells bearing mutant p53:

“Next, we determined the effects of the oligopyridylamides on viability of mutant p53-harboring cancer cells using the MTS assay. A screen of the compounds revealed a strong correlation between their toxicity in mutant R248W p53-bearing MIA PaCa-2 cells (Figure 4a and Supplementary Figure 11a,b) and their capacity to antagonize pR248W aggregation (Figure 1e,f and Supplementary Figure 1b). Indeed, ADH-6, which was the most potent antagonist of pR248W amyloid formation, was also the most toxic compound to MIA PaCa-2 cells. ADH-6 reduced MIA PaCa-2 cell viability in a concentration-dependent manner, with an effective concentration (EC_{50}) of 2.7 ± 0.4 and 2.5 ± 0.1 μM at 24 and 48 h incubation times, respectively (Figure 4a), which was almost half of that for ReACp53 ($EC_{50} = 5.2 \pm 0.5$ and 4.9 ± 0.3 μM at 24 and 48 h incubation, respectively) (Supplementary Figure 11d). On the other hand, ADH-1, which did not inhibit pR248W aggregation, had no adverse effect on viability of MIA PaCa-2 cells (Figure 4a). Importantly, the oligopyridylamides, including ADH-6, were completely nontoxic to WT p53-bearing breast cancer MCF-7 cells (Figure 4b and Supplementary Figure 11c). Moreover, treatment of a range of human cancer cells bearing WT p53 (AGS, A549, CAKI-1, CESS, LS 174T, MDA-MB-175-VII, NCI-H1882, SH-SY5Y, U-2 OS and U-87 MG) with ADH-6 yielded no significant toxicity (Supplementary Figure 11e). Conversely, the oligopyridylamide was highly toxic to mutant R175H p53-bearing SK-BR-3 cells ($EC_{50} = 2.6 \pm 0.1$ and 2.5 ± 0.2 μM at 24 and 48 h incubation, respectively) (Figure 4c). Likewise, we observed a substantial (~75–95%) decrease in viability of human cancer cells harboring other aggregation-prone p53 mutants (R248W: COLO 320DM and NCI-H1770; R248Q: HCC70 and OVCAR-3; R175H: LS123; R273H: WiDr and ARH-77; Y220C: NCI-H748 and NCI-H2342; and R280K: MDA-MB-231) following treatment with ADH-6 (Supplementary Figure 11f).

Notably, ADH-6 did not adversely affect viability of p53 null human bone cancer Saos-2 cells (Figure 4d). However, transfecting the cells with mutant R248W or R175H p53 rendered them highly susceptible to ADH-6 mediated cytotoxicity ($EC_{50} = 2.0 \pm 0.2$ and 2.3 ± 0.2 μM for R248W and R175H, respectively, at 48 h incubation) (Figure 4e,f and Supplementary Figure 12a). Similarly, p53 null human ovarian cancer SKOV-3 cells were unaffected by treatment with ADH-6, but following transfection with R248W or R175H, the oligopyridylamide induced significant toxicity in the cells (Supplementary Figure 12a–d). Together with the screen of cancer cells bearing WT and mutant p53, these results indicate that the observed cytotoxicity of ADH-6 in cancer cells is directly related to the oligopyridylamide's capacity to antagonize mutant p53 amyloid formation.”

RESULTS; *In vivo* administration of ADH-6 causes regression of mutant p53-bearing tumors:

“Having established that ADH-6 potently abrogates mutant p53 amyloid formation and restores WT-like tumor suppressor function *in vitro*, we next assessed the *in vivo* efficacy of the oligopyridylamide (Figure 7 and Supplementary Figures 20,21). Following intraperitoneal injection, ADH-6 quickly entered circulation, with the peak concentration in serum mice (~21 $\mu\text{g}/\text{mL}$) occurring at 2 h post-injection (Figure 7a). The *in vivo* circulation half-life (Meibohm and Derendorf 1997) of ADH-6 ($t_{1/2} \sim 3.6$ h) was much longer than that of ReACp53 ($t_{1/2} \sim 1.5$ h) (Soragni et al. 2016), or other chemotherapeutics of a comparable size, such as doxorubicin ($t_{1/2} < 30$ min) (Palanikumar, Al-Hosani, Kalmouni, Nguyen, et al. 2020) or paclitaxel ($t_{1/2} \sim 1.7$ h) (Sparreboom et al. 1997). Moreover, ADH-6 was detected in the plasma up to 48 h after administration, whereas ReACp53 was eliminated from the bloodstream in 24 h (Soragni et al.

2016). The relatively long *in vivo* circulation time of ADH-6 should facilitate greater accumulation in tumor tissue. Indeed, the amount of ADH-6 in the MIA PaCa-2 xenografts increased continuously over 48 h post treatment (Figure 7b). The high *in vivo* stability suggested that ADH-6 would exhibit potent antitumor activity.

To test this hypothesis, mice bearing MIA PaCa-2 xenografts were treated with ADH-6, ADH-1 or ReACp53 (155.6 μ M in PBS) (Supplementary Figure 20a,b). Treatment consisted of intraperitoneal (IP) injections every 2 days, for a total of 12 doses (Supplementary Figure 20b). IP injection was used since this is the preferred administration route for the control ReACp53 peptide (Soragni et al. 2016). As expected, ADH-1 did not affect tumor growth (Supplementary Figure 20d–f). Conversely, both ADH-6 and ReACp53 reduced tumor growth relative to the saline-treated control group. However, of the two treatments, ADH-6 exhibited significantly greater antitumor efficacy (Supplementary Figure 20d–f). The effects of the oligopyridylamides were recapitulated in mice bearing SK-BR-3 xenografts, with ADH-6 again reducing tumor growth substantially compared to ADH-1 (Supplementary Figure 21).”

DISCUSSION:

“Treatment with ADH-6 resulted in substantial toxicity in mutant R248W p53-bearing MIA PaCa-2 cells (Figure 4). Several lines of evidence from multiple experiments establish that ADH-6-mediated cytotoxicity is directly related to the oligopyridylamide’s capacity to antagonize mutant p53 amyloid formation: (i) a second screen, testing the effects of the oligopyridylamides on viability of MIA PaCa-2 cells (Figure 4 and Supplementary Figure 11), revealed a strong correlation between the compounds’ cytotoxicity and their capacity to antagonize pR248W aggregation (Figure 1 and Supplementary Figure 1), and again identified ADH-6 as the most potent compound in the library; (ii) ADH-6 is significantly more toxic to MIA PaCa-2 cells than the control ReACp53 peptide (Figure 4 and Supplementary Figure 11), which reflects their relative capacities to abrogate intracellular mutant p53 aggregation (Figure 3 and Supplementary Figures 7–9); (iii) ADH-6 induced substantial loss of viability in a range of human cancer cells bearing aggregation-prone mutant p53, but was completely nontoxic to WT p53 harboring human cancer cells (Figure 4 and Supplementary Figure 11); and (iv) p53 null Saos-2 and SKOV-3 cells, which were insensitive to ADH-6 treatment, became highly susceptible to the cytotoxic effects of the oligopyridylamide upon transfection with mutant R248W or R175H p53 (Figure 4 and Supplementary Figure 12).”

DISCUSSION:

“To establish whether the *in vitro* effects of ADH-6 are recapitulated *in vivo*, we evaluated the effects of ADH-6 on mice bearing MIA PaCa-2 (mutant R248W p53) or SK-BR-3 (mutant R175H p53) tumors, alone or with MCF-7 (WT p53) tumors on the opposite flank as an internal control (Figure 7 and Supplementary Section 5). ReACp53, serving as a positive control, reduced MIA PaCa-2 tumor growth and prolonged survival relative to the vehicle-treated control groups, which is in agreement with the peptide’s reported ability to inhibit growth of aggregation-prone mutant p53-bearing tumors (Soragni et al. 2016). However, ADH-6 was markedly more effective at decreasing MIA PaCa-2 tumor volume and mass, and prolonging survival, compared to ReACp53. The greater *in vivo* efficacy of ADH-6 compared to ReACp53 correlates well with their relative capacities to dissociate intracellular mutant p53 aggregates (Figure 3 and Supplementary Figures 7–9) and to induce toxicity in mutant p53-bearing cancer cells (Figure 4 and Supplementary Figure 11), as well as with their relative *in vivo* stabilities. Peptides possess a number of pharmaceutically desirable properties, including the ability to selectively bind to specific targets with high potency, thereby minimizing off-target interactions and reducing the potential for toxicity (Kalmouni, Al-Hosani, and Magzoub 2019; Henning-Knechtel et al. 2020).

On the other hand, a major disadvantage of peptides is their low *in vivo* stability (Palanikumar, Al-Hosani, Kalmouni, Nguyen, et al. 2020). Synthetic protein mimetics, with their constrained backbone, are inherently more stable than peptides (Azzarito et al. 2013; Jayatunga, Thompson, and Hamilton 2014), which is reflected in the substantially longer *in vivo* circulation half-life and prolonged presence in the bloodstream of ADH-6 relative to ReACp53. The extended *in vivo* circulation time of ADH-6 facilitates increased accumulation in tumor tissue and enhanced anti-cancer activity. Importantly, ADH-6 exhibited high potency against MIA PaCa-2 and SK-BR-3 tumors but did not affect growth of control MCF-7 xenografts, underlining the oligopyridylamide's specificity for tumors that harbor aggregation-prone mutant p53. Additionally, detailed necropsies of major organs revealed no damage or alterations, demonstrating the lack of toxicity of ADH-6 to healthy tissue. Thus, our results show that ADH-6 effectively shrinks tumors bearing aggregation-prone mutant p53 *in vivo*, without displaying the non-specific toxicity that is common to conventional cancer therapeutics, thereby greatly prolonging survival."

MATERIALS AND METHODS; Cell culture:

"Prior to use, all cell lines were authenticated and tested for mycoplasma contamination by Charles River Laboratories (Margate, United Kingdom). The following cell lines were purchased from American Type Culture Collection (ATCC; Manassas, VA): human bone (U-2 OS [ATCC no. HTB-96] and Saos-2 [HTB-85]), brain (U-87 MG [HTB-14] and SH-SY5Y [CRL-2266]), breast (HCC70 [CRL-2315], MCF-7 [HTB-22], MDA-MB-175-VII [HTB-25], MDA-MB-231 [HTB-26] and SK-BR-3 [HTB-30]), colon (COLO 320DM [CCL-220], LS123 [CCL-255], LS 174T [CL-188] and WiDr [CCL-218]), leukemia (ARH-77 [CRL-1621] and CESS [TIB-190]), lung (A549 [CCL-185], NCI-H1770 [CRL-5893], NCI-H1882 [CRL-5903], NCI-H2342 [CRL-5941] and NCI-H748 [CRL-5841]), ovarian (OVCAR-3 [HTB-161] and SKOV-3 [HTB-77]), pancreatic (MIA PaCa-2 [CRL-1420]), renal (CAKI-1 [HTB-46]) and gastric (AGS [CRL-1739]), cancer cells. The cells were cultured in medium (DMEM, RPMI 1640 or Ham's Nutrient Mixture F12; Sigma) supplemented with 10% fetal bovine serum (FBS; GE Healthcare Life Sciences, Logan, UT), 4 mM L-glutamine, 1 mM sodium pyruvate and 1% penicillin/streptomycin (all from Sigma) in 5% CO₂ at 37 °C, and viability was monitored regularly using the Trypan Blue exclusion test on a TC20 automated cell counter (Bio-Rad, Hercules, CA). Once the cells reached ~95% confluence, they were split (using 0.25% trypsin-EDTA; Sigma) into fractions and propagated or used in experiments."

MATERIALS AND METHODS; Transfection of p53 null cells with mutant p53:

"Human bone cancer Saos-2 and ovarian cancer SKOV-3 cells were plated at a density of 1×10^5 cells/well in 2 mL complete medium in 6-well plates and cultured for 48 h. For each well of cells to be transfected, 2.5 µg pCMV-Neo-Bam p53 R175H or pCMV-Neo-Bam p53 R248W vectors (plasmids 16436 or 16437, respectively; Addgene, Watertown, MA) containing the *amp^r* gene was diluted in 500 µL Opti-MEM Reduced Serum Medium in an Eppendorf tube, to which 10 µL Lipofectamine LTX Reagent was added, mixed gently and incubated for 30 min at room temperature. The medium in the wells was then replaced with the Opti-MEM Reduced Serum Medium containing the DNA–Lipofectamine LTX Reagent complexes and the cells were incubated in 5% CO₂ at 37 °C for 24 h. Subsequently, the cells were harvested and plated in T25 flasks (Corning). After culturing until ~95% confluence was reached, the cells were selected with 400 µg/mL G418 sulfate-supplemented medium and maintained thereafter in medium containing 100 µg/mL G418 sulfate.

To verify that the transfection was successful, the pCMV-Neo-Bam p53 R248W and pCMV-Neo-Bam p53 R175H vectors containing the *amp^r* gene were purified from the transfected

SKOV-3 and Saos-2 cells using the Qiaprep Miniprep Kit (Qiagen, Hilden, Germany) according to the manufacturer's instructions. The miniprep-purified DNA samples were subsequently quantified using a NanoDrop spectrophotometer. PCR was carried out to amplify the *amp^r* genes using the following primers: forward (5'-AGATTATCAAAAAGGATCTTCACCT-3') and reverse (5'-CCTCGTGATACGCCTATTTTATAG-3') (Integrated DNA Technologies, Leuven, Belgium). Each PCR reaction was prepared in 50 μ L volume containing: 1 μ L of each primer (0.2 μ M final primer concentration), 50 ng DNA, 25 μ L Taq 2 \times PCR Master Mix (New England Biolabs, Ipswich, MA) and nuclease-free water. The conditions for each PCR reaction were: initial denaturation at 95 $^{\circ}$ C for 30 s, 35 cycles of denaturation at 95 $^{\circ}$ C for 30 s, annealing at 50 $^{\circ}$ C for 30 s, and extension at 68 $^{\circ}$ C for 1 min. As a negative control, a PCR reaction was carried out with the same reagents sans DNA. The amplified products were then resolved by electrophoresis using a 3% agarose gel prepared in 1 \times TAE (Tris/acetic acid/EDTA) buffer (Bio-Rad) containing GelGreen Nucleic Acid Stain (Biotium, Fremont, CA), which was visualized on an E-Gel Imager Blue-Light Base (Thermo Fisher Scientific). The presence of the successfully amplified *amp^r* PCR products was confirmed using a standard 500 bp ladder."

MATERIALS AND METHODS; *In vivo* tumor inhibition studies:

"For the single (mutant p53-harboring) xenograft inhibition studies, 5×10^5 viable MIA PaCa-2 or SK-BR-3 cancer cells were injected subcutaneously into the right flank of each mouse at age 6–8 weeks. For the dual (WT and mutant p53-bearing) xenograft model, 5×10^5 viable MIA PaCa-2 and MCF-7 cancer cells were injected into the right and left flanks, respectively, of each mouse at age 6–8 weeks. Mice were assessed daily for overt signs of toxicity. Tumor volume was measured via high-precision calipers (Thermo Fisher Scientific) using the following formula:

$$Tumor\ volume\ (mm^3) = \frac{W^2 \times L}{2} \quad (Equation\ 3)$$

where W and L are tumor width and length in mm, respectively (Luo et al. 2013). Mice were euthanized once tumor volume approached burden defined by NYUAD-IACUC.

Once the tumor volume reached $\sim 25\text{ mm}^3$, the mice were randomized into 4 treatment groups ($n = 8$ per group), which were injected intraperitoneally with: vehicle or the indicated concentrations of ADH-1, ReACp53 or ADH-6. Injections were done every 2 days for a total of 12 doses, with the first day of treatment defined as day 0. Body weight and tumor volume were recorded for the duration of treatment, and survival ($n = 4$ per group) was monitored for a total of 30–60 days. At the end of treatment, 4 mice per treatment group were sacrificed and the tumor tissues were isolated to determine the tumor mass."

Concern #5. Figure 2: the score changes between LTIITLE and LTRITILE peptides were subtle. Statistics analysis may be helpful.

Response: For further analysis, we plotted histograms of the binding scores of $\sim 13,000$ small molecules/biologics, including the oligopyridylamides (ADH-1 and ADH-6) and peptide sequences (LTIITLE and LTRITILE) of interest, interacting with WT and mutant R248W p53 DBDs (see Figure below). The resulting plots have normal distributions, with mean scores of 43.69 and 48.26 and standard deviations of 13.08 and 14.49 for WT and R248W DBDs, respectively. As the binding scores are negated to show positive values, a larger score denotes a stronger interaction. Z-scores were calculated to demonstrate the relative strength of interaction of each oligopyridylamide and peptide with WT and

mutant p53 DBDs. The calculated Z-scores (for WT DBD: ADH-1 = 1.02, ADH-6 = 3.31, LTIITLE = 1.16, and LTRITILE = 1.38; for R248W DBD: ADH-1 = 0.80, ADH-6 = 3.80, LTIITLE = 1.28, and LTRITILE = 1.80) confirm that binding of LTRITILE to both WT and mutant p53 DBDs is more favorable than that of LTIITLE. However, ADH-6 interacts markedly more strongly with both DBD variants compared to either ADH-1 or the peptide sequences. Thus the docking simulations corroborate the peptide aggregation assay results (Figure 1e,f,h and Supplementary Figure 1b,d), as well as the intracellular imaging and western blot analysis (Figure 3 and Supplementary Figures 7–9), which showed that ADH-6 is more effective than either ADH-1 or ReACp53 at antagonizing mutant p53 aggregation,

Histograms of binding scores for interactions with WT and mutant R248W p53 DBDs. Histograms of binding scores (negated to show positive scores) of ~13,000 small molecules/biologics interacting with WT (left panel) and mutant R248W (right panel) p53 DBDs. The oligopyridylamides (ADH-1 and ADH-6) and peptide sequences (LTIITLE and LTRITILE) of interest are depicted with vertical lines. Z-scores for WT DBD: ADH-1 = 1.02; ADH-6 = 3.31; LTIITLE = 1.16; and LTRITILE = 1.38. Z-scores for R248W DBD: ADH-1 = 0.80; ADH-6 = 3.80; LTIITLE = 1.28; and LTRITILE = 1.80.

However, closer examination of the molecular docking results revealed a loss of rigidity of the oligopyridylamide structures in some of the generated binding poses. Despite rerunning the docking simulations multiple times, this problem persisted. The docking software, CANDOCK (Fine et al. 2020), uses a fragment-based approach to find the global minimum binding pose, which bypasses the need to generate innumerable conformations. These fragments are created at notable bonds, such as amide bonds. Consequently, the minimum poses often result in intermolecular hydrogen bonding taking precedence over intramolecular hydrogen bonding. In the case of the oligopyridylamides, this sometimes leads to loss of the bifurcated hydrogen bonding (between the oxygen of the side chains, nitrogen of the ring and the nitrogen of the amide bond) that stabilizes the structures and prevents rotation around the amide bond. This, in turn, yields artificially flexible oligopyridylamide structures in the binding poses. This issue was also observed with the other docking programs at our disposal, including AutoDock Vina.

Thus, while the molecular docking simulations corroborate the experimental results, we have decided to remove this data due to the inaccurate flexibility of the oligopyridylamide structures in some of the generated binding poses. It is important to note that removing the docking simulation results does not, in any way, alter or dilute the conclusions of the study, which are backed by extensive experimental data, particularly after the additional experiments done to address the Reviewers' concerns. As an aside, removing the simulation results also helps to shorten the length of the **Revised Manuscript** (which, despite removal of the docking section, is longer than the already lengthy original version). We hope that the Reviewer agrees with our reasoning and decision.

Concern #6. The proteomics analysis of phosphoprotein (Fig. 7) is interesting. However, how the study is connected to p53 reactivation is not clear.

Response: To complement our transcriptome analysis, we probed the effects of ADH-6 treatment on mutant p53-bearing cancer cells at the proteome level. Our choice to focus specifically on the phosphoproteome (**Supplementary Figure 18**) was due to the fact that many phosphoproteins are involved in regulating major pathways implicated in cancer (Altomare et al. 2004; Emaduddin et al. 2008; Takano et al. 2010; Ardito et al. 2017; Cicenas 2008). A comparable reduction of phosphoprotein expression, in a number of pathways, was observed following treatment with ReACp53 and ADH-6 (**Figure 6**). The biological roles of the downregulated phosphoproteins were inferred from published data (this data is summarized in **Supplementary Table 2**). This analysis revealed that the downregulated phosphoproteins can be divided into two major functional groups that positively regulate either DNA replication/repair or cell cycle progression/proliferation, both of which play a critical role in cancer initiation and progression. Importantly, these two groups are known to be directly regulated by p53 (Harris and Levine 2005; Giono and Manfredi 2006; Brady et al. 2011; J. Chen 2016; Williams and Schumacher 2016). This strongly supports that both the ReACp53 and ADH-6 treatments result in reactivation of p53, which downregulates key cancer-promoting phosphoproteins. Indeed, numerous studies have reported that p53 directly downregulates the expression of positive regulators of cancer in the p53 pathway, such as CD44 (Godar et al. 2008) and PCNA (Shivakumar et al. 1995; Saifudeen et al. 2002), as well as the oncoproteins c-Myc (Levy et al. 1993; Sachdeva et al. 2009), E2F (Timmers et al. 2007; Welch, Chen, and Stallings 2007) and mTORC1 (Stambolic et al. 2001; Feng et al. 2005; Hasty et al. 2013), all of which were shown to be downregulated in our analysis (**Figure 6** and **Supplementary Figure 19**). We believe that, together with the RNA-Seq results, the phosphoproteome analysis demonstrates that ADH-6-mediated dissociation of mutant p53 amyloid-like aggregates in cancer cells restores p53 function, leading to potent anticancer effects (Figure 4). We hope that this clarifies the relevance and importance of the phosphoproteome analysis to the evidence of ADH-6 mediated p53 reactivation.

Response to Reviewer #2

This manuscript by Palanikumar and coauthors, titled “Protein mimetic amyloid inhibitor potentially abrogates cancer-associated mutant p53 aggregation and restores tumor suppressor function” reports the study of oligopyridylamides as molecules able to inhibit the aggregation of mutant p53 proteins in cancer cells and to cause apoptotic cell death. The study is very well described and developed. The data highlight the potential of the identified compound to reactivate p53 activity in mutant p53-carrying cells without showing toxicity in cells bearing wt-p53 or in p53-null systems. A few concerns are listed below to improve the strength of the study.

Response: We thank the Reviewer for the positive comments regarding the manuscript. We are grateful for the feedback provided and have worked diligently to address all of the concerns. In the following pages, we respond point-by-point to the Reviewer’s comments. We believe, and hope the Reviewer agrees, that the paper is much improved in content and clarity.

Major Concerns

Concern #1. Figures 6-7: Authors interestingly show that treatment with ADH-6, but not ADH-1, not only impair the formation of aggregates by mutant p53 proteins but also reactivate p53 function. It is surprising and very relevant that mutant p53 proteins acquire again the ability of wt-p53 to transactivate its bona fide targets. Is p53 really regulating these genes or is this an indirect effect of other p53 family members?

To confer additional strength to these findings, the authors should analyze the recruitment of mutant p53 and other p53 family members on the p53 binding sites of target genes (ChIP), thus formally demonstrating the eventual reacquired transcriptional activity of mutant p53.

Response: ~90% of cancer-associated p53 mutations occur within its DNA-binding domain (DBD) (Bouaoun et al. 2016; Baugh et al. 2018). The p53 DBD is characterized by low thermodynamic and kinetic stability (Joerger and Fersht 2010), and mutations in the domain often decrease its stability further and prompt its unfolding, leading to exposure of a normally hidden aggregation-nucleating subdomain, p53₂₅₁₋₂₅₈ (Wang and Fersht 2012; Soragni et al. 2016). This prompts self-assembly of mutant p53 into amyloid-like aggregates within inactive cellular inclusions (Xu et al. 2011). The cationic tripyridylamide ADH-6 effectively targets and inhibits mutant p53 amyloid’s aggregation, thereby strongly shifting the folding equilibrium towards the functional, WT-like state. The restoration of p53’s transcriptional activity leads to cell cycle arrest and apoptosis *in vitro* and reduction in tumor growth *in vivo*.

To address the Reviewer’s concern and provide further support for the proposed ADH-6 mediated transcriptional reactivation of mutant p53, we have performed chromatin immunoprecipitation (ChIP) coupled with quantitative PCR (ChIP-qPCR) to detect recruitment of mutant p53 to the WT protein’s binding sites on promoters/enhancers of target genes in oligopyridylamide treated cancer cells. Treatment of MIA PaCa-2 cells with ADH-6 led to significant binding of R248W to target p53 target genes, *CDKN1A* (also known as *P21*), *PIG3* and *NOXA*. Importantly, we did not detect significant binding of the p53 homologs, p63 and p73, to *CDKN1A* following treatment with ADH-6. Moreover, western blot analysis confirmed elevated expression of p53 targets in ADH-6 treated MIA PaCa-2 cells. Thus, the ChIP-qPCR results strongly support our original conclusion that ADH-6 specifically targets and reactivates aggregation-prone mutant p53.

The results of the ChIP-qPCR experiments are presented in the new **Supplementary Figure 14**:

Supplementary Figure 14. ChIP-qPCR analysis of recruitment of mutant R248W p53 to the WT protein's binding sites on promoters/enhancers of target genes. Binding of mutant R248W p53 to the WT protein's transcriptional targets, CDKN1A (P21) (a), PIG3 (b) and NOXA (c), in MIA PaCa-2 cells treated with vehicle or 5 μ M ADH-6 for 24 h. For comparison, binding of p63 and p73 to CDKN1A in ADH-6 treated MIA PaCa-2 cells is also presented (a). * $P < 0.05$, ** $P < 0.01$, *** $P < 0.001$ or non-significant (ns, $P > 0.05$) for comparisons with vehicle-treated controls.

The primers used for the ChIP-qPCR experiment are shown in the new **Supplementary Table 1**:

Supplementary Table 1a. Mutant R248W p53			
Gene	Primer	References	Sequence
CDKN1A (P21)	Forward (5'-3')	(el-Deiry et al. 1993)	GTGGCTCTGATTGGCTTTCTG
	Reverse (3'-5')		CTGAAAACAGGCAGCCCAAG
BAX	Forward (5'-3')	(Miyashita and Reed 1995)	TAATCCCAGCGCTTTGGAA
	Reverse (3'-5')		TGCAGAGACCTGGATCTAGCAA
PUMA	Forward (5'-3')	(Nakano and Vousden 2001)	GCGAGACTGTGGCCTTGTGT
	Reverse (3'-5')		CGTCCAGGGTCCACAAAGT
NOXA	Forward (5'-3')	(Oda et al. 2000)	CAGCGTTTGCAGATGGTCAA
	Reverse (3'-5')		CCCCGAAATTACTTCCTTACAAAA
PIG3	Forward (5'-3')	(Kaeser and Iggo 2002)	CACTCCCAACGCCTCCTTT
	Reverse (3'-5')		GCCCATCTTGAGCATGGGT
MDM2	Forward (5'-3')	(Wu et al. 1993)	GGTTGACTCAGCTTTTCCTCTTG
	Reverse (3'-5')		GGAAAATGCATGGTTTAAATAGCC
GADD45	Forward (5'-3')	(Kastan et al. 1992)	AGCGGAAGAGATCCCTGTGA
	Reverse (3'-5')		CGGGAGGCAGGCAGATG

Supplementary Table 1b. p63			
Gene	Primer	References	Sequence
CDKN1A (P21)	Forward (5'-3')	(Kouwenhoven et al. 2010)	GTGGCTCTGATTGGCTTTCTG
	Reverse (3'-5')		CTGAAAACAGGCAGCCCAAG

Supplementary Table 1c. p73			
Gene	Primer	References	Sequence
CDKN1A (P21)	Forward (5'-3')	(Koeppel et al. 2011)	GTGGCTCTGATTGGCTTTCT
	Reverse (3'-5')		AGCCTTTCTATGCCAGAGC

In addressing the Reviewer's concern, we have expanded the relevant sections of the **Revised Manuscript and Revised Supplementary Material** as follows:

RESULTS; ADH-6 induces transcriptional reactivation of p53:

“To confirm that ADH-6 rescues p53 function, we first used ChIP-qPCR to detect recruitment of mutant R248W p53 to the WT protein's binding sites on promoters/enhancers of target genes in oligopyridylamide treated MIA PaCa-2 cells (primers used are listed in Supplementary Table 1). Treatment with ADH-6 led to binding of R248W to *CDKN1A* (also known as *P21*) (Supplementary Figure 14a). Importantly, we did not detect significant binding of the p53 homologs, p63 and p73, to *CDKN1A* under the same experimental conditions (Supplementary Figure 14a), supporting the notion that ADH-6 activity is mutant p53 dependent. Interaction of R248W with other well-established primary p53 transcriptional targets, *PIG3* and *NOXA*, was also observed in ADH-6 treated MIA PaCa-2 cells (Supplementary Figure 14b,c). (Please note that functions of the genes referenced in this section are described in Supplementary Section 4.)”

SUPPLEMENTARY SECTION 4.1.; Chromatin immunoprecipitation (ChIP) with quantitative real-time PCR (ChIP-qPCR) assay:

“ChIP-qPCR (primers used are listed in Supplementary Table 1) revealed that treatment of MIA PaCa-2 cells with ADH-6 resulted in binding of mutant R248W p53 to the WT protein's binding sites on promoters/enhancers of target genes, *CDKN1A* (also known as *P21*), *PIG3* and *NOXA* (Supplementary Figure 14). p21, a cyclin-dependent kinase inhibitor, is required for p53-mediated cell cycle arrest in response to DNA damage and other cellular stresses (Abbas and Dutta 2009; Karimian, Ahmadi, and Yousefi 2016). *PIG3* (p53-induced gene 3) is involved in both the early cellular response to DNA damage and p53-induced apoptosis (Polyak et al. 1997; Lee et al. 2010). *NOXA* (Latin for damage) encodes a member of the Bcl-2 family of apoptosis regulator proteins; upon p53-induced expression, Noxa localizes to mitochondria, where the protein binds to and inhibits anti-apoptotic Bcl-2 family members (Aubrey et al. 2018). Notably, Noxa has also been identified as a key determinant of cytotoxic response of cancer cells to standard chemotherapy and targeted cancer therapy (Albert, Brinkmann, and Kashkar 2014; Montero et al. 2019).”

DISCUSSION:

“Further analysis of the effects of ADH-6 on mutant p53 harboring cancer cells revealed that the oligopyridylamide induces cell cycle arrest and apoptosis, both of which are indicators of restored p53 function (Figure 4 and Supplementary Figure 11). ADH-6-mediated restoration of WT-like activity of mutant p53 was confirmed using a number of complementary assays. ChIP-qPCR analysis revealed that ADH-6 treatment resulted in recruitment of mutant p53 to the WT protein's primary transcriptional targets, such as *CDKN1A*, *PIG3* and *NOXA* (Supplementary Figure 14 and Supplementary Table 1), that are not only key mediators of p53-dependent cell cycle arrest and apoptosis (Abbas and Dutta 2009; Karimian, Ahmadi, and Yousefi 2016; Polyak et al. 1997; Lee et al. 2010; Oda et al. 2000), but also determinants of the cytotoxic response of cancer cells to standard chemotherapy and targeted cancer therapy (Albert, Brinkmann, and Kashkar 2014; Montero et al. 2019). Transcriptional reactivation of mutant p53 was further corroborated using RNA-Seq (Figure 5 and Supplementary Figures 16,17). Specifically, we observed activation of the *TP53* and apoptosis pathways, and suppression of genes involved in cell cycle progression (G2/M) and the *MYC* pathway, which are known to be negatively regulated by p53 (Sachdeva et al. 2009).”

MATERIALS AND METHODS; Chromatin immunoprecipitation (ChIP) coupled with quantitative PCR (ChIP-qPCR):

“MIA PaCa-2 cells were cultured in complete DMEM in T175 flasks (Corning) until 70–80% confluence was reached. Thereafter, the medium was replaced with serum-free DMEM containing vehicle or 5 μ M ADH-6, and the cells were incubated for a further 24 h. The cells were then treated with 1% PFA for 10 min at 37 °C for crosslinking, followed by a single wash with 5 mL PBS. 1.5 mL buffer A (5 mM PIPES, pH 8, 85 mM KCl, 0.5% NP40, Protease Inhibitor Cocktail) was added to the cells, which were harvested using a pipette tip and scrapper. The cell suspension was rotated for 10 min at 4 °C, centrifuged (3,700 rpm for 5 min at 4 °C), and the supernatant was discarded. To the cell pellet, 400 μ L buffer B (1% SDS, 10 mM EDTA, 50 mM Tris-HCL, pH 8.1) was added, and the cell suspension was sonicated (10 cycles of 5 s each at maximum power, with 2 min on ice between cycles to prevent overheating). Cell debris was removed by centrifugation (13,000 rpm for 10 min at 4 °C), and the supernatant was collected. The samples were diluted tenfold with the immunoprecipitation buffer (0.01% SDS, 1.1% Triton X-100, 1.2 mM EDTA, 16.7 mM Tris-HCL, pH 8.1), rotated for 30 min at room temperature, followed by centrifugation (1,500 rpm for 20 s at 4 °C). Of the collected supernatant, 100 μ L was used as input for each sample, with the remainder incubated with 1 μ g/mL of the antibody for p53 (DO-1; Santa Cruz), p63 (D2K8X; Cell Signaling Technology, Danvers, MA) or p73 (EPR19884; Abcam), overnight on a rotator at 4 °C. 50 μ L magnetic beads (Magna ChIP protein A+G bead blend; Sigma) was added to each sample solution and rotated for 30 min at room temperature. After rotation, the samples were incubated for 30 s on a magnetic stand to separate the chromatin, which was then washed twice with wash buffer 1 (0.1% SDS, 1% TritonX-100, 2 mM EDTA, 20 mM Tris-HCL, pH 8.1, 150 mM NaCl), followed by twice with wash buffer 2 (0.1% SDS, 1% TritonX-100, 2 mM EDTA, 20 mM Tris-HCL, pH 8.1, 300 mM NaCl), twice with wash buffer 3 (0.25 M LiCl, 1% NP40, 1% sodium deoxycholate, 1 mM EDTA, 10 mM Tris-HCL, pH 8.1), and finally once with TE buffer (10 mM Tris-HCL, pH 8.1, 1 mM EDTA). Immediately after the washes, the samples were eluted by adding the elution buffer (0.5% SDS, 5 mM EDTA, 25 mM Tris-HCL, pH 7.5, 20 mM NaCl), followed by vortexing and incubation for 15 min at 65 °C. Subsequently, the samples were incubated on a magnetic stand and 200 μ L supernatant was collected, to which 200 μ L immunoprecipitation buffer was added, followed by 1 μ L Protease Inhibitor Cocktail, and the samples were incubated overnight at 65 °C. DNA was recovered with a conventional in-house DNA precipitation method using sodium acetate, with samples eluted in volumes of 20 μ L. Finally, the recovered DNA was amplified and quantified (for qPCR reaction: 1 μ L DNA from ChIP samples or input, 1 μ L 2 mM primers (Supplementary Table 1), 5 μ L SYBr Green PCR Master Mix (Thermo Fisher Scientific), and 3 μ L nuclease-free water). PCR thermal reaction cycle included one cycle of 10 min at 95 °C, and 39 cycles of 15 s at 95 °C followed by 45 s at 60 °C. Normalization of PCR reaction was based on calculating the enrichment by comparing the threshold cycle number (C_T) for the input and ChIP samples.”

Concern #2. Authors list a number of cell lines as having wt-p53. To my knowledge SW480 and SW1990 cell lines carry mutated TP53, with mutants p.R273H and p.P309S reported in SW480, and mutant p.P191del in SW1990 (as from IARC database). Did the authors check the mutational status of TP53 coding exons in the cell lines used in their study? This surely needs to be done.

Response: We thank the Reviewer for their careful reading of the manuscript! In our original screen, we had tested the colon cancer cell line SW48, which was then erroneously recorded as another colon cancer cell line, SW480.

SW48 and SW1990 are both classed as bearing WT p53 by ATCC, the source from which the cells were acquired. However, as the Reviewer points out, the IARC TP53 database reports that SW1990 harbors a mutant form of p53 (p.P191del). Although this deletion does not appear to render the protein aggregation-prone (there are no reports, to our knowledge, of aggregation of mutant p53 bearing this mutation), we have decided to remove the cell line from our screen to avoid any potential confusion.

The SW48 cell line is widely reported in the literature to harbor WT p53. However, in the IARC TP53 database the cell line is recorded as carrying both WT and mutant p53. While we have been unable to locate this, apparently solitary, study reporting mutant p53 in SW48 (the link from the database to the article in PubMed is broken), we have decided to remove this cell line from our screen as well.

We have now tested the efficacy of ADH-6 in a wider range of cancer cell lines. The expanded screen includes the following WT and mutant p53 bearing cancer cell lines with verified p53 status (according to both ATCC and the IARC TP53 database):

- i. 11 cancer cell lines bearing wild-type p53 (compared to 5 cell lines in the original screen): Human bone (U-2 OS), brain (U-87 MG and SH-SY5Y), breast (MCF-7 and MDA-MB-175-VII), colon (LS 174T), renal (CAKI-1), leukemia (CESS), lung (A549 and NCI-H1882) and gastric (AGS), cancer cells.
- ii. 12 cancer cells bearing 6 different aggregation-prone p53 mutations (compared to 5 cell lines bearing 2 mutations in the original screen): R248W (colon: COLO 320DM; lung: NCI-H1770; pancreatic: MIA PaCa-2), R248Q (breast: HCC70; ovarian: OVCAR-3), R175H (breast: SK-BR-3; colon: LS123), R273H (colon: WiDr; leukemia: ARH-77), Y220C (lung: NCI-H748 and NCI-H2342) or R280K (breast: MDA-MB-231), cancer cells.

Treatment of the cancer cells bearing WT p53 with ADH-6 yielded no significant toxicity. Conversely, we observed a substantial (~75–95%) decrease in viability of cancer cells harboring aggregation-prone p53 mutants at the same oligopyridylamide concentration and incubation time.

Additionally, we have probed the effects of transfecting two p53 null human cancer cell lines with mutant p53 on their susceptibility to ADH-6 mediated cytotoxicity:

- i. We have transfected p53 null human ovarian cancer SKOV-3 cells with another commonly occurring aggregation-prone p53 mutant, R175H. Similar to the results of transfecting SKOV-3 cells with mutant R248W p53 reported in the original version of the manuscript, transfection with R175H rendered SKOV-3 cells highly susceptible to ADH-6 mediated cytotoxicity.
- ii. Furthermore, we have tested another p53 null cell line, human bone cancer Saos-2. Saos-2 cells were not adversely affected by ADH-6. However, following transfection with mutant R248W or R175H p53, the oligopyridylamide induced significant toxicity in the cells.

Along with the expanded screen, these studies with p53 null cell lines strongly support our original conclusion that ADH-6 induces its toxic effects in cancer cells via targeting of mutant p53.

To address the Reviewer's **Concern #2**, the relevant sections of the **Revised Manuscript** have been updated as follows:

RESULTS; ADH-6 causes selective cytotoxicity in cancer cells bearing mutant p53:

“Next, we determined the effects of the oligopyridylamides on viability of mutant p53-harboring cancer cells using the MTS assay. A screen of the compounds revealed a strong correlation between their toxicity in mutant R248W p53-bearing MIA PaCa-2 cells (Figure 4a and Supplementary Figure 11a,b) and their capacity to antagonize pR248W aggregation (Figure 1e,f and Supplementary Figure 1b). Indeed, ADH-6, which was the most potent antagonist of

pR248W amyloid formation, was also the most toxic compound to MIA PaCa-2 cells. ADH-6 reduced MIA PaCa-2 cell viability in a concentration-dependent manner, with an effective concentration (EC_{50}) of 2.7 ± 0.4 and 2.5 ± 0.1 μM at 24 and 48 h incubation times, respectively (Figure 4a), which was almost half of that for ReACp53 ($EC_{50} = 5.2 \pm 0.5$ and 4.9 ± 0.3 μM at 24 and 48 h incubation, respectively) (Supplementary Figure 11d). On the other hand, ADH-1, which did not inhibit pR248W aggregation, had no adverse effect on viability of MIA PaCa-2 cells (Figure 4a). Importantly, the oligopyridylamides, including ADH-6, were completely nontoxic to WT p53-bearing breast cancer MCF-7 cells (Figure 4b and Supplementary Figure 11c). Moreover, treatment of a range of human cancer cells bearing WT p53 (AGS, A549, CAKI-1, CESS, LS 174T, MDA-MB-175-VII, NCI-H1882, SH-SY5Y, U-2 OS and U-87 MG) with ADH-6 yielded no significant toxicity (Supplementary Figure 11e). Conversely, the oligopyridylamide was highly toxic to mutant R175H p53-bearing SK-BR-3 cells ($EC_{50} = 2.6 \pm 0.1$ and 2.5 ± 0.2 μM at 24 and 48 h incubation, respectively) (Figure 4c). Likewise, we observed a substantial (~75–95%) decrease in viability of human cancer cells harboring other aggregation-prone p53 mutants (R248W: COLO 320DM and NCI-H1770; R248Q: HCC70 and OVCAR-3; R175H: LS123; R273H: WiDr and ARH-77; Y220C: NCI-H748 and NCI-H2342; and R280K: MDA-MB-231) following treatment with ADH-6 (Supplementary Figure 11f).

Notably, ADH-6 did not adversely affect viability of p53 null human bone cancer Saos-2 cells (Figure 4d). However, transfecting the cells with mutant R248W or R175H p53 rendered them highly susceptible to ADH-6 mediated cytotoxicity ($EC_{50} = 2.0 \pm 0.2$ and 2.3 ± 0.2 μM for R248W and R175H, respectively, at 48 h incubation) (Figure 4e,f and Supplementary Figure 12a). Similarly, p53 null human ovarian cancer SKOV-3 cells were unaffected by treatment with ADH-6, but following transfection with R248W or R175H, the oligopyridylamide induced significant toxicity in the cells (Supplementary Figure 12a–d). Together with the screen of cancer cells bearing WT and mutant p53, these results indicate that the observed cytotoxicity of ADH-6 in cancer cells is directly related to the oligopyridylamide's capacity to antagonize mutant p53 amyloid formation.”

DISCUSSION:

“Treatment with ADH-6 resulted in substantial toxicity in mutant R248W p53-bearing MIA PaCa-2 cells (Figure 4). Several lines of evidence from multiple experiments establish that ADH-6-mediated cytotoxicity is directly related to the oligopyridylamide's capacity to antagonize mutant p53 amyloid formation: (i) a second screen, testing the effects of the oligopyridylamides on viability of MIA PaCa-2 cells (Figure 4 and Supplementary Figure 11), revealed a strong correlation between the compounds' cytotoxicity and their capacity to antagonize pR248W aggregation (Figure 1 and Supplementary Figure 1), and again identified ADH-6 as the most potent compound in the library; (ii) ADH-6 is significantly more toxic to MIA PaCa-2 cells than the control ReACp53 peptide (Figure 4 and Supplementary Figure 11), which reflects their relative capacities to abrogate intracellular mutant p53 aggregation (Figure 3 and Supplementary Figures 7–9); (iii) ADH-6 induced substantial loss of viability in a range of human cancer cells bearing aggregation-prone mutant p53, but was completely nontoxic to WT p53 harboring human cancer cells (Figure 4 and Supplementary Figure 11); and (iv) p53 null Saos-2 and SKOV-3 cells, which were insensitive to ADH-6 treatment, became highly susceptible to the cytotoxic effects of the oligopyridylamide upon transfection with mutant R248W or R175H p53 (Figure 4 and Supplementary Figure 12).”

MATERIALS AND METHODS; Cell culture:

“Prior to use, all cell lines were authenticated and tested for mycoplasma contamination by Charles River Laboratories (Margate, United Kingdom). The following cell lines were purchased from American Type Culture Collection (ATCC; Manassas, VA): human bone (U-2 OS [ATCC no. HTB-96] and Saos-2 [HTB-85]), brain (U-87 MG [HTB-14] and SH-SY5Y [CRL-2266]), breast (HCC70 [CRL-2315], MCF-7 [HTB-22], MDA-MB-175-VII [HTB-25], MDA-MB-231 [HTB-26] and SK-BR-3 [HTB-30]), colon (COLO 320DM [CCL-220], LS123 [CCL-255], LS 174T [CL-188] and WiDr [CCL-218]), leukemia (ARH-77 [CRL-1621] and CESS [TIB-190]), lung (A549 [CCL-185], NCI-H1770 [CRL-5893], NCI-H1882 [CRL-5903], NCI-H2342 [CRL-5941] and NCI-H748 [CRL-5841]), ovarian (OVCAR-3 [HTB-161] and SKOV-3 [HTB-77]), pancreatic (MIA PaCa-2 [CRL-1420]), renal (CAKI-1 [HTB-46]) and gastric (AGS [CRL-1739]), cancer cells. The cells were cultured in medium (DMEM, RPMI 1640 or Ham’s Nutrient Mixture F12; Sigma) supplemented with 10% fetal bovine serum (FBS; GE Healthcare Life Sciences, Logan, UT), 4 mM L-glutamine, 1 mM sodium pyruvate and 1% penicillin/streptomycin (all from Sigma) in 5% CO₂ at 37 °C, and viability was monitored regularly using the Trypan Blue exclusion test on a TC20 automated cell counter (Bio-Rad, Hercules, CA). Once the cells reached ~95% confluence, they were split (using 0.25% trypsin-EDTA; Sigma) into fractions and propagated or used in experiments.”

MATERIALS AND METHODS; Transfection of p53 null cells with mutant p53:

“Human bone cancer Saos-2 and ovarian cancer SKOV-3 cells were plated at a density of 1×10^5 cells/well in 2 mL complete medium in 6-well plates and cultured for 48 h. For each well of cells to be transfected, 2.5 µg pCMV-Neo-Bam p53 R175H or pCMV-Neo-Bam p53 R248W vectors (plasmids 16436 or 16437, respectively; Addgene, Watertown, MA) containing the *amp^r* gene was diluted in 500 µL Opti-MEM Reduced Serum Medium in an Eppendorf tube, to which 10 µL Lipofectamine LTX Reagent was added, mixed gently and incubated for 30 min at room temperature. The medium in the wells was then replaced with the Opti-MEM Reduced Serum Medium containing the DNA–Lipofectamine LTX Reagent complexes and the cells were incubated in 5% CO₂ at 37 °C for 24 h. Subsequently, the cells were harvested and plated in T25 flasks (Corning). After culturing until ~95% confluence was reached, the cells were selected with 400 µg/mL G418 sulfate-supplemented medium and maintained thereafter in medium containing 100 µg/mL G418 sulfate.

To verify that the transfection was successful, the pCMV-Neo-Bam p53 R248W and pCMV-Neo-Bam p53 R175H vectors containing the *amp^r* gene were purified from the transfected SKOV-3 and Saos-2 cells using the Qiaprep Miniprep Kit (Qiagen, Hilden, Germany) according to the manufacturer’s instructions. The miniprep-purified DNA samples were subsequently quantified using a NanoDrop spectrophotometer. PCR was carried out to amplify the *amp^r* genes using the following primers: forward (5’-AGATTATCAAAAAGGATCTTACCT-3’) and reverse (5’-CCTCGTGATACGCCTATTTTATAG-3’) (Integrated DNA Technologies, Leuven, Belgium). Each PCR reaction was prepared in 50 µL volume containing: 1 µL of each primer (0.2 µM final primer concentration), 50 ng DNA, 25 µL Taq 2× PCR Master Mix (New England Biolabs, Ipswich, MA) and nuclease-free water. The conditions for each PCR reaction were: initial denaturation at 95 °C for 30 s, 35 cycles of denaturation at 95 °C for 30 s, annealing at 50 °C for 30 s, and extension at 68 °C for 1 min. As a negative control, a PCR reaction was carried out with the same reagents sans DNA. The amplified products were then resolved by electrophoresis using a 3% agarose gel prepared in 1× TAE (Tris/acetic acid/EDTA) buffer (Bio-Rad) containing GelGreen Nucleic Acid Stain (Biotium, Fremont, CA), which was visualized

on an E-Gel Imager Blue-Light Base (Thermo Fisher Scientific). The presence of the successfully amplified *amp^r* PCR products was confirmed using a standard 500 bp ladder.”

The results of the expanded screen are presented in the updated **Supplementary Figure 11**:

Supplementary Figure 11. Effects of the oligopyridylamides on cancer cells harboring WT or mutant (R248W) p53. (a–c) Screen to identify oligopyridylamides that are toxic to cancer cells bearing mutant, but not WT, p53. MIA PaCa-2 (mutant R248W p53) (a,b) or MCF-7 (WT p53) (c) cells were treated with the indicated concentrations of ADH compounds for 24 or 48 h. (d) Effects of control peptide ReAcP53 on cancer cells bearing mutant p53. MIA PaCa-2 cells were treated with the indicated concentrations of ReAcP53 for 24 or 48 h. (e,f) Probing the effects of ADH-6 on viability of different cancer cells bearing WT or mutant p53. Bone (U-2 OS), brain (U-87 MG and SH-SY5Y), breast (MDA-MB-175-VII), colon (LS 174T), renal (CAKI-1), leukemia (CESS), lung (A549 and NCI-H1882) and gastric (AGS), cancer cells harboring WT p53 (e), or aggregation-prone R248W (colon: COLO 320DM; lung: NCI-H1770), R248Q (breast: HCC70; ovarian: OVCAR-3), R175H (colon: LS123), R273H (colon: WiDr; leukemia: ARH-77), Y220C (lung: NCI-H748 and NCI-H2342) and R280K (breast: MDA-MB-231), mutant p53 harboring cancer cells (f), were treated with 10 μM ADH-6 for 48 h. Cell viability (a–f) was assessed using the MTS assay, with the % viability determined from the ratio of the absorbance of the treated cells to the control cells (treated with vehicle alone). (g) Effects of ADH-6 on cell cycle distribution of mutant p53 bearing cancer cells. MIA PaCa-2 cells were treated with vehicle or 5 μM ADH-1, ReAcP53 or ADH-6 for 6 h. Cell cycle distribution of the cells was then evaluated using a cell cycle assay kit (Abcam), with measurements done on a BD FACSAria III cell sorter (n = 4). Shown are representative flow cytometry histograms for the different treatment groups. *P < 0.05, **P < 0.01, ***P < 0.001 or non-significant (ns, P > 0.05) compared with controls.

The results of the studies on mutant p53 transfected cancer cells are presented in the updated **Figure 4** and new **Supplementary Figure 12**:

Figure 4. ADH-6 causes death of cancer cells bearing mutant, but not WT, p53. (a–d) Effects of ADH-6 on viability of cancer cells bearing WT or mutant p53. (a–c) MIA PaCa-2 (mutant R248W p53) (a), MCF-7 (WT p53) (b), and SK-BR-3 (mutant R175H p53) (c), cells treated with increasing oligopyridylamide concentrations for 24 or 48 h. (d–f) p53 null Saos-2 cells before (d) and after transfection with p53 mutants, R248W (e) or R175H (f), treated with increasing concentrations of ADH-6 for 24 or 48 h. Cell viability in (a–f) was assessed using the MTS assay, with the % viability determined from the ratio of the absorbance of the treated cells to the control cells (n = 3). (g,h) Flow cytometry analysis of annexin V/propidium iodide (PI) staining of MIA PaCa-2 cells that were treated with vehicle (control), or 5 μM ADH-1, ReAcP53 or ADH-6, for 24 h (g). The bottom left quadrant (annexin V-/PI-) represents live cells; bottom right (annexin V+/PI-), early apoptotic cells; top right (annexin V+/PI+), late apoptotic cells; and top left (annexin V-/PI+), necrotic cells. A summary of the incidence of early/late apoptosis and necrosis in the different treatment groups determined from the flow cytometry analysis of annexin V/PI staining (n = 4) (h). (i) Cell cycle distribution of MIA PaCa-2 cells treated with vehicle (control), or 5 μM ADH-1, ReAcP53 or ADH-6, for 6 h as determined by measurement of DNA content using flow cytometry (n = 4). *P < 0.05, **P < 0.01, ***P < 0.001 or non-significant (ns, P > 0.05) compared with controls.

Supplementary Figure 12. Effects of aggregation-prone mutant p53 transfection on cancer cell susceptibility to ADH-6 mediated cytotoxicity. (a) Verification of successful transfection of p53 null Saos-2 and SKOV-3 cells with mutant R248W or R175H p53. The vectors were purified from the transfected cells, and the amplified amp^r PCR product was electrophoresed on a 3% agarose gel containing GelGreen Nucleic Acid Stain. (b,c) Viability of SKOV-3 cells before (b) and after transfection with mutant R248W p53 (c) treated with increasing concentrations of ADH-6 for 24 or 48 h. Cell viability in (b,c) was assessed using the MTS assay, with the % viability determined from the ratio of the absorbance of the treated cells to the control cells (n = 4). *P < 0.05, **P < 0.01, ***P < 0.001 or non-significant (ns, P > 0.05) for comparisons with controls.

Minor Concerns

Concern #1. Figures 6A and 6D: Please explain better the different analyses from which these expression matrices were generated.

Response: As per the Reviewer’s suggestion, the relevant sections of the **Revised Manuscript** have been edited to better explain the analyses used to generate the heatmaps in **Figure 5a** and **d** (**Figure 6a** and **d** in the original version of the manuscript):

RESULTS; ADH-6 induces transcriptional reactivation of p53:

“In order to further probe the effects of ADH-6 treatment, the expression patterns of the 485 DEGs identified were analyzed using hierarchical clustering (performed by the JMP Genomics software) following variance-stabilizing transformation (VST) of the count data. As shown in the heatmap, distinctive clustering patterns were observed, with 196 DEGs identified as upregulated, while 289 were downregulated, in the ADH-6 treatment group relative to the vehicle-treated controls (ADH-6/C) (Figure 5a). To characterize the functions of these DEGs, gene ontology (GO) analysis was performed, focusing specifically on the ‘biological process’ category (Figure 5b). With a P-value < 0.05 (normalized to -log10) cut-off, several biological process enrichments were identified, the top five of which were related to regulation of cell cycle (GO:0051726), cell cycle arrest (GO:0007050), cell proliferation (GO:0008283), regulation of apoptosis (GO:0042981) and aging (GO:0007568) (Figure 5b). A Venn diagram was then used to delineate overlap of DEGs between the selected processes in order to remove redundancies (Figure 5c). Based on the Venn diagram, the selected enrichments were further refined into a finalized DEGs list, as displayed in the heatmap (scaled to log2 count per million reads (log2cpm_voom)) generated using DESeq2 (Figure 5d). From the heatmap, 25 DEGs were identified as upregulated, whereas 49 were downregulated, in ADH-6/C.”

FIGURE LEGENDS:

“Figure 5. Transcriptome analysis of oligopyridylamide-treated MIA PaCa-2 cells. (a) Variance-stabilized count data heatmap showing clustering patterns of differentially expressed genes (DEGs) identified based on statistical significance of $P\text{-adj} < 0.05$ from the ADH-6 treatment group relative to vehicle-treated controls (C) (denoted ADH-6/C). The ADH-1 treatment group is included for comparison. The adjacent legend indicates the scale of expression, with red signifying upregulation, and blue downregulation, in ADH-6/C. **(b)** Gene ontology (GO) analysis of ADH-6/C showing the top five ‘biological process’ term enrichments based on a cut-off of $P\text{-value} < 0.05$ (normalized to $-\log_{10}$). **(c)** Venn diagram delineating overlap of DEGs between the processes in **(b)**. **(d)** Heatmap (scaled to \log_2 counts per million reads ($\log_2\text{cpm}_{\text{voom}}$)) displaying a finalized list of DEGs based on the Venn diagram. **(e)** Gene expression boxplots of DEGs selected from the heatmap in **(d)**. $*P < 0.05$ for comparison amongst the different treatment groups.”

Concern #2. Did the authors compare the gene expression changes obtained with ADH-6 treatment with those observed previously with ReACp53 in ovarian cancer?

Response: In the **Revised Manuscript**, we compare the changes in expression of established p53 targets in MIA PaCa-2 cells following ADH-6 treatment with those previously reported for ReACp53 in ovarian cancer cells (Soragni et al. 2016). For instance, ADH-6 upregulates p21, Noxa, MDM2 and Bax in MIA PaCa-2 cells (**Figure 5** and **Supplementary Figures 15**), and ReACp53 was reported to upregulate these same targets in ovarian cancer cells (Soragni et al. 2016). Moreover, similar to ReACp53, ADH-6 downregulates p73 (**Figure 5**), whose overexpression is associated with advanced-stage cancer (C.-L. Chen et al. 2000). However, we also observed differences between the oligopyridylamide and the peptide. As an example, ADH-6 treatment led to upregulation of the mevalonate pathway (**Supplementary Figure 16**), while ReACp53 downregulated this pathway (Soragni et al. 2016).

In our study, we also conducted a direct comparison of ADH-6 and ReACp53 by probing their effects on the phosphoproteome in MIA PaCa-2 cells. Interestingly, we observed a comparable reduction of phosphoprotein expression, in a number of pathways, in the ReACp53 and ADH-6 treated cells compared to the ADH-1 treatment group (**Figure 6**). The biological roles of the downregulated phosphoproteins were inferred from published data (this data is summarized in **Supplementary Table 2**). This analysis revealed that the downregulated phosphoproteins can be divided into two major functional groups that positively regulate either DNA replication/repair or cell cycle progression/proliferation, both of which play a critical role in cancer initiation and progression. Importantly, these two groups are known to be directly regulated by p53 (Harris and Levine 2005; Giono and Manfredi 2006; Brady et al. 2011; J. Chen 2016; Williams and Schumacher 2016). This strongly supports a similar mechanism of mutant p53 reactivation for ReACp53 and ADH-6, and suggests that the differences between our results and those reported by Soragni *et al.* are likely due to the differences in cancer models (pancreatic vs ovarian) and p53 mutations (R248W vs R248Q) between the two studies.

Comparisons of changes in expression of p53 targets in ADH-6 treated MIA PaCa-2 cells vs ReACp53 treated ovarian cancer cells are presented in the following paragraphs of the **Revised Manuscript**:

RESULTS; ADH-6 induces transcriptional reactivation of p53:

“In agreement with the ChIP-qPCR results, western blot analysis revealed elevated expression of both p21 and Noxa in ADH-6 treated MIA PaCa-2 cells (Supplementary Figure 15). Interestingly,

we also observed significantly higher expression of p53-inducible MDM2 and proapoptotic Bax in response to ADH-6 treatment (Supplementary Figure 15). Surprisingly, recruitment to the *MDM2* and *BAX* genes was not observed by ChIP-qPCR, which may be a consequence of the interaction of mutant p53 with the WT protein's binding sites on the promoters of these genes being too transient or weak to be detected by the assay (Kaeser and Iggo 2002; Swift and Coruzzi 2017). Of relevance, the mutant p53 disaggregating peptide ReACp53 was also reported to upregulate p21, Noxa, MDM2 and Bax in ovarian cancer cells that harbor another aggregating p53 mutant, R248Q (Soragni et al. 2016). Taken together, these results indicate that ADH-6 specifically targets and reactivates aggregation-prone mutant p53.”

“On the other hand, exposure to ADH-6 led to significant downregulation of p53 targets TP73 and Six1 (Figure 5e). ADH-6 induced a 1-fold downregulation of TP73 compared to controls. Interestingly, downregulation of p73 was observed following treatment of mutant p53-bearing cells with ReACp53 (Soragni et al. 2016). ADH-6 treatment also resulted in a 0.3–0.5-fold downregulation of the SIX1 oncogene relative to both controls and ADH-1 treatment groups.”

RESULTS; ADH-6 downregulates cancer-promoting phosphoproteins:

“We subsequently carried out proteome analysis of oligopyridylamide-treated MIA PaCa-2 cells. We chose to focus specifically on the phosphoproteome (Supplementary Figure 18) as many phosphoproteins are involved in regulating major pathways implicated in cancer (Altomare et al. 2004; Emaduddin et al. 2008; Takano et al. 2010; Ardito et al. 2017; Cicenas 2008). Unsupervised hierarchical clustering (UHC) revealed two distinct expression profiles pertaining to the ADH-1 and the ReACp53/ADH-6 treatment groups (Figure 6a). PCA analysis further highlighted that the main source of variation in the two groups was the peptide/compound treatment (Figure 6a,b). Moreover, compared to ADH-1, most of the phosphoproteins downregulated upon ReACp53 treatment were also downregulated in the ADH-6 samples (Figure 6c). Next, we carried out GSEA of hallmark signatures for the differentially expressed phosphoproteins. We observed a comparable reduction of phosphoprotein expression, in a number of pathways, in the ReACp53 and ADH-6 treated cells compared to the ADH-1 treatment group (Figure 6d).

The biological roles of the downregulated phosphoproteins were inferred from published data (Supplementary Table 2), and revealed that these proteins can be divided into two major functional groups that positively regulate either DNA replication/repair or cell cycle progression/proliferation, although many downregulated proteins in the ADH-6 treatment group had overlapping gene signatures (Figure 6e). Importantly, these two major groups, which play a critical role in cancer initiation and progression, are known to be directly regulated by p53 (Harris and Levine 2005; Giono and Manfredi 2006; Brady et al. 2011; J. Chen 2016; Williams and Schumacher 2016). This strongly supports that both the ReACp53 and ADH-6 treatments result in reactivation of p53, which downregulates key cancer-promoting phosphoproteins (Supplementary Table 2a). Indeed, numerous studies have reported that p53 directly downregulates the expression of key positive regulators of cancer in the p53 pathway, such as CD44 (Godar et al. 2008) and PCNA (Shivakumar et al. 1995; Saifudeen et al. 2002), as well as the oncoproteins c-Myc (Levy et al. 1993; Sachdeva et al. 2009), E2F (Timmers et al. 2007; Welch, Chen, and Stallings 2007) and mTORC1 (Stambolic et al. 2001; Feng et al. 2005; Hasty et al. 2013) (Supplementary Figure 19). Interestingly, only a few upregulated phosphoproteins were identified in our phosphoproteome screen (Figure 6f; Supplementary Table 2b). Together, the transcriptomic and proteomic analyses clearly demonstrate that ADH-6-mediated dissociation of mutant p53 amyloid-like aggregates in cancer cells restores p53 function, leading to cell cycle arrest and activation of apoptosis (Figure 4).”

Concern #3. The description of the various functions of the genes deregulated by ADH-6 treatment could be moved from the results to the discussion.

Response: In the original draft of the manuscript, most of the functions of the genes deregulated by ADH-6 were described in the **Discussion**. However, we felt that the **Discussion** was overly long, and consequently made the decision to move the descriptions to the **Results** instead. In the **Revised Manuscript**, the number of genes highlighted and discussed has increased due to the new ChIP-qPCR and western blot experiments added, and moving the descriptions of the functions of the genes to the **Discussion** now increases its length to an unwieldy 5+ pages. Moreover, our aim is to tie in the numerous experiments and analyses of the study into a cohesive narrative in the **Discussion**. By incorporating the gene function descriptions in the **Discussion**, we feel that the narrative loses focus somewhat.

However, we are inclined to agree with the Reviewer that these descriptions are not ideally situated in the **Results**. We are also aware that, with all the additional experiments and analyses, the length of the **Revised Manuscript** has increased considerably. Therefore, to address the Reviewer's concern and to reduce the length of the **Revised Manuscript**, we have moved the gene function descriptions to the **Revised Supplementary Material**.

In addressing the Reviewer's concern, descriptions of the gene functions have been removed from the relevant paragraphs of the **Results**, which now read as follows:

RESULTS; ADH-6 induces transcriptional reactivation of p53:

“To confirm that ADH-6 rescues p53 function, we first used ChIP-qPCR to detect recruitment of mutant R248W p53 to the WT protein's binding sites on promoters/enhancers of target genes in oligopyridylamide treated MIA PaCa-2 cells (primers used are listed in Supplementary Table 1). Treatment with ADH-6 led to binding of R248W to *CDKN1A* (also known as *P21*) (Supplementary Figure 14a). Importantly, we did not detect significant binding of the p53 homologs, p63 and p73, to *CDKN1A* under the same experimental conditions (Supplementary Figure 14a), supporting the notion that ADH-6 activity is mutant p53 dependent. Interaction of R248W with other well-established primary p53 transcriptional targets, *PIG3* and *NOXA*, was also observed in ADH-6 treated MIA PaCa-2 cells (Supplementary Figure 14b,c). (Please note that functions of the genes referenced in this section are described in Supplementary Section 4.)

In agreement with the ChIP-qPCR results, western blot analysis revealed elevated expression of both p21 and Noxa in ADH-6 treated MIA PaCa-2 cells (Supplementary Figure 15). Interestingly, we also observed significantly higher expression of p53-inducible MDM2 and proapoptotic Bax in response to ADH-6 treatment (Supplementary Figure 15). Surprisingly, recruitment to the *MDM2* and *BAX* genes was not observed by ChIP-qPCR, which may be a consequence of the interaction of mutant p53 with the WT protein's binding sites on the promoters of these genes being too transient or weak to be detected by the assay (Kaeser and Iggo 2002; Swift and Coruzzi 2017). Of relevance, the mutant p53 disaggregating peptide ReACp53 was also reported to upregulate p21, Noxa, MDM2 and Bax in ovarian cancer cells that harbor another aggregating p53 mutant, R248Q (Soragni et al. 2016). Taken together, these results indicate that ADH-6 specifically targets and reactivates aggregation-prone mutant p53.”

“The observed expression patterns strongly support transcriptional activation of p53 by ADH-6. In agreement with the ChIP-qPCR analysis, treatment of MIA PaCa-2 cells with ADH-6 resulted in significant (0.7–1.3-fold) upregulation of *CDKN1A* relative to both the ADH-1 and control groups (Figure 5e). ADH-6 treatment also significantly upregulated other p53 target genes that

are important mediators of cell cycle arrest and apoptosis (Shahbazi, Lock, and Liu 2013; Elkeles et al. 1999; Yu et al. 2007, 1) (Figure 5e). For instance, a 0.4-fold upregulation of the tumor protein p53-inducible nuclear protein 1 (*TP53INP1*) was observed in the ADH-6 treatment group compared to controls. Likewise, ADH-6 induced a 0.6–1-fold upregulation of *FOS* relative to treatment with vehicle or ADH-1. Finally, treatment with ADH-6 resulted in a 0.9–1.3-fold upregulation of *EGR1* compared to both the vehicle and ADH-1 treatment groups. On the other hand, exposure to ADH-6 led to significant downregulation of p53 targets TP73 and Six1 (Figure 5e). ADH-6 induced a 1-fold downregulation of TP73 compared to controls. Interestingly, downregulation of p73 was observed following treatment of mutant p53-bearing cells with ReAcP53 (Soragni et al. 2016). ADH-6 treatment also resulted in a 0.3–0.5-fold downregulation of the *SIX1* oncogene relative to both controls and ADH-1 treatment groups.”

The descriptions have been moved to the relevant sections of the **Revised Supplementary Material**:

SUPPLEMENTARY SECTION 4.1.; Chromatin immunoprecipitation (ChIP) with quantitative real-time PCR (ChIP-qPCR) assay:

“ChIP-qPCR (primers used are listed in Supplementary Table 1) revealed that treatment of MIA PaCa-2 cells with ADH-6 resulted in binding of mutant R248W p53 to the WT protein’s binding sites on promoters/enhancers of target genes, *CDKN1A* (also known as *P21*), *PIG3* and *NOXA* (Supplementary Figure 14). p21, a cyclin-dependent kinase inhibitor, is required for p53-mediated cell cycle arrest in response to DNA damage and other cellular stresses (Abbas and Dutta 2009; Karimian, Ahmadi, and Yousefi 2016). *PIG3* (p53-induced gene 3) is involved in both the early cellular response to DNA damage and p53-induced apoptosis (Polyak et al. 1997; Lee et al. 2010). *NOXA* (Latin for damage) encodes a member of the Bcl-2 family of apoptosis regulator proteins; upon p53-induced expression, Noxa localizes to mitochondria, where the protein binds to and inhibits anti-apoptotic Bcl-2 family members (Aubrey et al. 2018). Notably, Noxa has also been identified as a key determinant of cytotoxic response of cancer cells to standard chemotherapy and targeted cancer therapy (Albert, Brinkmann, and Kashkar 2014; Montero et al. 2019).

The ChIP-qPCR results were confirmed using western blot, which showed elevated expression of both p21 and Noxa in ADH-6 treated MIA PaCa-2 cells (Supplementary Figure 15). Interestingly, we also observed significantly higher expression of p53-inducible MDM2 and Bax in response to ADH-6 treatment (Supplementary Figure 15). Bax is another member of the Bcl-2 family, which, once activated, induces permeabilization of the outer mitochondrial membrane, leading to release of cytochrome c and activation of the apoptosis initiator caspase-9 (Oda et al. 2000). Surprisingly, recruitment to the *MDM2* and *BAX* genes was not observed by ChIP-qPCR. This could be due to the interaction of mutant p53 with these genes being too transient or weak to be detected by the assay (Kaeser and Iggo 2002; Swift and Coruzzi 2017).”

SUPPLEMENTARY SECTION 4.2.; Supplementary transcriptome analysis:

“RNA-Seq analysis showed that ADH-6 treatment of mutant R248W p53-bearing MIA PaCa-2 cells led to significant upregulation of p53 target genes, *CDKN1A*, *TP53INP1*, *FOS* and *EGR1* (Figure 5e). Upon translation, tumor protein p53-inducible nuclear protein 1 (*TP53INP1*) forms a protein complex with homeodomain-interacting protein kinase-2 (HIPK2) or protein kinase C δ (PKC δ) that phosphorylates p53 at residue S46 (Shahbazi, Lock, and Liu 2013). This stabilizes p53 and enhances its activity, which leads to transcriptional activation of target genes, such as *CDKN1A*, and subsequent cell cycle arrest and apoptosis (Shahbazi, Lock, and Liu 2013). The first intron of the *FOS* gene contains a p53-responsive element, and overexpression of p53 has been shown to induce *FOS* upregulation, which contributes to both p53-mediated apoptosis and

cell cycle arrest (Elkeles et al. 1999). EGR1 is a transcriptional regulator that can be directly activated by p53 via a non-consensus p53 binding site on the *EGR1* promoter. Although EGR1 has a range of roles, its genotoxic stress-induced upregulation results in apoptosis in most mammalian cells (Yu et al. 2007, 1).

On the other hand, ADH-6 caused significant downregulation of *TP73* and *SIX1* compared to controls (Figure 5e). Of note, downregulation of p73, whose overexpression is associated with advanced-stage cancer (C.-L. Chen et al. 2000), was also observed following treatment of mutant p53-bearing cells with ReACp53 (Soragni et al. 2016). *SIX1* is a homeodomain-containing transcription factor that regulates cell migration, invasion and proliferation in progenitor cell populations, but is not expressed in normal adult tissue (Towers et al. 2015). Importantly, *SIX1* is re-expressed in many cancers and acts as a p53 down-regulator through an MDM2-independent pathway (Towers et al. 2015).

Other enrichments, such as isoprenoid biosynthesis (GO:0008299), lipid biosynthesis (GO:0008610) and cholesterol biosynthesis (GO:0006695) were also identified in ADH-6 treated cells (Supplementary Figure 16c,d). The heatmap generated from genes involved in these processes revealed upregulation of the mevalonate pathway, which is essential for cancer cell survival and growth (Mullen et al. 2016). Of relevance, the mevalonate pathway is mediated by non-aggregated mutant p53 (Freed-Pastor et al. 2012). Thus, ADH-6-induced release of overexpressed mutant p53 from the cellular inclusions likely leads to activation of some oncogenic pathways. However, it appears that the bulk of rescued mutant p53 behaves similar to the WT protein and upregulates major tumor-suppressive pathways, which overwhelms the pro-tumor activities and results in the observed inhibition of proliferation via cell cycle arrest and induction of apoptosis (Figures 4,5 and Supplementary Figures 11,17).”

Response to Reviewer #3

This is a comprehensive paper describing the synthesis of a new compound, a tripyridylamide, ADH-6, that binds to mutant p53 causing it to reduce its state of aggregation and be reactivated as a tumor suppressor. The compound induces tumor regression only in p53 wild type tumors in vivo and does not act on p53 wild type cancers. The work builds on earlier studies on a peptide known as ReACp53 that is reported to act by a similar mechanism of blocking and reversing aggregation. The work here though represents a significant advance because the ADH-6 compound is much more stable and potent than ReACp53 because of its chemical nature. On the face of it then we have a remarkable new medicine that can correct the effects of mutations in p53 and show highly selective tumor cell killing.

Response: We thank the Reviewer for the positive comments regarding the manuscript. We are grateful for the feedback provided and have worked diligently to address all of the concerns. In the following pages, we respond point-by-point to the Reviewer's comments. We believe, and hope the Reviewer agrees, that the paper is much improved in content and clarity.

Concern #1. Of especial importance is the demonstration that expression of the mutant p53 allele in cells that do not normally express the protein (SKOV3 cells) makes them sensitive to ADH-6 Figure 5c,d. This experiment is the strongest direct proof of the proposed mechanism in the manuscript. Given the huge number of screens done in the past to obtain such compounds and their comprehensive failure to be validated which has usually revealed other mechanisms of action the authors do need to explore every avenue of validation of "on target" activity. Extension of the SKOV 3 study using other null cell lines , other mutants and inducible expression systems would add additional weight to the data.

Response: To address the Reviewer's concern, we have transfected p53 null human ovarian cancer SKOV-3 cells with another commonly occurring aggregation-prone p53 mutant, R175H. Similar to the results of transfecting SKOV-3 cells with mutant R248W p53 reported in the original version of the manuscript, transfection with R175H rendered SKOV-3 cells highly susceptible to ADH-6 mediated cytotoxicity.

We have also tested another p53 null cell line, human bone cancer Saos-2. Saos-2 cells were not adversely affected by ADH-6. However, following transfection with mutant R248W or R175H p53, the oligopyridylamide induced significant toxicity in the cells. These additional experiments strongly support our original conclusion that ADH-6 induces its toxic effects in cancer cells via targeting of aggregation-prone mutant p53.

In addressing the Reviewer's concern, we have expanded the relevant sections of the **Revised Manuscript**, which now read as follows:

RESULTS; ADH-6 causes selective cytotoxicity in cancer cells bearing mutant p53:

“Notably, ADH-6 did not adversely affect viability of p53 null human bone cancer Saos-2 cells (Figure 4d). However, transfecting the cells with mutant R248W or R175H p53 rendered them highly susceptible to ADH-6 mediated cytotoxicity ($EC_{50} = 2.0 \pm 0.2$ and $2.3 \pm 0.2 \mu\text{M}$ for R248W and R175H, respectively, at 48 h incubation) (Figure 4e,f and Supplementary Figure 12a). Similarly, p53 null human ovarian cancer SKOV-3 cells were unaffected by treatment with ADH-

6, but following transfection with R248W or R175H, the oligopyridylamide induced significant toxicity in the cells (Supplementary Figure 12a–d). Together with the screen of cancer cells bearing WT and mutant p53, these results indicate that the observed cytotoxicity of ADH-6 in cancer cells is directly related to the oligopyridylamide's capacity to antagonize mutant p53 amyloid formation.”

DISCUSSION:

“Treatment with ADH-6 resulted in substantial toxicity in mutant R248W p53-bearing MIA PaCa-2 cells (Figure 4). Several lines of evidence from multiple experiments establish that ADH-6-mediated cytotoxicity is directly related to the oligopyridylamide's capacity to antagonize mutant p53 amyloid formation: ... (iv) p53 null Saos-2 and SKOV-3 cells, which were insensitive to ADH-6 treatment, became highly susceptible to the cytotoxic effects of the oligopyridylamide upon transfection with mutant R248W or R175H p53 (Figure 4 and Supplementary Figure 12).”

MATERIALS AND METHODS; Cell culture:

“Prior to use, all cell lines were authenticated and tested for mycoplasma contamination by Charles River Laboratories (Margate, United Kingdom). The following cell lines were purchased from American Type Culture Collection (ATCC; Manassas, VA): human bone (U-2 OS [ATCC no. HTB-96] and Saos-2 [HTB-85]), brain (U-87 MG [HTB-14] and SH-SY5Y [CRL-2266]), breast (HCC70 [CRL-2315], MCF-7 [HTB-22], MDA-MB-175-VII [HTB-25], MDA-MB-231 [HTB-26] and SK-BR-3 [HTB-30]), colon (COLO 320DM [CCL-220], LS123 [CCL-255], LS 174T [CL-188] and WiDr [CCL-218]), leukemia (ARH-77 [CRL-1621] and CESS [TIB-190]), lung (A549 [CCL-185], NCI-H1770 [CRL-5893], NCI-H1882 [CRL-5903], NCI-H2342 [CRL-5941] and NCI-H748 [CRL-5841]), ovarian (OVCAR-3 [HTB-161] and SKOV-3 [HTB-77]), pancreatic (MIA PaCa-2 [CRL-1420]), renal (CAKI-1 [HTB-46]) and gastric (AGS [CRL-1739]), cancer cells. The cells were cultured in medium (DMEM, RPMI 1640 or Ham's Nutrient Mixture F12; Sigma) supplemented with 10% fetal bovine serum (FBS; GE Healthcare Life Sciences, Logan, UT), 4 mM L-glutamine, 1 mM sodium pyruvate and 1% penicillin/streptomycin (all from Sigma) in 5% CO₂ at 37 °C, and viability was monitored regularly using the Trypan Blue exclusion test on a TC20 automated cell counter (Bio-Rad, Hercules, CA). Once the cells reached ~95% confluence, they were split (using 0.25% trypsin-EDTA; Sigma) into fractions and propagated or used in experiments.”

MATERIALS AND METHODS; Transfection of p53 null cells with mutant p53:

“Human bone cancer Saos-2 and ovarian cancer SKOV-3 cells were plated at a density of 1×10^5 cells/well in 2 mL complete medium in 6-well plates and cultured for 48 h. For each well of cells to be transfected, 2.5 µg pCMV-Neo-Bam p53 R175H or pCMV-Neo-Bam p53 R248W vectors (plasmids 16436 or 16437, respectively; Addgene, Watertown, MA) containing the *amp^r* gene was diluted in 500 µL Opti-MEM Reduced Serum Medium in an Eppendorf tube, to which 10 µL Lipofectamine LTX Reagent was added, mixed gently and incubated for 30 min at room temperature. The medium in the wells was then replaced with the Opti-MEM Reduced Serum Medium containing the DNA–Lipofectamine LTX Reagent complexes and the cells were incubated in 5% CO₂ at 37 °C for 24 h. Subsequently, the cells were harvested and plated in T25 flasks (Corning). After culturing until ~95% confluence was reached, the cells were selected with 400 µg/mL G418 sulfate-supplemented medium and maintained thereafter in medium containing 100 µg/mL G418 sulfate.

To verify that the transfection was successful, the pCMV-Neo-Bam p53 R248W and pCMV-Neo-Bam p53 R175H vectors containing the *amp^r* gene were purified from the transfected SKOV-3 and Saos-2 cells using the Qiaprep Miniprep Kit (Qiagen, Hilden, Germany) according

to the manufacturer’s instructions. The miniprep-purified DNA samples were subsequently quantified using a NanoDrop spectrophotometer. PCR was carried out to amplify the *amp^r* genes using the following primers: forward (5’-AGATTATCAAAAAGGATCTTCACCT-3’) and reverse (5’-CCTCGTGATACGCCTATTTTATAG-3’) (Integrated DNA Technologies, Leuven, Belgium). Each PCR reaction was prepared in 50 μ L volume containing: 1 μ L of each primer (0.2 μ M final primer concentration), 50 ng DNA, 25 μ L Taq 2 \times PCR Master Mix (New England Biolabs, Ipswich, MA) and nuclease-free water. The conditions for each PCR reaction were: initial denaturation at 95 $^{\circ}$ C for 30 s, 35 cycles of denaturation at 95 $^{\circ}$ C for 30 s, annealing at 50 $^{\circ}$ C for 30 s, and extension at 68 $^{\circ}$ C for 1 min. As a negative control, a PCR reaction was carried out with the same reagents sans DNA. The amplified products were then resolved by electrophoresis using a 3% agarose gel prepared in 1 \times TAE (Tris/acetic acid/EDTA) buffer (Bio-Rad) containing GelGreen Nucleic Acid Stain (Biotium, Fremont, CA), which was visualized on an E-Gel Imager Blue-Light Base (Thermo Fisher Scientific). The presence of the successfully amplified *amp^r* PCR products was confirmed using a standard 500 bp ladder.”

The results of the studies on mutant p53 transfected cancer cells are presented in the updated **Figure 4** and new **Supplementary Figure 12**:

Figure 4. ADH-6 causes death of cancer cells bearing mutant, but not WT, p53. (a–d) Effects of ADH-6 on viability of cancer cells bearing WT or mutant p53. (a–c) MIA PaCa-2 (mutant R248W p53) (a), MCF-7 (WT p53) (b), and SK-BR-3 (mutant R175H p53) (c), cells treated with increasing oligopyridylamide concentrations for 24 or 48 h. (d–f) p53 null Saos-2 cells before (d) and after transfection with p53 mutants, R248W (e) or R175H (f), treated with increasing concentrations of ADH-6 for 24 or 48 h. Cell viability in (a–f) was assessed using the MTS assay, with the % viability determined from the ratio of the absorbance of the treated cells to

the control cells (n = 3). (g,h) Flow cytometry analysis of annexin V/propidium iodide (PI) staining of MIA PaCa-2 cells that were treated with vehicle (control), or 5 μM ADH-1, ReACp53 or ADH-6, for 24 h (g). The bottom left quadrant (annexin V-/PI-) represents live cells; bottom right (annexin V+/PI-), early apoptotic cells; top right (annexin V+/PI+), late apoptotic cells; and top left (annexin V-/PI+), necrotic cells. A summary of the incidence of early/late apoptosis and necrosis in the different treatment groups determined from the flow cytometry analysis of annexin V/PI staining (n = 4) (h). (i) Cell cycle distribution of MIA PaCa-2 cells treated with vehicle (control), or 5 μM ADH-1, ReACp53 or ADH-6, for 6 h as determined by measurement of DNA content using flow cytometry (n = 4). *P < 0.05, **P < 0.01, ***P < 0.001 or non-significant (ns, P > 0.05) compared with controls.

Supplementary Figure 12. Effects of aggregation-prone mutant p53 transfection on cancer cell susceptibility to ADH-6 mediated cytotoxicity. (a) Verification of successful transfection of p53 null Saos-2 and SKOV-3 cells with mutant R248W or R175H p53. The vectors were purified from the transfected cells, and the amplified amp^r PCR product was electrophoresed on a 3% agarose gel containing GelGreen Nucleic Acid Stain. (b,c) Viability of SKOV-3 cells before (b) and after transfection with mutant R248W p53 (c) treated with increasing concentrations of ADH-6 for 24 or 48 h. Cell viability in (b,c) was assessed using the MTS assay, with the % viability determined from the ratio of the absorbance of the treated cells to the control cells (n = 4). *P < 0.05, **P < 0.01, ***P < 0.001 or non-significant (ns, P > 0.05) for comparisons with controls.

Concern #2. One of the most powerful approaches is the use of large cell line panels as the existing limited data look so promising an extended analysis of many more cell lines would be very convincing. This has been done for example for the small molecule MDM2 inhibitors where an almost perfect correlation between p53 wild type status and drug sensitivity has emerged. As a second approach a CRISPR knockout screen can also be very helpful as this has recently demonstrated that many small molecule drugs are off target (Lin et al., *Sci. Transl. Med.* 11, eaaw8412 (2019)).

Response: As suggested by the Reviewer, we have tested the efficacy of ADH-6 in a wider range of cancer cell lines (as many with confirmed p53 status as we could acquire). This expanded screen contains more than twice as many cell lines with three times as many mutations as in the original screen. The cells lines tested include:

- i. 11 cancer cell lines bearing wild-type p53 (compared to 5 cell lines in the original screen): Human bone (U-2 OS), brain (U-87 MG and SH-SY5Y), breast (MCF-7 and MDA-MB-175-VII), colon (LS 174T), renal (CAKI-1), leukemia (CESS), lung (A549 and NCI-H1882) and gastric (AGS), cancer cells.
- ii. 12 cancer cells bearing 6 different aggregation-prone p53 mutations (compared to 5 cell lines bearing 2 mutations in the original screen): R248W (colon: COLO 320DM; lung: NCI-H1770; pancreatic: MIA PaCa-2), R248Q (breast: HCC70; ovarian: OVCAR-3), R175H (breast: SK-

BR-3; colon: LS123), R273H (colon: WiDr; leukemia: ARH-77), Y220C (lung: NCI-H748 and NCI-H2342) or R280K (breast: MDA-MB-231), cancer cells.

Treatment of the cancer cells bearing WT p53 with ADH-6 yielded no significant toxicity. Conversely, we observed a substantial (~75–95%) decrease in viability of cancer cells harboring aggregation-prone p53 mutants at the same oligopyridylamide concentration and incubation time. The expanded screen further supports the conclusion that ADH-6 induces its toxic effects in cancer cells via targeting of mutant p53.

We have also studied the effects of the oligopyridylamides on SK-BR-3 xenografts (harboring mutant R175H p53) in mice. The results confirm that the oligopyridylamide effectively shrinks aggregation-prone mutant p53 bearing xenografts *in vivo*. Together with the expanded screen, the new tumor studies underline the generality of the effects of ADH-6 by demonstrating the efficacy of the oligopyridylamide at targeting a number of aggregation-prone p53 mutants *in vitro* and *in vivo*.

We thank the Reviewer for the suggested CRISPR/Cas9 knockout screen and for bringing the cited paper to our attention. Unfortunately, due to the current health crisis, which has led to severe restrictions on access to the labs/research facilities, working hours and import of materials, we were unable to perform these experiments. However, we believe that all the experiments that we have been able to do, which are summarized in the response to the Reviewer's **Concern #4**, strongly support our conclusion that ADH-6 specifically targets and dissociates mutant p53 aggregates, which leads to transcriptional reactivation of the protein. Nevertheless, we agree with the Reviewer that the CRISPR/Cas9 knockout screen is an elegant approach that would be a helpful addition to the project, and we plan to incorporate it in the follow-up studies to identify and characterize other protein mimetic inhibitors of cancer-associated mutant p53 amyloid formation.

To address the Reviewer's concern, the relevant sections of the **Revised Manuscript** have been updated as follows:

RESULTS; ADH-6 causes selective cytotoxicity in cancer cells bearing mutant p53:

“Next, we determined the effects of the oligopyridylamides on viability of mutant p53-harboring cancer cells using the MTS assay. A screen of the compounds revealed a strong correlation between their toxicity in mutant R248W p53-bearing MIA PaCa-2 cells (Figure 4a and Supplementary Figure 11a,b) and their capacity to antagonize pR248W aggregation (Figure 1e,f and Supplementary Figure 1b). Indeed, ADH-6, which was the most potent antagonist of pR248W amyloid formation, was also the most toxic compound to MIA PaCa-2 cells. ADH-6 reduced MIA PaCa-2 cell viability in a concentration-dependent manner, with an effective concentration (EC_{50}) of 2.7 ± 0.4 and 2.5 ± 0.1 μ M at 24 and 48 h incubation times, respectively (Figure 4a), which was almost half of that for ReACp53 ($EC_{50} = 5.2 \pm 0.5$ and 4.9 ± 0.3 μ M at 24 and 48 h incubation, respectively) (Supplementary Figure 11d). On the other hand, ADH-1, which did not inhibit pR248W aggregation, had no adverse effect on viability of MIA PaCa-2 cells (Figure 4a). Importantly, the oligopyridylamides, including ADH-6, were completely nontoxic to WT p53-bearing breast cancer MCF-7 cells (Figure 4b and Supplementary Figure 11c). Moreover, treatment of a range of human cancer cells bearing WT p53 (AGS, A549, CAKI-1, CESS, LS 174T, MDA-MB-175-VII, NCI-H1882, SH-SY5Y, U-2 OS and U-87 MG) with ADH-6 yielded no significant toxicity (Supplementary Figure 11e). Conversely, the oligopyridylamide was highly toxic to mutant R175H p53-bearing SK-BR-3 cells ($EC_{50} = 2.6 \pm 0.1$ and 2.5 ± 0.2 μ M at 24 and 48 h incubation, respectively) (Figure 4c). Likewise, we observed a substantial (~75–95%) decrease in viability of human cancer cells harboring other aggregation-

prone p53 mutants (R248W: COLO 320DM and NCI-H1770; R248Q: HCC70 and OVCAR-3; R175H: LS123; R273H: WiDr and ARH-77; Y220C: NCI-H748 and NCI-H2342; and R280K: MDA-MB-231) following treatment with ADH-6 (Supplementary Figure 11f).”

RESULTS; *In vivo* administration of ADH-6 causes regression of mutant p53-bearing tumors:

“Having established that ADH-6 potently abrogates mutant p53 amyloid formation and restores WT-like tumor suppressor function *in vitro*, we next assessed the *in vivo* efficacy of the oligopyridylamide (Figure 7 and Supplementary Figures 20,21). Following intraperitoneal injection, ADH-6 quickly entered circulation, with the peak concentration in serum mice (~21 µg/mL) occurring at 2 h post-injection (Figure 7a). The *in vivo* circulation half-life (Meibohm and Derendorf 1997) of ADH-6 ($t_{1/2}$ ~3.6 h) was much longer than that of ReACp53 ($t_{1/2}$ ~1.5 h) (Soragni et al. 2016), or other chemotherapeutics of a comparable size, such as doxorubicin ($t_{1/2}$ < 30 min) (Palanikumar, Al-Hosani, Kalmouni, Nguyen, et al. 2020) or paclitaxel ($t_{1/2}$ ~1.7 h) (Sparreboom et al. 1997). Moreover, ADH-6 was detected in the plasma up to 48 h after administration, whereas ReACp53 was eliminated from the bloodstream in 24 h (Soragni et al. 2016). The relatively long *in vivo* circulation time of ADH-6 should facilitate greater accumulation in tumor tissue. Indeed, the amount of ADH-6 in the MIA PaCa-2 xenografts increased continuously over 48 h post treatment (Figure 7b). The high *in vivo* stability suggested that ADH-6 would exhibit potent antitumor activity.

To test this hypothesis, mice bearing MIA PaCa-2 xenografts were treated with ADH-6, ADH-1 or ReACp53 (155.6 µM in PBS) (Supplementary Figure 20a,b). Treatment consisted of intraperitoneal (IP) injections every 2 days, for a total of 12 doses (Supplementary Figure 20b). IP injection was used since this is the preferred administration route for the control ReACp53 peptide (Soragni et al. 2016). As expected, ADH-1 did not affect tumor growth (Supplementary Figure 20d–f). Conversely, both ADH-6 and ReACp53 reduced tumor growth relative to the saline-treated control group. However, of the two treatments, ADH-6 exhibited significantly greater antitumor efficacy (Supplementary Figure 20d–f). The effects of the oligopyridylamides were recapitulated in mice bearing SK-BR-3 xenografts, with ADH-6 again reducing tumor growth substantially compared to ADH-1 (Supplementary Figure 21).”

DISCUSSION:

“Treatment with ADH-6 resulted in substantial toxicity in mutant R248W p53-bearing MIA PaCa-2 cells (Figure 4). Several lines of evidence from multiple experiments establish that ADH-6-mediated cytotoxicity is directly related to the oligopyridylamide’s capacity to antagonize mutant p53 amyloid formation: (i) a second screen, testing the effects of the oligopyridylamides on viability of MIA PaCa-2 cells (Figure 4 and Supplementary Figure 11), revealed a strong correlation between the compounds’ cytotoxicity and their capacity to antagonize pR248W aggregation (Figure 1 and Supplementary Figure 1), and again identified ADH-6 as the most potent compound in the library; (ii) ADH-6 is significantly more toxic to MIA PaCa-2 cells than the control ReACp53 peptide (Figure 4 and Supplementary Figure 11), which reflects their relative capacities to abrogate intracellular mutant p53 aggregation (Figure 3 and Supplementary Figures 7–9); (iii) ADH-6 induced substantial loss of viability in a range of human cancer cells bearing aggregation-prone mutant p53, but was completely nontoxic to WT p53 harboring human cancer cells (Figure 4 and Supplementary Figure 11)...”

DISCUSSION:

“To establish whether the *in vitro* effects of ADH-6 are recapitulated *in vivo*, we evaluated the effects of ADH-6 on mice bearing MIA PaCa-2 (mutant R248W p53) or SK-BR-3 (mutant

R175H p53) tumors, alone or with MCF-7 (WT p53) tumors on the opposite flank as an internal control (Figure 7 and Supplementary Section 5). ReACp53, serving as a positive control, reduced MIA PaCa-2 tumor growth and prolonged survival relative to the vehicle-treated control groups, which is in agreement with the peptide's reported ability to inhibit growth of aggregation-prone mutant p53-bearing tumors (Soragni et al. 2016). However, ADH-6 was markedly more effective at decreasing MIA PaCa-2 tumor volume and mass, and prolonging survival, compared to ReACp53. The greater *in vivo* efficacy of ADH-6 compared to ReACp53 correlates well with their relative capacities to dissociate intracellular mutant p53 aggregates (Figure 3 and Supplementary Figures 7–9) and to induce toxicity in mutant p53-bearing cancer cells (Figure 4 and Supplementary Figure 11), as well as with their relative *in vivo* stabilities. Peptides possess a number of pharmaceutically desirable properties, including the ability to selectively bind to specific targets with high potency, thereby minimizing off-target interactions and reducing the potential for toxicity (Kalmouni, Al-Hosani, and Magzoub 2019; Henning-Knechtel et al. 2020). On the other hand, a major disadvantage of peptides is their low *in vivo* stability (Palanikumar, Al-Hosani, Kalmouni, Nguyen, et al. 2020). Synthetic protein mimetics, with their constrained backbone, are inherently more stable than peptides (Azzarito et al. 2013; Jayatunga, Thompson, and Hamilton 2014), which is reflected in the substantially longer *in vivo* circulation half-life and prolonged presence in the bloodstream of ADH-6 relative to ReACp53. The extended *in vivo* circulation time of ADH-6 facilitates increased accumulation in tumor tissue and enhanced anti-cancer activity. Importantly, ADH-6 exhibited high potency against MIA PaCa-2 and SK-BR-3 tumors but did not affect growth of control MCF-7 xenografts, underlining the oligopyridylamide's specificity for tumors that harbor aggregation-prone mutant p53. Additionally, detailed necropsies of major organs revealed no damage or alterations, demonstrating the lack of toxicity of ADH-6 to healthy tissue. Thus, our results show that ADH-6 effectively shrinks tumors bearing aggregation-prone mutant p53 *in vivo*, without displaying the non-specific toxicity that is common to conventional cancer therapeutics, thereby greatly prolonging survival."

The **Cell culture** section of the **Materials and Methods** has also been updated as shown in the response to the Reviewer's **Concern #1**, while the ***In vivo*** section has been modified as follows:

MATERIALS AND METHODS; *In vivo* tumor inhibition studies:

"For the single (mutant p53-harboring) xenograft inhibition studies, 5×10^5 viable MIA PaCa-2 or SK-BR-3 cancer cells were injected subcutaneously into the right flank of each mouse at age 6–8 weeks. For the dual (WT and mutant p53-bearing) xenograft model, 5×10^5 viable MIA PaCa-2 and MCF-7 cancer cells were injected into the right and left flanks, respectively, of each mouse at age 6–8 weeks. Mice were assessed daily for overt signs of toxicity. Tumor volume was measured via high-precision calipers (Thermo Fisher Scientific) using the following formula:

$$Tumor\ volume\ (mm^3) = \frac{W^2 \times L}{2} \quad (Equation\ 3)$$

where W and L are tumor width and length in mm, respectively (Luo et al. 2013). Mice were euthanized once tumor volume approached burden defined by NYUAD-IACUC.

Once the tumor volume reached $\sim 25\ mm^3$, the mice were randomized into 4 treatment groups ($n = 8$ per group), which were injected intraperitoneally with: vehicle or the indicated concentrations of ADH-1, ReACp53 or ADH-6. Injections were done every 2 days for a total of

12 doses, with the first day of treatment defined as day 0. Body weight and tumor volume were recorded for the duration of treatment, and survival ($n = 4$ per group) was monitored for a total of 30–60 days. At the end of treatment, 4 mice per treatment group were sacrificed and the tumor tissues were isolated to determine the tumor mass.”

The results of the expanded screen are presented in the updated **Supplementary Figure 11**:

Supplementary Figure 11. Effects of the oligopyridylamides on cancer cells harboring WT or mutant (R248W) p53. (a–c) Screen to identify oligopyridylamides that are toxic to cancer cells bearing mutant, but not WT, p53. MIA PaCa-2 (mutant R248W p53) (a,b) or MCF-7 (WT p53) (c) cells were treated with the indicated concentrations of ADH compounds for 24 or 48 h. (d) Effects of control peptide ReAcP53 on cancer cells bearing mutant p53. MIA PaCa-2 cells were treated with the indicated concentrations of ReAcP53 for 24 or 48 h. (e,f) Probing the effects of ADH-6 on viability of different cancer cells bearing WT or mutant p53. Bone (U-2 OS), brain (U-87 MG and SH-SY5Y), breast (MDA-MB-175-VII), colon (LS 174T), renal (CAKI-1), leukemia (CESS), lung (A549 and NCI-H1882) and gastric (AGS), cancer cells harboring WT p53 (e), or aggregation-prone R248W (colon: COLO 320DM; lung: NCI-H1770), R248Q (breast: HCC70; ovarian: OVCAR-3), R175H (colon: LS123), R273H (colon: WiDr; leukemia: ARH-77), Y220C (lung: NCI-H748 and NCI-H2342) and R280K (breast: MDA-MB-231), mutant p53 harboring cancer cells (f), were treated with 10 μM ADH-6 for 48 h. Cell viability (a–f) was assessed using the MTS assay, with the % viability determined from the ratio of the absorbance of the treated cells to the control cells (treated with vehicle alone). (g) Effects of ADH-6 on cell cycle distribution of mutant p53 bearing cancer cells. MIA PaCa-2 cells were treated with vehicle or 5 μM ADH-1, ReAcP53 or ADH-6 for 6 h. Cell cycle distribution of the cells was then evaluated using a cell cycle assay kit (Abcam), with measurements done on a BD FACSAria III cell sorter ($n = 4$). Shown are representative flow cytometry histograms for the different treatment groups. * $P < 0.05$, ** $P < 0.01$, *** $P < 0.001$ or non-significant (ns, $P > 0.05$) compared with controls.

The results of testing ADH-6 in a new xenograft model (SK-BR-3 bearing mutant R175H p53) are shown in the new **Supplementary Figure 21**:

Supplementary Figure 21. Effects of ADH-6 on SK-BR-3 xenografts in vivo. (a,b) A representative mouse bearing SK-BR-3 (mutant R175H p53) xenografts (a) and the treatment schedule (b). Once the tumor volume reached $\sim 25 \text{ mm}^3$, the mice were randomized into the different treatment groups ($n = 8$ per group), which were injected intraperitoneally with vehicle (0.02% DMSO) or 716.4 μM ADH-1 or ADH-6. Injections were done every 2 days for a total of 12 doses, with the first day of treatment defined as day 0. (c) Body weight changes of the tumor-bearing mice in the different treatment groups monitored for the duration of the experiment. (d) Tumor volume growth curves for the SK-BR-3 xenografts in the different treatment groups over 25 days of treatment ($n = 8$ per group). (e,f) Tumor mass analysis for the different treatment groups. After 25 days of treatment, the mice were sacrificed and the tumor tissues were isolated and imaged (e) and subsequently weighed to determine the tumor mass (f). * $P < 0.05$, ** $P < 0.01$, *** $P < 0.001$ or non-significant (ns, $P > 0.05$) for comparison with controls and amongst the different treatment groups.

Concern #3. Finally a CETSA analysis can be used to establish target engagement (Scientific Reports | 7: 13000 | DOI:10.1038/s41598-017-12513-1).

Response: As per the Reviewer's suggestion, binding of ADH-6 to mutant p53 was assessed using the cellular thermal shift assay (CETSA), which measures ligand-induced changes in thermal stability of target proteins (Molina et al. 2013; Savitski et al. 2014). The inherently unstable p53 has a melting temperature (T_m) of $\sim 45^\circ\text{C}$, and mutations destabilize the protein further and lower its T_m by 5–10 $^\circ\text{C}$ (Selivanova and Wiman 2007; Joerger and Fersht 2016). Consistent with these reports, the CETSA generated melting curve in human pancreatic cancer MIA PaCa-2 cells yielded T_m of $39.6 \pm 1.5^\circ\text{C}$ for mutant R248W p53. Treatment of the cells with ADH-6, but not ADH-1, markedly increased the R248W T_m to $48.2 \pm 0.9^\circ\text{C}$. Likewise, CETSA generated melting curves in human breast cancer SK-BR-3 cells revealed that treatment with ADH-6, but not ADH-1, significantly increased T_m of mutant R175H p53 from $38.8 \pm 1.4^\circ\text{C}$ to $45.1 \pm 1.2^\circ\text{C}$. Along with the confocal fluorescence microscopy imaging of colocalization of fluorescently-labeled ADH-6 and antibody-stained mutant p53, the CETSA results confirm that ADH-6 directly interacts with and stabilizes mutant p53, which shifts the folding equilibrium towards the soluble state, leading to dissociation of the protein's inactive amyloid-like cytosolic aggregates.

The results of the new control experiments done to address the Reviewer’s concern are presented in the updated **Figure 3h,i** and new **Supplementary Figure 10**:

Figure 3. ADH-6 dissociates mutant p53 aggregates in cancer cells. (a) Confocal fluorescence microscopy images showing thioflavin S (ThS) staining of mutant p53 (R248W) aggregates in MIA PaCa-2 cells treated with vehicle (0.02% DMSO) or ADH-6 (5 μM) for 0.5 or 6 h. (b) Quantification of ThS-positive MIA PaCa-2 cells after treatment with vehicle or ADH-6. (c) Confocal fluorescence microscopy images of ThS and PAb 240 antibody staining of R248W aggregates in MIA PaCa-2 treated with vehicle or 5 μM ADH-6 for 0.5 or 6 h. (d–f) Quantification of PAb 240-positive MIA PaCa-2 cells after treatment with the indicated concentrations of ADH-1, ReAcP53 or ADH-6 for 0.5 or 6 h relative to controls (vehicle-treated cells). (g) Colocalization of FITC-labeled ADH-6 (ADH-6_{FITC}) with PAb 240-stained R248W aggregates following incubation with the oligopyridylamide (5 μM) for 0.5 or 6 h. Imaging experiments were performed in triplicate and representative images are shown. Colocalization was quantified using Pearson’s correlation coefficient, *r*, which measures pixel-by-pixel covariance in the signal level of two images (Dunn, Kamocka, and McDonald 2011). Scale bar = 5 μm. For quantification, the number of positively stained cells in 3–5 different fields of view are expressed as % of the total number of cells (*n* = 4). (h,i) Cellular thermal shift assay (CETSA) analysis of intracellular target engagement. Melting curves for p53 mutants R248W (h) and R175H (i) in MIA PaCa-2 and SK-BR-3 cells, respectively, in the absence or presence of the oligopyridylamides (*n* = 3). **P* < 0.05, ***P* < 0.01, ****P* < 0.001 or non-significant (ns, *P* > 0.05) for comparisons with controls.

Supplementary Figure 10. Cellular thermal shift assay (CETSA) analysis of target engagement in MIA PaCa-2 and SK-BR-3 cells. (a) Immunoblots of mutant R248W (a) and R175H (b) p53 in the soluble fractions of MIA PaCa-2 and SK-BR-3 cells, respectively, following treatment with vehicle or 5 μ M ADH-1 or ADH-6 for 6 h and then heating to the indicated temperatures for 3 min. Mutant p53 was detected by the anti-p53 antibody DO-7. Densitometric quantification of the immunoblot bands was used to generate the melting curves shown in Figure 3h,i.

In addressing the Reviewer’s concern, the relevant sections of the **Revised Manuscript** have been updated as follows:

RESULTS; ADH-6 dissociates intracellular mutant p53 aggregates:

“In order to ascertain whether ADH-6-mediated dissociation of mutant R248W p53 aggregates is a result of direct interaction with the oligopyridylamide, PAb 240-stained MIA PaCa-2 cells were treated with FITC-labeled ADH-6 (ADH-6_{FITC}) (Figure 3g). Initially, strong colocalization of ADH-6_{FITC} and PAb 240 was observed, indicating direct interaction of the oligopyridylamide with mutant p53 aggregates. Eventually, the extent of colocalization decreased as the PAb 240 staining was reduced due to disaggregation of mutant p53. Intracellular target engagement was further confirmed using the cellular thermal shift assay (CETSA), which measures ligand-induced changes in thermal stability of target proteins (Molina et al. 2013; Savitski et al. 2014). The labile WT p53 has a melting temperature (T_m) of \sim 45 $^{\circ}$ C, and mutations destabilize the protein further and lower its T_m by 5–10 $^{\circ}$ C (Selivanova and Wiman 2007; Joerger and Fersht 2016). Consistent with these reports, CETSA generated melting curves in MIA PaCa-2 and SK-BR-3 cells yielded T_m of 39.6 ± 1.5 and 38.8 ± 1.4 $^{\circ}$ C for p53 mutants R248W and R175H, respectively (Figure 3h,i and Supplementary Figure 10). Treatment of the cells with ADH-6, but not ADH-1, significantly increased T_m to 48.2 ± 0.9 and 45.1 ± 1.2 $^{\circ}$ C for R248W and R175H, respectively (Figure 3h,i and Supplementary Figure 10), indicating strong stabilization of aggregation-prone mutant p53 by the oligopyridylamide. Taken together, these results show that, similar to ReACp53 (Soragni et al. 2016), ADH-6 efficiently enters cells to directly interact with and stabilize mutant p53, which shifts the folding equilibrium towards the soluble state, leading to dissociation of the protein’s inactive amyloid-like cytosolic aggregates.”

DISCUSSION:

“Here, we have extended our protein mimetic-based approach, previously developed to modulate various aberrant PPIs (Kumar and Hamilton 2017; Kumar et al. 2018; 2015), towards mutant p53 aggregation. Screening a focused library of oligopyridylamide-based α -helix mimetics, we identified a cationic tripyridylamide, ADH-6, that potently inhibited oligomerization and amyloid formation of pR248W, a mutant p53 DBD-derived peptide containing both the aggregation-nucleating sequence and the commonly occurring R248W mutation, by stabilizing the peptide’s

native conformation (Figure 1). It should be noted that ADH-6 only modestly antagonized A β amyloid formation (Kumar and Hamilton 2017), indicating a degree of specificity of the oligopyridylamide for mutant p53. Importantly, ADH-6 also dissociated pre-formed pR248W aggregates, and prevented further aggregation of the peptide (Figure 1). Subsequent studies in human pancreatic carcinoma MIA PaCa-2 cells harboring mutant R248W p53 (Figure 3 and Supplementary Figures 7,9), and *N. benthamiana* cells transfected with YFP-tagged WT and R248W p53 DBDs (Supplementary Figure 8), showed that ADH-6 effectively dissociates intracellular mutant p53 amyloid-like aggregates. Evidence that these effects are due to direct intracellular target engagement comes from the following experiments: (i) confocal microscopy, where extensive colocalization of fluorescently labeled ADH-6 with antibody-stained intracellular mutant p53 was observed; and (ii) CETSA, which demonstrated strong stabilization of p53 mutants in cancer cells by ADH-6 (Figure 3 and Supplementary Figure 10).”

MATERIALS AND METHODS; Cellular thermal shift assay (CETSA):

“CETSA experiments were performed according to a previously published protocol (Molina et al. 2013). Briefly, MIA PaCa-2 or SK-BR-3 cells were treated with vehicle or 5 μ M ADH-1 or ADH-6. After incubation for 6 h, the cells were washed with ice-cold PBS, harvested by trypsinization, centrifuged and resuspended in PBS supplemented with complete Protease Inhibitor Cocktail. Equal amounts of cell suspensions were aliquoted into 0.2 mL PCR microtubes and incubated for 30 min at room temperature. Subsequently, the cell suspension aliquots were heated individually at the indicated temperatures for 3 min (Ristretto Thermal Cycler; VWR, Darmstadt, Germany), followed by cooling for 3 min at on ice. Finally, the cells were lysed using 2 cycles of freeze-thawing, and the soluble fractions were isolated by centrifugation and analyzed by SDS-polyacrylamide gel electrophoresis (SDS-PAGE) followed by western blot as described above. Fold-changes in the immunoblot band densities (normalized to the β -actin controls and using the lowest temperature condition as reference) were plotted as a function of temperature to generate mutant p53 melt curves for the different treatments. Curve fitting and calculation of the melting point, i.e. the temperature at which half of the protein amount has been denatured (T_m), were done using published procedures (Molina et al. 2013; Savitski et al. 2014).”

Concern #4. These additional studies will be needed frankly to convince a skeptical community and justify publication in this journal. If these further validations are successful then this paper would represent a great breakthrough in the field and generate enormous enthusiasm.

Response: We have carried out a number of additional experiments based on the Reviewers’ comments. These have been summarized below (within the three broad areas outlined in the Editors’ summary in the **Decision Letter**):

A. Investigating and confirming the transactivation mechanism (Reviewers #1 and #2):

- 1. Chromatin immunoprecipitation (ChIP) coupled with quantitative PCR (ChIP-qPCR)** was carried out to detect recruitment of mutant R248W p53 to the WT protein’s binding sites on promoters/enhancers of target genes in ADH-6 treated MIA PaCa-2 cells (primers used are listed in **Supplementary Table 1**). Treatment with the oligopyridylamide led to binding of R248W to well-established primary p53 transcriptional targets, *CDKN1A* (also known as *P21*), *PIG3* and *NOX4* (**Supplementary Figure 14**). Importantly, we did not detect significant binding of the p53 homologs, p63 and p73, to *CDKN1A* under the same experimental conditions (**Supplementary Figure 14**), supporting the notion that ADH-6 specifically targets and

reactivates mutant p53.

2. In support of the ChIP-qPCR results, **western blot analysis** revealed elevated expression of both **p21** and **Noxa** in ADH-6 treated MIA PaCa-2 cells (**Supplementary Figure 15**). Of relevance, we also observed significantly higher expression of p53-inducible **MDM2** and **Bax** in response to ADH-6 treatment (**Supplementary Figure 15**).

These additional experiments strongly support our original conclusion that ADH-6 restores p53 transcriptional activity in mutant p53-bearing cancer cells.

B. Extending the work to more cell lines and other mutants (Reviewers #1 and #3):

1. We have expanded the screen to include many more cell lines (**as many with confirmed p53 status as we could acquire**) compared to the original version of the manuscript. This expanded screen contains more than twice as many cell lines with three times as many mutations as in the original screen. The tested cell lines now include:
 - i. **11 cancer cell lines bearing wild-type p53 (compared to 5 cell lines in the original screen):** Human bone (U-2 OS), brain (U-87 MG and SH-SY5Y), breast (MCF-7 and MDA-MB-175-VII), colon (LS 174T), renal (CAKI-1), leukemia (CESS), lung (A549 and NCI-H1882) and gastric (AGS), cancer cells.
 - ii. **12 cancer cells bearing 6 different aggregation-prone p53 mutations (compared to 5 cell lines bearing 2 mutations in the original screen):** mutant R248W p53 (colon: COLO 320DM; lung: NCI-H1770; pancreatic: MIA PaCa-2), R248Q (breast: HCC70; ovarian: OVCAR-3), R175H (breast: SK-BR-3; colon: LS123), R273H (colon: WiDr; leukemia: ARH-77), Y220C (lung: NCI-H748 and NCI-H2342) or R280K (breast: MDA-MB-231), cancer cells.

Treatment of the cancer cells bearing WT p53 yielded no significant toxicity (**Figure 4** and **Supplementary Figure 11**). Conversely, we observed a substantial (~75–95%) decrease in viability of cancer cells harboring aggregation-prone p53 mutants at the same oligopyridylamide concentration and incubation time (**Figure 4** and **Supplementary Figure 11**).

2. We have also transfected p53 null human ovarian cancer **SKOV-3** cells with **another commonly occurring aggregation-prone p53 mutant, R175H**. Similar to the results of transfecting SKOV-3 cells with **mutant R248W p53** reported in the original version of the manuscript, transfection with R175H rendered SKOV-3 cells highly susceptible to ADH-6 mediated cytotoxicity (**Supplementary Figure 12**).
3. Furthermore, we have tested another p53 null cell line, **human bone cancer Saos-2**. Saos-2 cells were not adversely affected by ADH-6. However, following transfection with mutant R248W or R175H p53, the oligopyridylamide induced significant toxicity in the cells (**Figure 4** and **Supplementary Figure 12**).
4. Finally, we have probed the effects of the oligopyridylamides on **SK-BR-3** xenografts (harboring mutant **R175H p53**) in mice. The results confirm that ADH-6 effectively shrinks aggregation-prone mutant p53 bearing tumors *in vivo* (**Supplementary Figure 21**).

Taken together, these additional experiments confirm that the observed cytotoxicity of ADH-6 in cancer cells is indeed mutant p53 dependent. Further, the results underline the generality of the effects of ADH-6 by demonstrating the efficacy of the oligopyridylamide at targeting a range of aggregation-prone p53 mutants *in vitro* and *in vivo*.

C. Demonstrating target engagement (Reviewers #1 and #3):

1. Intracellular target engagement was probed further using the **cellular thermal shift assay (CETSA)**. The labile p53 has a **melting temperature (T_m)** of ~45 °C, and mutations

destabilize the protein further and lower its T_m by 5–10 °C (Selivanova and Wiman 2007; Joerger and Fersht 2016). Consistent with these reports, the CETSA generated melting curve in **MIA PaCa-2 cells** yielded T_m of 39.6 ± 1.5 °C for **mutant R248W p53** (**Figure 3** and **Supplementary Figure 10**). Treatment of the cells with ADH-6, but not ADH-1, markedly increased the R248W T_m to 48.2 ± 0.9 °C (**Figure 3** and **Supplementary Figure 10**).

2. Likewise, CETSA generated melting curves in **SK-BR-3 cells** revealed that treatment with ADH-6 significantly increased T_m of **mutant R175H p53** from 38.8 ± 1.4 °C to 45.1 ± 1.2 °C (**Figure 3** and **Supplementary Figure 10**).

These results confirm that ADH-6 directly interacts with and stabilizes mutant p53, which shifts the folding equilibrium towards the soluble state, leading to dissociation of the protein's inactive amyloid-like cytosolic aggregates.

We sincerely hope the Reviewer agrees that these additional experiments strengthen the paper considerably and justify its publication in *Nature Communications*.

REFERENCES

- Abbas, Tarek, and Anindya Dutta. 2009. "P21 in Cancer: Intricate Networks and Multiple Activities." *Nature Reviews. Cancer* 9 (6): 400–414. <https://doi.org/10.1038/nrc2657>.
- Albert, Marie-Christine, Kerstin Brinkmann, and Hamid Kashkar. 2014. "Noxa and Cancer Therapy." *Molecular & Cellular Oncology* 1 (1). <https://doi.org/10.4161/mco.29906>.
- Altomare, Deborah A., Hui Qin Wang, Kristine L. Skele, Assunta De Rienzo, Andres J. Klein-Szanto, Andrew K. Godwin, and Joseph R. Testa. 2004. "AKT and MTOR Phosphorylation Is Frequently Detected in Ovarian Cancer and Can Be Targeted to Disrupt Ovarian Tumor Cell Growth." *Oncogene* 23 (34): 5853–57. <https://doi.org/10.1038/sj.onc.1207721>.
- Ardito, Fatima, Michele Giuliani, Donatella Perrone, Giuseppe Troiano, and Lorenzo Lo Muzio. 2017. "The Crucial Role of Protein Phosphorylation in Cell Signaling and Its Use as Targeted Therapy (Review)." *International Journal of Molecular Medicine* 40 (2): 271–80. <https://doi.org/10.3892/ijmm.2017.3036>.
- Aubrey, Brandon J., Gemma L. Kelly, Ana Janic, Marco J. Herold, and Andreas Strasser. 2018. "How Does P53 Induce Apoptosis and How Does This Relate to P53-Mediated Tumour Suppression?" *Cell Death & Differentiation* 25 (1): 104–13. <https://doi.org/10.1038/cdd.2017.169>.
- Azzarito, Valeria, Kérya Long, Natasha S. Murphy, and Andrew J. Wilson. 2013. "Inhibition of α -Helix-Mediated Protein–Protein Interactions Using Designed Molecules." *Nature Chemistry* 5 (3): 161–73. <https://doi.org/10.1038/nchem.1568>.
- Baugh, Evan H., Hua Ke, Arnold J. Levine, Richard A. Bonneau, and Chang S. Chan. 2018. "Why Are There Hotspot Mutations in the TP53 Gene in Human Cancers?" *Cell Death & Differentiation* 25 (1): 154–60. <https://doi.org/10.1038/cdd.2017.180>.
- Bouaoun, Liacine, Dmitriy Sonkin, Maude Ardin, Monica Hollstein, Graham Byrnes, Jiri Zavadil, and Magali Olivier. 2016. "TP53 Variations in Human Cancers: New Lessons from the IARC TP53 Database and Genomics Data." *Human Mutation* 37 (9): 865–76. <https://doi.org/10.1002/humu.23035>.
- Brady, Colleen A., Dadi Jiang, Stephano S. Mello, Thomas M. Johnson, Lesley A. Jarvis, Margaret M. Kozak, Daniela Kenzelmann Broz, et al. 2011. "Distinct P53 Transcriptional Programs Dictate Acute DNA-Damage Responses and Tumor Suppression." *Cell* 145 (4): 571–83. <https://doi.org/10.1016/j.cell.2011.03.035>.
- Chen, Chun-Ling, Sin-Ming Ip, Danny Cheng, Ling-Chui Wong, and Hextan Y. S. Ngan. 2000. "P73 Gene Expression in Ovarian Cancer Tissues and Cell Lines." *Clinical Cancer Research* 6 (10): 3910–15.
- Chen, Jiandong. 2016. "The Cell-Cycle Arrest and Apoptotic Functions of P53 in Tumor Initiation and Progression." *Cold Spring Harbor Perspectives in Medicine* 6 (3). <https://doi.org/10.1101/cshperspect.a026104>.
- Cicenas, J. 2008. "The Potential Role of Akt Phosphorylation in Human Cancers." *The International Journal of Biological Markers* 23 (1): 1–9. <https://doi.org/10.5301/jbm.2008.618>.
- Deiry, W. S. el-, T. Tokino, V. E. Velculescu, D. B. Levy, R. Parsons, J. M. Trent, D. Lin, W. E. Mercer, K. W. Kinzler, and B. Vogelstein. 1993. "WAF1, a Potential Mediator of P53 Tumor Suppression." *Cell* 75 (4): 817–25. [https://doi.org/10.1016/0092-8674\(93\)90500-p](https://doi.org/10.1016/0092-8674(93)90500-p).
- Dunn, K.W., M.M. Kamocka, and J.H. and McDonald. 2011. "A Practical Guide to Evaluating Colocalization in Biological Microscopy." *American Journal of Physiology - Cell Physiology* 300: 723–42.
- Elkeles, Adi, Tamar Juven-Gershon, David Israeli, Sylvia Wilder, Amir Zalcenstein, and Moshe Oren. 1999. "The C-Fos Proto-Oncogene Is a Target for Transactivation by the P53 Tumor Suppressor." *Molecular and Cellular Biology* 19 (4): 2594–2600.

- Emaduddin, Muhammad, David C. Bicknell, Walter F. Bodmer, and Stephan M. Feller. 2008. “Cell Growth, Global Phosphotyrosine Elevation, and c-Met Phosphorylation through Src Family Kinases in Colorectal Cancer Cells.” *Proceedings of the National Academy of Sciences of the United States of America* 105 (7): 2358–62. <https://doi.org/10.1073/pnas.0712176105>.
- Feng, Zhaohui, Haiyan Zhang, Arnold J. Levine, and Shengkan Jin. 2005. “The Coordinate Regulation of the P53 and MTOR Pathways in Cells.” *Proceedings of the National Academy of Sciences of the United States of America* 102 (23): 8204–9. <https://doi.org/10.1073/pnas.0502857102>.
- Fine, Jonathan, Janez Konc, Ram Samudrala, and Gaurav Chopra. 2020. “CANDOCK: Chemical Atomic Network-Based Hierarchical Flexible Docking Algorithm Using Generalized Statistical Potentials.” *Journal of Chemical Information and Modeling* 60 (3): 1509–27. <https://doi.org/10.1021/acs.jcim.9b00686>.
- Freed-Pastor, William A., Hideaki Mizuno, Xi Zhao, Anita Langerød, Sung-Hwan Moon, Ruth Rodriguez-Barrueco, Anthony Barsotti, et al. 2012. “Mutant P53 Disrupts Mammary Tissue Architecture via the Mevalonate Pathway.” *Cell* 148 (1–2): 244–58. <https://doi.org/10.1016/j.cell.2011.12.017>.
- Giono, Luciana E., and James J. Manfredi. 2006. “The P53 Tumor Suppressor Participates in Multiple Cell Cycle Checkpoints.” *Journal of Cellular Physiology* 209 (1): 13–20. <https://doi.org/10.1002/jcp.20689>.
- Godar, Samuel, Tan A. Ince, George W. Bell, David Feldser, Joana Liu Donaher, Jonas Bergh, Anne Liu, et al. 2008. “Growth-Inhibitory and Tumor-Suppressive Functions of P53 Depend on Its Repression of CD44 Expression.” *Cell* 134 (1): 62–73. <https://doi.org/10.1016/j.cell.2008.06.006>.
- Harris, Sandra L., and Arnold J. Levine. 2005. “The P53 Pathway: Positive and Negative Feedback Loops.” *Oncogene* 24 (17): 2899–2908. <https://doi.org/10.1038/sj.onc.1208615>.
- Hasty, Paul, Zelton Dave Sharp, Tyler J. Curiel, and Judith Campisi. 2013. “MTORC1 and P53.” *Cell Cycle* 12 (1): 20–25. <https://doi.org/10.4161/cc.22912>.
- Henning-Knechtel, Anja, Sunil Kumar, Cecilia Wallin, Sylwia Król, Sebastian K. T. S. Wärmländer, Jüri Jarvet, Gennaro Esposito, et al. 2020. “Designed Cell-Penetrating Peptide Inhibitors of Amyloid-Beta Aggregation and Cytotoxicity.” *Cell Reports Physical Science* 1 (2): 100014. <https://doi.org/10.1016/j.xcrp.2020.100014>.
- Jayatunga, Madura K. P., Sam Thompson, and Andrew D. Hamilton. 2014. “ α -Helix Mimetics: Outwards and Upwards.” *Bioorganic & Medicinal Chemistry Letters* 24 (3): 717–24. <https://doi.org/10.1016/j.bmcl.2013.12.003>.
- Joerger, Andreas C., and Alan R. Fersht. 2010. “The Tumor Suppressor P53: From Structures to Drug Discovery.” *Cold Spring Harbor Perspectives in Biology* 2 (6). <https://doi.org/10.1101/cshperspect.a000919>.
- Joerger, Andreas C., and Alan R. Fersht. 2016. “The P53 Pathway: Origins, Inactivation in Cancer, and Emerging Therapeutic Approaches.” *Annual Review of Biochemistry* 85 (1): 375–404. <https://doi.org/10.1146/annurev-biochem-060815-014710>.
- Kaesler, M. D., and R. D. Iggo. 2002. “Chromatin Immunoprecipitation Analysis Fails to Support the Latency Model for Regulation of P53 DNA Binding Activity in Vivo.” *Proceedings of the National Academy of Sciences of the United States of America* 99 (1): 95–100. <https://doi.org/10.1073/pnas.012283399>.
- Kalmouni, Mona, Sumaya Al-Hosani, and Mazin Magzoub. 2019. “Cancer Targeting Peptides.” *Cellular and Molecular Life Sciences* 76 (11): 2171–83. <https://doi.org/10.1007/s00018-019-03061-0>.
- Karimian, Ansar, Yasin Ahmadi, and Bahman Yousefi. 2016. “Multiple Functions of P21 in Cell Cycle, Apoptosis and Transcriptional Regulation after DNA Damage.” *DNA Repair* 42: 63–71. <https://doi.org/10.1016/j.dnarep.2016.04.008>.

- Kastan, M. B., Q. Zhan, W. S. el-Deiry, F. Carrier, T. Jacks, W. V. Walsh, B. S. Plunkett, B. Vogelstein, and A. J. Fornace. 1992. "A Mammalian Cell Cycle Checkpoint Pathway Utilizing P53 and GADD45 Is Defective in Ataxia-Telangiectasia." *Cell* 71 (4): 587–97. [https://doi.org/10.1016/0092-8674\(92\)90593-2](https://doi.org/10.1016/0092-8674(92)90593-2).
- Koeppel, Max, Simon J. van Heeringen, Daniela Kramer, Leonie Smeenk, Eva Janssen-Megens, Marianne Hartmann, Hendrik G. Stunnenberg, and Marion Lohrum. 2011. "Crosstalk between C-Jun and TAp73 α/β Contributes to the Apoptosis–Survival Balance." *Nucleic Acids Research* 39 (14): 6069–85. <https://doi.org/10.1093/nar/gkr028>.
- Kouwenhoven, Evelyn N., Simon J. van Heeringen, Juan J. Tena, Martin Oti, Bas E. Dutilh, M. Eva Alonso, Elisa de la Calle-Mustienes, et al. 2010. "Genome-Wide Profiling of P63 DNA-Binding Sites Identifies an Element That Regulates Gene Expression during Limb Development in the 7q21 SHFM1 Locus." *PLoS Genetics* 6 (8): e1001065. <https://doi.org/10.1371/journal.pgen.1001065>.
- Kumar, Sunil, and Andrew D. Hamilton. 2017. " α -Helix Mimetics as Modulators of A β Self-Assembly." *Journal of the American Chemical Society* 139 (16): 5744–55. <https://doi.org/10.1021/jacs.6b09734>.
- Kumar, Sunil, Anja Henning-Knechtel, Mazin Magzoub, and Andrew D. Hamilton. 2018. "Peptidomimetic-Based Multidomain Targeting Offers Critical Evaluation of A β Structure and Toxic Function." *Journal of the American Chemical Society* 140 (21): 6562–74. <https://doi.org/10.1021/jacs.7b13401>.
- Kumar, Sunil, Diana E. Schlamadinger, Mark A. Brown, Joanna M. Dunn, Brandon Mercado, James A. Hebda, Ishu Saraogi, Elizabeth Rhoades, Andrew D. Hamilton, and Andrew D. Miranker. 2015. "Islet Amyloid Induced Cell Death and Bilayer Integrity Loss Share a Molecular Origin Targetable with Oligopyridylamide-Based α -Helical Mimetics." *Chemistry & Biology* 22 (3): 369–78. <https://doi.org/10.1016/j.chembiol.2015.01.006>.
- Lee, J.-H., Y. Kang, V. Khare, Z.-Y. Jin, M.-Y. Kang, Y. Yoon, J.-W. Hyun, et al. 2010. "The P53-Inducible Gene 3 (PIG3) Contributes to Early Cellular Response to DNA Damage." *Oncogene* 29 (10): 1431–50. <https://doi.org/10.1038/onc.2009.438>.
- Levy, N., E. Yonish-Rouach, M. Oren, and A. Kimchi. 1993. "Complementation by Wild-Type P53 of Interleukin-6 Effects on M1 Cells: Induction of Cell Cycle Exit and Cooperativity with c-Myc Suppression." *Molecular and Cellular Biology* 13 (12): 7942–52. <https://doi.org/10.1128/mcb.13.12.7942>.
- Luo, Zhong, Xingwei Ding, Yan Hu, Shaojue Wu, Yang Xiang, Yongfei Zeng, Beilu Zhang, et al. 2013. "Engineering a Hollow Nanocontainer Platform with Multifunctional Molecular Machines for Tumor-Targeted Therapy in Vitro and in Vivo." *ACS Nano* 7 (11): 10271–84. <https://doi.org/10.1021/nn404676w>.
- Meibohm, B., and H. Derendorf. 1997. "Basic Concepts of Pharmacokinetic/Pharmacodynamic (PK/PD) Modelling." *International Journal of Clinical Pharmacology and Therapeutics* 35 (10): 401–13.
- Miyashita, T., and J. C. Reed. 1995. "Tumor Suppressor P53 Is a Direct Transcriptional Activator of the Human Bax Gene." *Cell* 80 (2): 293–99. [https://doi.org/10.1016/0092-8674\(95\)90412-3](https://doi.org/10.1016/0092-8674(95)90412-3).
- Molina, Daniel Martinez, Rozbeh Jafari, Marina Ignatushchenko, Takahiro Seki, E. Andreas Larsson, Chen Dan, Lekshmy Sreekumar, Yihai Cao, and Pär Nordlund. 2013. "Monitoring Drug Target Engagement in Cells and Tissues Using the Cellular Thermal Shift Assay." *Science* 341 (6141): 84–87. <https://doi.org/10.1126/science.1233606>.
- Montero, Joan, Cécile Gstalder, Daniel J. Kim, Dorota Sadowicz, Wayne Miles, Michael Manos, Justin R. Cidado, et al. 2019. "Destabilization of NOXA mRNA as a Common Resistance Mechanism to Targeted Therapies." *Nature Communications* 10 (1): 5157. <https://doi.org/10.1038/s41467-019-12477-y>.

- Mullen, Peter J., Rosemary Yu, Joseph Longo, Michael C. Archer, and Linda Z. Penn. 2016. "The Interplay between Cell Signalling and the Mevalonate Pathway in Cancer." *Nature Reviews Cancer* 16 (11): 718–31. <https://doi.org/10.1038/nrc.2016.76>.
- Muller, Patricia A. J., and Karen H. Vousden. 2014. "Mutant P53 in Cancer: New Functions and Therapeutic Opportunities." *Cancer Cell* 25 (3): 304–17. <https://doi.org/10.1016/j.ccr.2014.01.021>.
- Nakano, K., and K. H. Vousden. 2001. "PUMA, a Novel Proapoptotic Gene, Is Induced by P53." *Molecular Cell* 7 (3): 683–94. [https://doi.org/10.1016/s1097-2765\(01\)00214-3](https://doi.org/10.1016/s1097-2765(01)00214-3).
- Oda, E., R. Ohki, H. Murasawa, J. Nemoto, T. Shibue, T. Yamashita, T. Tokino, T. Taniguchi, and N. Tanaka. 2000. "Noxa, a BH3-Only Member of the Bcl-2 Family and Candidate Mediator of P53-Induced Apoptosis." *Science (New York, N.Y.)* 288 (5468): 1053–58. <https://doi.org/10.1126/science.288.5468.1053>.
- Palanikumar, L., Sumaya Al-Hosani, Mona Kalmouni, Vanessa P. Nguyen, Liaqat Ali, Renu Pasricha, Francisco N. Barrera, and Mazin Magzoub. 2020. "PH-Responsive High Stability Polymeric Nanoparticles for Targeted Delivery of Anticancer Therapeutics." *Communications Biology* 3 (1): 1–17. <https://doi.org/10.1038/s42003-020-0817-4>.
- Palanikumar, L., Sumaya Al-Hosani, Mona Kalmouni, Hadi Omar Saleh, and Mazin Magzoub. 2020. "Hexokinase II-Derived Cell-Penetrating Peptide Mediates Delivery of MicroRNA Mimic for Cancer-Selective Cytotoxicity." *Biochemistry* 59 (24): 2259–73. <https://doi.org/10.1021/acs.biochem.0c00141>.
- Polyak, Kornelia, Yong Xia, Jay L. Zweier, Kenneth W. Kinzler, and Bert Vogelstein. 1997. "A Model for P53-Induced Apoptosis." *Nature* 389 (6648): 300–305. <https://doi.org/10.1038/38525>.
- Sachdeva, Mohit, Shoumin Zhu, Fangting Wu, Hailong Wu, Vijay Walia, Sumit Kumar, Randolph Elble, Kounosuke Watabe, and Yin-Yuan Mo. 2009. "P53 Represses C-Myc through Induction of the Tumor Suppressor MiR-145." *Proceedings of the National Academy of Sciences of the United States of America* 106 (9): 3207–12. <https://doi.org/10.1073/pnas.0808042106>.
- Saifudeen, Zubaida, Jessica Marks, Hong Du, and Samir S. El-Dahr. 2002. "Spatial Repression of PCNA by P53 during Kidney Development." *American Journal of Physiology-Renal Physiology* 283 (4): F727–33. <https://doi.org/10.1152/ajprenal.00114.2002>.
- Savitski, Mikhail M., Friedrich B. M. Reinhard, Holger Franken, Thilo Werner, Maria Fälth Savitski, Dirk Eberhard, Daniel Martinez Molina, et al. 2014. "Tracking Cancer Drugs in Living Cells by Thermal Profiling of the Proteome." *Science* 346 (6205). <https://doi.org/10.1126/science.1255784>.
- Selivanova, G., and K G Wiman. 2007. "Reactivation of Mutant P53: Molecular Mechanisms and Therapeutic Potential | Oncogene." *Oncogene* 26 (15): 2243–54. <https://doi.org/10.1038/sj.onc.1210295>.
- Shahbazi, Jeyran, Richard Lock, and Tao Liu. 2013. "Tumor Protein 53-Induced Nuclear Protein 1 Enhances P53 Function and Represses Tumorigenesis." *Frontiers in Genetics* 4: 80. <https://doi.org/10.3389/fgene.2013.00080>.
- Shivakumar, C V, D R Brown, S Deb, and S P Deb. 1995. "Wild-Type Human P53 Transactivates the Human Proliferating Cell Nuclear Antigen Promoter." *Molecular and Cellular Biology* 15 (12): 6785–93.
- Smith, P. K., R. I. Krohn, G. T. Hermanson, A. K. Mallia, F. H. Gartner, M. D. Provenzano, E. K. Fujimoto, N. M. Goeke, B. J. Olson, and D. C. Klenk. 1985. "Measurement of Protein Using Bicinchoninic Acid." *Analytical Biochemistry* 150 (1): 76–85. [https://doi.org/10.1016/0003-2697\(85\)90442-7](https://doi.org/10.1016/0003-2697(85)90442-7).
- Song, Hoseok, Monica Hollstein, and Yang Xu. 2007. "P53 Gain-of-Function Cancer Mutants Induce Genetic Instability by Inactivating ATM." *Nature Cell Biology* 9 (5): 573–80. <https://doi.org/10.1038/ncb1571>.

- Soragni, Alice, Deanna M. Janzen, Lisa M. Johnson, Anne G. Lindgren, Anh Thai-Quynh Nguyen, Ekaterina Tiourin, Angela B. Soriaga, et al. 2016. "A Designed Inhibitor of P53 Aggregation Rescues P53 Tumor Suppression in Ovarian Carcinomas." *Cancer Cell* 29 (1): 90–103. <https://doi.org/10.1016/j.ccell.2015.12.002>.
- Sparreboom, A., J. van Asperen, U. Mayer, A. H. Schinkel, J. W. Smit, D. K. Meijer, P. Borst, W. J. Nooijen, J. H. Beijnen, and O. van Tellingen. 1997. "Limited Oral Bioavailability and Active Epithelial Excretion of Paclitaxel (Taxol) Caused by P-Glycoprotein in the Intestine." *Proceedings of the National Academy of Sciences of the United States of America* 94 (5): 2031–35. <https://doi.org/10.1073/pnas.94.5.2031>.
- Stambolic, V., D. MacPherson, D. Sas, Y. Lin, B. Snow, Y. Jang, S. Benchimol, and T. W. Mak. 2001. "Regulation of PTEN Transcription by P53." *Molecular Cell* 8 (2): 317–25. [https://doi.org/10.1016/s1097-2765\(01\)00323-9](https://doi.org/10.1016/s1097-2765(01)00323-9).
- Swift, Joseph, and Gloria Coruzzi. 2017. "A Matter of Time - How Transient Transcription Factor Interactions Create Dynamic Gene Regulatory Networks." *Biochimica et Biophysica Acta* 1860 (1): 75–83. <https://doi.org/10.1016/j.bbagr.2016.08.007>.
- Takano, S., K. Sogawa, H. Yoshitomi, T. Shida, K. Mogushi, F. Kimura, H. Shimizu, et al. 2010. "Increased Circulating Cell Signalling Phosphoproteins in Sera Are Useful for the Detection of Pancreatic Cancer." *British Journal of Cancer* 103 (2): 223–31. <https://doi.org/10.1038/sj.bjc.6605734>.
- Timmers, Cynthia, Nidhi Sharma, Rene Opavsky, Baidehi Maiti, Lizhao Wu, Juan Wu, Daniel Orringer, Prashant Tripathi, Harold I. Saavedra, and Gustavo Leone. 2007. "E2f1, E2f2, and E2f3 Control E2F Target Expression and Cellular Proliferation via a P53-Dependent Negative Feedback Loop." *Molecular and Cellular Biology* 27 (1): 65–78. <https://doi.org/10.1128/MCB.02147-06>.
- Towers, Christina G., Anna L. Guarnieri, Doug S. Micalizzi, J. Chuck Harrell, Austin E. Gillen, Jihye Kim, Chu-An Wang, et al. 2015. "The Six1 Oncoprotein Downregulates P53 via Concomitant Regulation of RPL26 and MicroRNA-27a-3p." *Nature Communications* 6 (December). <https://doi.org/10.1038/ncomms10077>.
- Wang, GuoZhen, and Alan R. Fersht. 2012. "First-Order Rate-Determining Aggregation Mechanism of P53 and Its Implications." *Proceedings of the National Academy of Sciences* 109 (34): 13590–95. <https://doi.org/10.1073/pnas.1211557109>.
- Welch, C., Y. Chen, and R. L. Stallings. 2007. "MicroRNA-34a Functions as a Potential Tumor Suppressor by Inducing Apoptosis in Neuroblastoma Cells." *Oncogene* 26 (34): 5017–22. <https://doi.org/10.1038/sj.onc.1210293>.
- Williams, Ashley B., and Björn Schumacher. 2016. "P53 in the DNA-Damage-Repair Process." *Cold Spring Harbor Perspectives in Medicine* 6 (5). <https://doi.org/10.1101/cshperspect.a026070>.
- Woldetsadik, Abiy D., Maria C. Vogel, Wael M. Rabeh, and Mazin Magzoub. 2017. "Hexokinase II-Derived Cell-Penetrating Peptide Targets Mitochondria and Triggers Apoptosis in Cancer Cells." *The FASEB Journal* 31 (5): 2168–84. <https://doi.org/10.1096/fj.201601173R>.
- Wong, K.-B., B. S. DeDecker, S. M. V. Freund, M. R. Proctor, M. Bycroft, and A. R. Fersht. 1999. "Hot-Spot Mutants of P53 Core Domain Evince Characteristic Local Structural Changes." *Proceedings of the National Academy of Sciences* 96 (15): 8438–42. <https://doi.org/10.1073/pnas.96.15.8438>.
- Wu, X., J. H. Bayle, D. Olson, and A. J. Levine. 1993. "The P53-Mdm-2 Autoregulatory Feedback Loop." *Genes & Development* 7 (7A): 1126–32. <https://doi.org/10.1101/gad.7.7a.1126>.
- Xu, Jie, Joke Reumers, José R. Couceiro, Frederik De Smet, Rodrigo Gallardo, Stanislav Rudyak, Ann Cornelis, et al. 2011. "Gain of Function of Mutant P53 by Coaggregation with Multiple Tumor Suppressors." *Nature Chemical Biology* 7 (5): 285–95. <https://doi.org/10.1038/nchembio.546>.
- Yu, J., V. Baron, D. Mercola, T. Mustelin, and E. D. Adamson. 2007. "A Network of P73, P53 and Egr1 Is Required for Efficient Apoptosis in Tumor Cells." *Cell Death and Differentiation* 14 (3): 436–46. <https://doi.org/10.1038/sj.cdd.4402029>.

Reviewers' Comments:

Reviewer #1:

Remarks to the Author:

The authors have properly addressed most of my concerns and the manuscript is significantly improved. Thus, I support its publication in this journal.

Reviewer #2:

Remarks to the Author:

The authors have adequately addressed to the concerns raised previously by this reviewer. The manuscript is suitable for full acceptance.

Reviewer #3:

Remarks to the Author:

The authors have done a great job in responding to the reviewers comments. They have greatly improved the manuscript . The CETSA assays are convincing that p53 has been engaged as a target. The CHIP analysis shows that it is indeed p53 and not p63 or p73 that is binding to the p53 response elements and that p53 target genes are activated. The additional cell line studies and the experiments where sensitivity to the molecule is developed in null cell lines reconstituted with mutant p53 is very well done and convincing. The xenograft data are a great addition. While I would have liked to have seen the CRISPR screen I accept the authors response . I am now convinced that the compound is active and congratulate the authors